# CAIFormer: A Causal Informed Transformer for Multivariate Time Series Forecasting

## Abstract

Most existing multivariate time series forecasting methods adopt an all-to-all paradigm that feeds all variable histories into a unified model to predict their future values without distinguishing their individual roles. However, this undifferentiated paradigm makes it difficult to identify variable-specific causal influences and often entangles causally relevant information with spurious correlations. To address this limitation, we propose an all-to-one forecasting paradigm that predicts each target variable separately. Specifically, we first construct a Structural Causal Model from observational data and then, for each target variable, we partition the historical sequence into four sub-segments according to the inferred causal structure: endogenous, direct causal, collider causal, and spurious correlation. The prediction relies solely on the first three causally relevant sub-segments, while the spurious correlation sub-segment is excluded. Furthermore, we propose Causal Informed Transformer (CAIFormer), a novel forecasting model comprising three components: Endogenous Sub-segment Prediction Block, Direct Causal Sub-segment Prediction Block, and Collider Causal Sub-segment Prediction Block, which process the endogenous, direct causal, and collider causal sub-segments, respectively. Their outputs are then combined to produce the final prediction. Extensive experiments on multiple benchmark datasets demonstrate the effectiveness of the CAIFormer.

## 1 Introduction

Multivariate Time Series Forecasting (MTSF) is a fundamental problem in various fields, including energy consumption (Bilal et al., 2022), economic planning (Hidalgo, 2009), weather prediction (Duchon & Hale, 2012), and traffic forecasting (Li et al., 2015). The goal of MTSF is to predict the future values of multiple interrelated variables based on their historical observations (Box et al., 2015). Unlike univariate time series forecasting, MTSF must capture not only individual temporal patterns but also the interactions among multiple interdependent variables. This makes it crucial to identify which variables influence the target and how. Failing to distinguish relevant from irrelevant inter-variable dependencies properly could either result in information loss or introduce spurious correlations that degrade forecasting performance.

Based on the above statement, an MTSF method should be capable of capturing the intrinsic temporal patterns of each variable, correctly identifying how other variables causally influence the target, and eliminating spurious correlations that obscure true dependence. However, most existing MTSF methods overlook this structural heterogeneity, including Transformer-based models (Wen et al., 2023; Tang & Zhang, 2023). These models typically take all variables' histories without differentiation, and train a single shared model to jointly forecast all targets in one forward pass (Liu et al., 2025; 2022). This design, whether channel-independent (Nie et al., 2023) or channel-mixed (Liu et al., 2024), makes no distinction in contribution among variables, ignoring the distinct causal roles they may play with respect to the target. Although this all-to-all design is easy to implement, it overlooks an important observation: when focusing on forecasting a specific target variable, the history segments of different variables often play very different roles. For instance, in weather forecasting, temperature and humidity are influenced by wind direction, yet they have no direct causal relationship. Meanwhile, temperature and atmospheric pressure jointly affect precipitation, forming a collider structure: temperature → precipitation ← pressure (Wilks, 2011). When forecasting temperature, different variables influence in distinct ways: (i) temperature's own past provides an autoregressive signal; (ii) wind direction exerts a direct causal influence by determining the

inflow of warm or cold air; (iii) through the collider structure, precipitation activates conditional dependence between temperature and pressure; and (iv) humidity appears correlated with temperature only through their shared cause wind direction, but becomes independent once wind direction is conditioned on. Feeding all of these histories indiscriminately into an all-to-all model conflates true causal drivers with spurious signals, leading to noisy attention weights, entangled parameter learning, and ultimately degraded forecasting performance.

To address the above challenges, we propose a novel all-to-one MTSF strategy that predicts the future trajectory of each target variable individually. This design enables capturing the heterogeneous influences of different historical segments on the target's future. This naturally raises the question: how should the historical window be decomposed for each target variable? As discussed in Section 3, motivated by the structural properties of causal graphs and the d-separation criterion in structural causal models (SCMs), we partition the complete historical window for each target variable into four sub-segments: 1) Endogenous Sub-segment (ES): the target variable's own history; 2) Direct Causal Sub-segment (DCS): histories of other variables that exhibit a direct influence on the target; 3) Collider Causal Sub-segment (CCS): histories that, along with the target, participate in collider patterns such as $V_i \rightarrow V_c \leftarrow V_s$; 4) Spurious Correlation Sub-segment (SCS): histories that become independent of the target once all other sub-segments are conditioned on.

Based on the above decomposition, we learn the conditional distribution $P(\text{target variable future} \mid \text{sub-segment history})$ for sub-segments 1)- 3), while discarding sub-segment 4) to avoid spurious correlations. We thus propose the Causal Informed Transformer (CAIFormer), comprising three blocks: Endogenous Sub-segment Prediction Block (ESPB), Direct Causal Sub-segment Prediction Block (DCSPB), and Collider Causal Sub-segment Prediction Block (CCSPB). ESPB applies an attention mechanism to capture both local and global temporal dependencies on ES. DCSPB applies a masked attention mechanism to attend exclusively to DCS, capturing the influence of direct causal variables. CCSPB first computes a preliminary prediction using masked attention over CCS and then projects it onto the kernel space (Section 3.5) to enforce the collider constraint and improve generalization. Finally, we combine the three blocks' outputs via an output projection layer that adaptively weights their contributions to produce the final forecast.

**Our contributions: 1**) We propose an all-to-one forecasting paradigm for MTSF, in which each target variable's future is predicted individually. For each target, we partition its complete history into four sub-segments: ES, DCS, CCS, and SCS; **2**) We propose CAIFormer, a novel forecasting method that separately captures the roles of different categories of variables via ESPB, DCSPB, and CCSPB, and combines their predictions to achieve accurate and interpretable MTSF; **3**) Extensive experiments and ablation studies on multiple benchmark datasets demonstrate that CAIFormer achieves superior predictive accuracy, robustness, and interpretability compared to existing methods.

## 2 RELATED WORKS

### 2.1 MULTIVARIATE TIME SERIES FORECASTING (MTSF)

With the development of deep learning (LeCun et al., 2015), numerous models have been proposed for MTSF (Hu & Xiao, 2022; Wen et al., 2023; Box et al., 2015; Lim & Zohren, 2021), including CNN-based (Zhan et al., 2023; Bai et al., 2018), RNN-based (Hewamalage et al., 2021; Tang et al., 2021), MLP-based (Zhang et al., 2024b), and Transformer-based (Zhang et al., 2024a) architectures. These approaches are all based on all-to-all strategy and can be broadly categorized based on their modeling focus into three groups: temporal-domain, frequency-domain, and variable-domain methods. Temporal domain methods, such as PatchTST (Nie et al., 2023) and TimesNet (Wu et al., 2023), focus on intra-variable dependencies by modeling patch-wise or point-wise relations. Frequency domain methods, such as FEDFormer (Zhou et al., 2022) and FreDF (Zhang et al., 2024a), transform sequences into the Fourier domain to capture frequency-specific dynamics. Variable domain methods, such as iTransformer (Liu et al., 2024), model inter-variable dependencies via attention mechanisms, and TimeXer (Wang et al., 2024) proposes to treat variables' endogenous and exogenous signals differently. A more detailed discussion of these approaches, particularly from the perspective of variable modeling and causal inference, is provided in Appendix **??**. This paper proposes a MTSF method that can explore the causal relationships between the future of the target variable and different sub-segments of the input history.

## 2.2 Modeling Variable Relationships in MTSF

The complex causal dependencies among variables in multivariate time series data pose significant challenges for modeling. Existing approaches exhibit primary modeling paradigms: Temporal-based methods (e.g., TimesNet (Wu et al., 2023) and PatchTST (Nie et al., 2023)) focus on intra-variable temporal patterns by analyzing relationships between time points or segments; Frequency-based methods (e.g., FreTS (Yi et al., 2023) and FEDformer (Zhou et al., 2022)) decompose temporal patterns through spectral transformations, but remain limited to single-variable analysis; Variable-based methods (e.g., iTransformer (Liu et al., 2024)) attempt to capture cross-variable interactions through attention mechanisms. Similarly, TimePro (Ma et al., 2025) introduces a multi-lag approach, incorporating both time- and variable-aware hyper-state embeddings to capture complex, dynamic inter-variable dependencies. However, these methods often perform unconstrained pairwise computations which may conflate causal relationships with spurious correlations, lacking explicit mechanisms to distinguish different types of inter-variable dependencies. The limitations of these approaches reveal a gap: current methods either oversimplify cross-variable correlation or naively aggregate all potential interactions without causal discrimination. Therefore, effectively modeling inter-variable relationships in MTSF remains an open research problem.

## 2.3 Causal Discovery for MTSF

Causal discovery provides a systematic framework for identifying genuine cause-effect relationships from observational data (Glymour et al., 2016; Pearl, 2009b; Gong et al., 2023) through directed acyclic graphs (DAG) (Lauritzen & Wermuth, 1989). The development of causal discovery methods has evolved through several stages: Constraint-based approaches (e.g., Inductive Causation algorithm (Verma & Pearl, 1990) and the Peter-Clark algorithm (Spirtes & Glymour, 1991)) rely on conditional independence tests to reconstruct DAG. However, they suffer from high computational costs and struggle with hidden confounders. Later methods improved computational (Spirtes et al., 2001) efficiency and enhanced confounder modeling (Spirtes, 2001). In time series settings, causal discovery faces additional challenges. For instance, tsFCI (Entner & Hoyer, 2010a) extends FCI to accommodate time-lagged dependencies, while Granger causality (Granger, 1969) infers temporal precedence based on predictive accuracy tests. Recent studies have attempted to integrate causal discovery with MTSF, primarily through two paradigms: 1) Causal Markov Models: These methods use causal inference as a preprocessing step to remove spurious correlations (Li et al., 2021), and 2) Proxy Variable Methods: These approaches leverage latent variable recovery techniques to infer hidden causal structures (Liu et al., 2023). CausalFormer (Kong et al., 2025) further introduces a Transformer variant that jointly learns temporal dependencies and a Granger-causal graph to guide multivariate forecasting. Causal-TSF (Gong et al., 2025) instead tackles hidden confounding bias by estimating latent confounders and applying causal interventions during forecasting. However, these methods typically treat causal discovery as either a preprocessing step, a regularizer, or a confounder correction mechanism, rather than integrating it directly into the model's parametrization. As a result, they fail to dynamically incorporate causal information during the forecasting process. Unlike existing methods, our approach incorporates DAG as architectural constraints, enforcing causal dependencies during model training. This ensures that the forecasting mechanism aligns with underlying causal structures, eliminating spurious correlations while enhancing model interpretability.

# 3 Causal Analysis and Motivation

In this section, we first present some notations. Then, we explain from a causal analysis perspective why it is essential to separate the influence of different historical segments on the target variable. At last, we convert these causal analyses into concrete modeling guidelines for our MTSF framework.

## 3.1 Notation and Problem Definition

Let $X = [x_1, \cdots, x_T] \in \mathbb{R}^{T \times D}$ be a historical sequence with $T$ time steps and $D$ variables. At each timestamp $t \in \{1, \cdots, T\}$, the state of $X$ is represented as $x_t = [V_1^t, \cdots, V_D^t] \in \mathbb{R}^D$, where $V_i^t \in \mathbb{R}$ is the observed value of the variable $V_i$ at time step $t$. Let $Y = [x_{T+1}, \cdots, x_{T+S}] \in \mathbb{R}^{S \times D}$ be the future sequence with $S$ time steps. Given a training dataset $D_{\text{train}} = \{(X^i, Y^i)\}_{i=1}^K$, where $K$ is the

number of training samples, $X^i$ represents the $i$-th historical sequence, and $Y^i$ is its corresponding future sequence. The learning process of MTSF can be formalized as finding an optimal predictor $f^*$ within a hypothesis space $\mathcal{F}$, such that $f^*(X) = Y$. Specifically, the forecasting model is learned by solving the following empirical risk minimization problem:

$$f^* = \arg\min_{f \in \mathcal{F}} \frac{1}{K} \sum_{i=1}^{K} \mathcal{L}(Y^i, f(X^i)), \tag{1}$$

where $\mathcal{L}(\cdot)$ denotes the loss function, e.g., the MSE loss. As shown in Equation (1), the learning process of $f^*$ doesn't constrain correlations among variables in the input.

In Section 1, we identified a potential limitation in existing MTSF methods: when analyzing model predictions from the perspective of a specific target variable, all-to-all based strategies may inadvertently encode spurious correlations between variables. This can lead to inaccurate forecasting and significantly degrade the model's generalization ability. This issue is further substantiated by the empirical analysis presented in Section 3.3. To address this challenge, the key lies in understanding how each variable's historical values contribute to target variable future evolution. We propose a segmentation strategy based on semantic consistency: the historical sequences of variables that influence the target variable through similar causal mechanisms are grouped into the same sub-segment, while segments with distinct mechanisms are separated. In following section, we elaborate on the rationale and theoretical foundations behind this segmentation approach.

### 3.2 WHY DIVIDE HISTORY INTO ENDOGENOUS AND EXOGENOUS COMPONENTS?

For any target variable in MTSF, its own history can be regarded as a discrete sampling of an underlying dynamical system (Takens, 1981), encoding all the information needed to describe the variable's intrinsic evolution (Wang et al., 2024). What we really want to uncover is how the history of other variables influences that evolution. If we do not explicitly separate the endogenous sub-segment from the exogenous sub-segment, any apparent improvement in prediction will be confounded by the history of the target variable itself. We will observe that the target changes, but cannot determine whether that change is driven by external histories or merely by its own history. Clear causal attribution therefore, demands an explicit distinction between endogenous and exogenous historical sub-segments.

### 3.3 WHY FURTHER SUBDIVIDE THE EXOGENOUS SEGMENT?

Having explicitly distinguished the endogenous and exogenous sub-segments, we next examine the causal structure of the exogenous sub-segment. According to (Pearl, 2009b), causal relationships between variables can be classified as either direct or indirect. From the SCM perspective, a direct causal relationship indicates an immediate connection between two variables, whereas an indirect causal relationship involves one or more intermediate nodes. As noted in Appendix C, two variables that are indirectly connected may become independent once we condition on intermediate variables. If we indiscriminately use all variables' histories to predict the target, variables with no causal relevance can nonetheless appear informative in the learned predictor, creating apparent associations with the target that do not reflect genuine causal influence, that is, spurious correlation. Therefore, the exogenous sub-segment should be further divided into causally relevant sub-segment and spurious correlation sub-segment.

To empirically validate the necessity of further subdividing the exogenous sub-segment, we conducted experiments on ETTh1, ETTm1, and Exchange-rate datasets. We employed Granger causality analysis, a common statistical test in time series analysis, to examine inter-variable predictive relationships. Specifically, if past values of one variable improve the forecasting accuracy of target variable, indicating a Granger-causal relationship, we classify its historical observations into the causally relevant sub-segment. Otherwise, it is assigned to the spurious correlation sub-segment because modeling its history introduces spurious correlations without improving predictive accuracy. Granger causality analysis outputs a $P$-value indicating statistical significance. To better visualize these influences, we apply a $-\log(P)$ transformation and present the results as a heatmap in Figure 1a-1c. In the heatmap, the cell at row $m$ and column $n$ represents causal influence from the $m$-th variable to the $n$-th variable, with darker colors corresponding to stronger influences. Diagonal elements are filled with a uniform color to exclude self-influence. As shown in Figure 1 left, some

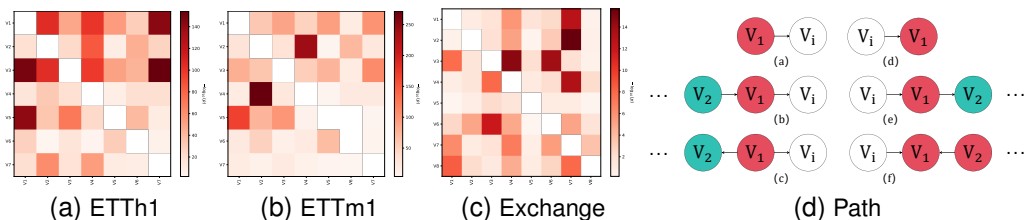

(a) ETTh1    (b) ETTm1    (c) Exchange    (d) Path

Figure 1: (a)-(c) Visualization of Granger causality across variables in ETTh1, ETTm1, and Exchange datasets. Each heatmap shows the transformed causal strength matrix using $-\log(P)$ values, where a darker color indicates a stronger causal influence from the row variable to the column variable. Diagonal entries are masked. (d) Representative partial SCM commonly encountered in MTSF. White nodes represent the target variable $V_i$, red and green nodes represent causally related variables and spurious correlated variables separately.

variables exhibit strong causal impacts on others, while many pairs display negligible or no causal effect. This observation underscores the necessity of clearly identifying genuine causal relationships to avoid spurious correlations, thereby improving predictive performance.

### 3.4 How should the exogenous segment be further subdivided?

To precisely distinguish genuine causal relationships from spurious ones, we systematically analyze all relevant causal pathways connected to the target variable $V_i$ and construct a local SCM centered around it. Based on the causal analysis in Appendix C, the relationship between a non-target variable and the target $V_i$ in Fig.1d can be treat separately:

First, $V_1$ can be an ancestor of $V_i$, including the direct-parent case and more general upstream structures such as chains and forks (e.g., $V_2 \rightarrow V_1 \rightarrow V_i$ or $V_2 \leftarrow V_1 \rightarrow V_i$, corresponding to Paths (a)–(c)). In all these cases, $V_1$ lies on a directed path into $V_i$, and conditioning on all remaining variables $\mathcal{Z} = \{V_1, \ldots, V_{i-1}, V_{i+1}, \ldots, V_D\}$ leaves $V_1$ dependent on $V_i$, while any other $V_j \in \mathcal{Z} \setminus \{V_1\}$ becomes conditionally independent of $V_i$.

Second, $V_1$ can be a descendant of $V_i$, including direct children and any downstream chain (corresponding to Paths (d)–(e)). This situation is symmetric: $V_1$ lies on a directed path out of $V_i$, and after conditioning on $\mathcal{Z}$, only $V_1$ remains conditionally dependent on $V_i$, whereas all other $V_j \in \mathcal{Z} \setminus \{V_1\}$ are independent of $V_i$.

Finally, $V_1$ can act as a collider on a path from $V_i$ to another variable $V_2$ (Path (f)), e.g., $V_i \rightarrow V_1 \leftarrow V_2 - \cdots - V_D$. In this case, conditioning on $\mathcal{Z}$ makes both $V_1$ and its spouse $V_2$ dependent on $V_i$, while any $V_j \in \mathcal{Z} \setminus \{V_1, V_2\}$ is conditionally independent of $V_i$. Formal proofs of the conditional independencies asserted for each path are provided in Appendix D.

Without loss of generality, consider any SCM defined over a set of variables $\{V_1, \cdots, V_D\}$. For any target variable $V_i$ and the local SCM relevant to $V_i$ can be represented by the combination of elements in $\{\textbf{Path a}, \cdots, \textbf{Path f}\}$. Then, we can identify a subset of variables that are conditionally dependent on the $V_i$ and eliminate other independent variables. Specifically, the simplified SCM includes: 1) **Direct Parents**: Variables that have a direct causal influence on $V_i$, denoted as $V_p \rightarrow V_i$; 2) **Direct Children**: Variables that are directly influenced by $V_i$, denoted as $V_i \rightarrow V_k$ and each $V_k$ is not a collider; 3) **Collider Structures**: Variables that form collider structure involving $V_i$, e.g., $V_i \rightarrow V_c \leftarrow V_s$, where $V_c$ is the collider and $V_s$ denotes the spouse variables.

Ultimately, for each variable, we partition the exogenous segment into three sub-segments: 1) Direct Causal Sub-segment (DCS): including all variables that are direct parents or direct children of the target variable, representing direct causal affect on target variable; 2) Collider Causal Sub-segment (CCS): consisting of variables that, together with the target, form collider patterns such as $V_i \rightarrow V_c \leftarrow V_s$, where $V_c$ is the collider node; 3) Spurious Correlation Sub-segment (SCS): comprising every remaining variable that is not part of the direct parent, direct child, or collider structures; these variables do not reflect genuine causality with the target variable. In the next section,

we further elaborate on why causal relationships should be distinguished specifically between DCS and CCS to enhance predictive generalization.

### 3.5 WHY ARE CAUSAL RELATIONSHIPS DIVIDED INTO DCS AND CCS?

While both DCS and CCS are causally relevant to the target, we treat CCS as a separate component because its collider-induced dependencies behave differently in prediction: they do not directly affect the target, but can affect generalization if not properly constrained. As we show below, isolating CCS enables us to enforce a conditional independence constraint that reduces the generalization gap.

Consider any variable pair $(V_{c,j}, V_{s,j})$ that constitutes a collider structure $V_i \to V_{c,j} \leftarrow V_{s,j}$ within the set $\{V_{c,j}, V_{s,j}\}_{j=1}^m$, where $m$ denotes the number of colliders. Both $V_{c,j}$ and $V_{s,j}$ consist of only one variable. Then, the optimal predictor $f_{\text{IP}}^*$ under MSE loss for the future values of $V_i$ is defined as:

$$V_i^{T:T+S} = f_{\text{IP}}^*(V_{c,j}^{0:T}, V_{s,j}^{0:T}) = \mathbb{E}[V_i^{T:T+S} | V_{c,j}^{0:T}, V_{s,j}^{0:T}]. \tag{2}$$

For notational simplicity, we henceforth denote the history $V_{c,j}^{0:T}$ and $V_{s,j}^{0:T}$ simply as $V_{c,j}$ and $V_{s,j}$ respectively, and $V_i^{T:T+S}$ simply as $V_i$. Thus, we have: $f_{\text{IP}}^*(V_{c,j}^{0:T}, V_{s,j}^{0:T}) \cong f_{\text{IP}}^*(V_{c,j}, V_{s,j}) = \mathbb{E}[V_i | V_{c,j}, V_{s,j}]$. Collider structure implies the independence relationship $V_i \perp\!\!\!\perp V_{s,j}$, we can obtain:

$$\mathbb{E}[f_{\text{IP}}^*(V_{c,j}, V_{s,j}) | V_{s,j}] = \mathbb{E}[\mathbb{E}[V_i | V_{c,j}, V_{s,j}] | V_{s,j}] = \mathbb{E}[V_i | V_{s,j}] = \mathbb{E}[V_i], \tag{3}$$

where the second equality follows from the tower property (Pearl & Paz, 2022). Let $\mathcal{S}_{V_c} = \{V_{c,1}, \cdots, V_{c,m}\}$ and $\mathcal{S}_{V_s} = \{V_{s,1}, \cdots, V_{s,m}\}$. Since each path $V_i \to V_{c,j} \leftarrow V_{s,j}$, $j = 1, \cdots, m$, forms a separate collider structure and these structures do not intersect, the corresponding independence relations hold. $V_i \perp\!\!\!\perp \mathcal{S}_{V_s}$ means every $V_{s,j} \in \mathcal{S}_{V_s}$ is $V_i \perp\!\!\!\perp V_{s,j}$, and $V_i \not\perp\!\!\!\perp \mathcal{S}_{V_s} \mid \mathcal{S}_{V_c}$ means condition on $\mathcal{S}_{V_c}$, $\forall V_{s,j} \in \mathcal{S}_{V_s}$ exists $V_i \not\perp\!\!\!\perp V_{s,j} \mid \mathcal{S}_{V_c}$. For all variable pairs in $\{V_{c,j}, V_{s,j}\}_{j=1}^m$, Equation (3) equals to:

$$\begin{aligned}\mathbb{E}[f_{\text{IP}}^*(\mathcal{S}_{V_c}, \mathcal{S}_{V_s}) | \mathcal{S}_{V_s}] &= \mathbb{E}[f_{\text{IP}}^*(V_{c,1}, V_{s,1}, \cdots, V_{c,m}, V_{s,m}) | V_{s,1}, \cdots, V_{s,m}] \\ &= \mathbb{E}[\mathbb{E}[V_i | V_{c,1}, V_{s,1}, \cdots, V_{c,m}, V_{s,m}] | V_{s,1}, \cdots, V_{s,m}] = \mathbb{E}[V_i | V_{s,1}, \cdots, V_{s,m}] = \mathbb{E}[V_i],\end{aligned} \tag{4}$$

Without loss of generality, we assume that $\mathbb{E}[V_i] = C$, here $C$ is a constant. This implies that:

$$f_{\text{IP}}^* \in \mathcal{F}_\Psi = \left\{ f \in \mathcal{F} \mid \mathbb{E}[f(\mathcal{S}_{V_c}, \mathcal{S}_{V_s}) \mid \mathcal{S}_{V_s}] - C = 0 \right\}, \tag{5}$$

where $\mathcal{F}$ is denoted as $L^2(V)$, a space of the square-integrable functions. Let $\Phi : L^2(V) \to L^2(V)$ denote the following conditional expectation operator:

$$\Phi f(\mathcal{S}_{V_c}, \mathcal{S}_{V_s}) = \mathbb{E}[f(\mathcal{S}_{V_c}, \mathcal{S}_{V_s}) \mid \mathcal{S}_{V_s}] - C. \tag{6}$$

Based on Equation (5) and Equation (6), the space $L^2(V)$ can be decomposed orthogonally as $L^2(V) = \text{Preimage}(\Phi) \oplus \text{Kernel}(\Phi)$, where $\text{Kernel}(\Phi) = \mathcal{F}_\Psi$ denotes the kernel (null space) of $\Phi$, while $\text{Preimage}(\Phi)$ denotes the preimage space (inverse image) of $\Phi$. Based on Equation (2) and (5), we obtain that $f_{\text{IP}}^*$ lies in $\text{Kernel}(\Phi)$. Then, for any $f \in L^2(V)$, define the projection $\Psi$ as:

$$\Psi f = f - \Phi f, \tag{7}$$

where $\Psi = I - \Phi$ and $I$ is an identity mapping. Then, $\Psi$ orthogonal projects $f$ into $\text{Kernel}(\Phi)$. Thus, we want $\Psi f$ as our ideal prediction function. Then, we can obtain the following theorem, which demonstrates that $\Psi f$ can improve generalization of $f$:

**Theorem 3.1** (*Generalization Gap Reduction*) *For any predictor $f \in L^2(V)$, we can obtain $\Delta(f, \Psi f) = \|\Phi f\|_{L^2(V)}^2 \geq 0$, where $\Delta(f, \Psi f)$ denotes the generalization gap, which is defined by $\Delta(f, \Psi f) = \mathbb{E}[(V_i - f(\mathcal{S}_{V_c}, \mathcal{S}_{V_s}))^2] - \mathbb{E}[(V_i - \Psi f(\mathcal{S}_{V_c}, \mathcal{S}_{V_s}))^2]$.*

See the proof in Appendix E. From a causal perspective, $\Phi f$ captures the portion of $f$ that is spuriously correlated with $\mathcal{S}_{V_s}$. In other words, the operator $\Phi$ extracts those components of $f$ that vary systematically with $\mathcal{S}_{V_s}$ but offer no real predictive benefit for $V_i$. Under collider-induced independence ($V_i \perp\!\!\!\perp \mathcal{S}_{V_s}$), any apparent correlation with $\mathcal{S}_{V_s}$ reflects noise or sampling artifacts.

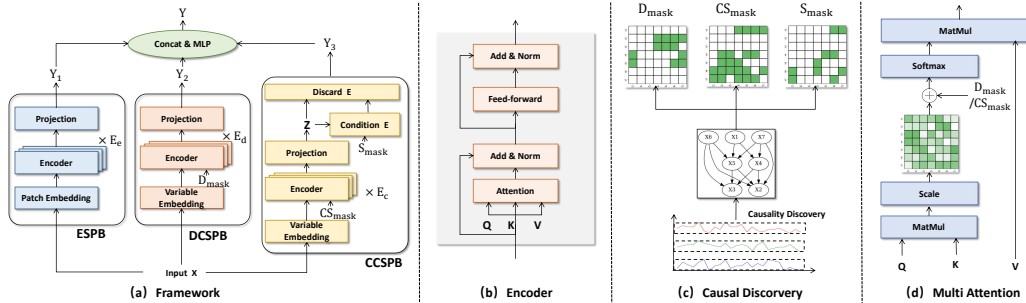

Figure 2: Visualization of the CAIFormer. (a) depicts the overall architecture, consisting of ESPB, DCSPB, and CCSPB, three Blocks. (b) illustrates the Encoder structure, featuring Multi-Patch Attention in ESPB and Multi-variate Attention in both DCSPB and CCSPB. (c) shows the causal discovery on a dataset. (d) demonstrates how we impose constraints on Multi-variate Attention. In CCSPB, Condition E refers to the collider constraint.

Consequently, incorporating this dependence not only fails to improve prediction but can also inflate variance by fitting irrelevant fluctuations. By contrast, the projection $\Psi f$ removes these spurious elements-effectively filtering out parts of $f$ that do not help predict $V_i$. Eliminating such irrelevant dependencies tightens the generalization bound (Mohri et al., 2018) because reducing the hypothesis space naturally curbs overfitting. This observation clarifies the reasoning behind **Theorem 3.1**, and shows how constraining $f$ to the kernel space of $\Phi$ directly mitigates the generalization gap. The above theoretical insights naturally motivate the decomposition strategy employed in our final predictive model, as described in the next sections.

**Motivation**. Drawing on the analyses in Sections 3.2-3.5, we partition the historical influence of all variables on the target variable into three components: the ES, DCS, and CCS. This decomposition motivates an MTSF architecture that assigns a dedicated modeling module to each component and then combines their outputs to yield the final prediction.

## 4 THE PROPOSED METHOD

We formalize MTSF under an SCM perspective. For each variable $V_i$, its next-step value is determined by ES, DCS, and CCS, together with an exogenous noise term $\epsilon_i$: $V_i^{t+1} = g_i(\mathrm{ES}_i^t, \mathrm{DCS}_i^t, \mathrm{CCS}_i^t, \epsilon_i^t), \epsilon_i^t \perp\!\!\!\perp \epsilon_j^t (i \neq j)$. Most existing Transformer-based methods for MTSF follow an all-to-all forecasting principle, which we found insufficient given the causal decomposition motivated in Section 3. Thus, we adopt a decomposed MTSF strategy that first selects each variable in $Y$ and then forecasts it individually to better isolate causal influences. This finer-grained approach enables examination of how each historical segment affects each target variable's future. Guided by this idea, we refine the forecasting process in three ways: 1) we explicitly distinguish the target's own history from other variables' histories (Section 3.2); 2) we identify and remove spurious associations while preserving genuine causal effects (Section 3.3 and 3.4); and 3) we further partition the endogenous segment into DCS and CCS based on their distinct roles in prediction (Section 3.5). Therefore, we propose Causal Informed Transformer (CAIFormer), a novel MTSF architecture that implements this decomposition. In this section, we detail the framework of CAIFormer.

**Causal Discovery.** As the first step, we apply the Peter-Clark (PC) algorithm (Spirtes et al., 2001; Cai et al., 2023) on observational data (Figure 2(c)) to construct a Directed Acyclic Graph (DAG). Although PC is originally designed for static data, in our setting each lookback window is treated as a fixed-dimensional vector of variables, making PC applicable. Our framework is not limited to PC; alternative methods for time series, such as PCMCI (Runge et al., 2019) or tsFCI (Entner & Hoyer, 2010b), could also be adopted in future work. We conducted a series of experiments in Appendices H and I to verify the sufficiency of the DAGs obtained by PC.

In our method, we assume causal sufficiency, meaning there are no latent confounders beyond the observed variables. The algorithm generates an adjacency matrix $W_{\mathrm{adjm}} \in \{-1, 0, 1\}^{D \times D}$, where each element $W_{\mathrm{adjm}}[i][j] = -1$ indicates a directed edge $V_i \to V_j$, and $W_{\mathrm{adjm}}[i][j] = W_{\mathrm{adjm}}[j][i] = -1$ represents an undirected edge. For each target variable $V_i$, we define its causal relationships by

extracting sets from the adjacency matrix. Specifically, we identify Direct Parents $\mathcal{S}_i^P$, Direct Children $\mathcal{S}_i^K$, and Colliders $\mathcal{S}_i^C$, based on the structure of the matrix. The corresponding masks $D_{\text{mask}}$, $CS_{\text{mask}}$, and $S_{\text{mask}}$ are then created to capture the causal influences, where the mask values are 1 if the relationship is present and 0 otherwise. Detailed set definitions and their corresponding formulas can be found in Appendix R.

**Endogenous Sub-segment Prediction Block (ESPB).** As discussed in Section 3.2, each target's endogenous segment encodes its intrinsic evolution. We design ESPB to capture this intrinsic temporal dynamics. Let the input history be $X = \{V_1^{0:T}, \cdots, V_D^{0:T}\} \in \mathbb{R}^{T \times D}$, and the future sequence be $Y = \{V_1^{T:T+S}, \cdots, V_D^{T:T+S}\}$ with prediction horizon $S$. For each target $V_i$, ESPB predicts $V_i^{T:T+S} = f_e(V_i^{0:T})$. We adopt the Patching module (Nie et al., 2023) to segment $X$ into overlapping patches: $X_{\text{Patch}} = f_{\text{Patch}}(X)$, where $f_{\text{Patch}} : \mathbb{R}^{T \times D} \to \mathbb{R}^{H \times P \times D}$ is a variable-wise independent process, with $H$ patches of length $P$. These patches are embedded as $X_{\text{Enc}}^0 = f_{\text{Emb}}^t(X_{\text{Patch}}) \in \mathbb{R}^{D \times H \times d_E}$, and then passed through $E_e$ Encoder layers: $X_{\text{Enc}}^e = \text{Encoder}(X_{\text{Enc}}^{e-1})$, $e = 1, \cdots, E_e$. Here, for each variable, the $H$ patches along the temporal axis are treated as the token sequence fed into the Transformer encoder, while the patch length $P$ is absorbed into the embedding dimension $d_E$. In other words, ESPB performs patch-wise self-attention over time for each variable independently, with the variable index acting as a batch dimension rather than a sequence dimension. The final embeddings are projected to the forecast space: $Y_e = f_{\text{Projection}}^e(X_{\text{Enc}}^{E_e})$.

**Direct Causal Sub-segment Prediction Block (DCSPB).** As discussed in Section 3.4, direct parents exert direct influence on the target, while direct children are directly influenced by it. We model capture causal impact by $V_i^{T:T+S} = f_d(V_p^{0:T}, V_k^{0:T}) = \sum_{\alpha \in V_p^{0:T}} g_d(\alpha)/\delta + \sum_{\beta \in V_k^{0:T}} g_d(\beta)/\delta$, where $g_d$ transforms a direct causal sub-segment into a predictive representation, and $\delta$ is a normalization factor. Specifically, DCSPB employs a Transformer whose attention is masked by $D_{\text{mask}}$.

**Transformer with Variable Attention Mask.** We first apply a variable-wise embedding: $X_{\text{Enc}}^0 = f_{\text{Emb}}^v(X)$, where $X_{\text{Enc}}^0 \in \mathbb{R}^{D \times d_D}$ and $d_D$ is the embedding dimension. In this embedding step, the temporal dimension is compressed: for each variable $V_j$, $f_{\text{Emb}}^v$ aggregates its entire history $V_j^{0:T}$ into a single $d_D$-dimensional vector. As a result, $X_{\text{Enc}}^0$ can be viewed as a length-$D$ sequence where each token corresponds to one variable's historical trajectory. The Transformer encoder in DCSPB therefore operates along the variable axis, and the attention masks $D_{\text{mask}}$ restrict which variable pairs are allowed to attend to each other according to the learned DAG. Then, for $e = 1, \cdots, E_d$, we compute $X_{\text{Enc}}^e = \text{Encoder}(X_{\text{Enc}}^{e-1}, D_{\text{mask}})$. We apply $D_{\text{mask}}$ in attention: $\text{Attention}(Q, K, V) = \text{softmax}(QK^T \odot D_{\text{mask}}/\sqrt{d_k})V$, where $\odot$ is element-wise multiplication and $d_k$ is the dimensionality factor used for scaling. Finally, we project to the forecast space: $Y_d = f_{\text{Projection}}^d(X_{\text{Enc}}^{E_d})$, where $Y_d \in \mathbb{R}^{S \times D}$ yielding the DCSPB output.

**Collider Causal Sub-segment Prediction Block (CCSPB).** As discussed in Section 3.4, CCSPB predicts $V_i^{T:T+S} = f_c(V_c^{0:T}, V_s^{0:T})$ for collider structure. Specifically, $f_c$ follows DCSPB similar pipeline: embedding, $E_c$ mask-attention encoder layers, and projection, apply to $V_c^{0:T}$ and $V_s^{0:T}$. The attention in each encoder layer is masked by $CS_{\text{mask}}$, thus, the model attends only to the collider structure. Producing preliminary predictions $Z$, which are then refined under the collider constraint.

**Collider Constraint.** For the preliminary prediction $Z$, we enforce the constraint in Equation (7) by projecting into $\text{Range}(\Phi)$. Specifically, we extract spouse sub-segment via $X_{\text{Collider}} = X \odot S_{\text{mask}}$, where $\odot$ denotes element-wise multiplication. We then compute the conditional expectation $EZ = \mathbb{E}[Z \mid X_{\text{Collider}}] - C$ according to Equation (6). Finally, we project $Z$ into the Kernel($\Phi$) by subtracting $EZ$, yielding $Y_c = Z - EZ$, thereby enforcing the constraint in Equation (5). Here, $Y_c \in \mathbb{R}^{S \times D}$ is the final output of the CCSPB.

**Output Projection Layer.** We concatenate the three outputs along the variable dimension: $Y_{\text{cat}} = \text{Concat}(Y_e, Y_d, Y_c) \in \mathbb{R}^{S \times (3D)}$. We then fuse via an MLP: $Y = f_o(Y_{\text{cat}}) = Y_{\text{cat}} W_o + b_o$, where $W_o \in \mathbb{R}^{3D \times D}$ and $b_o \in \mathbb{R}^{S \times D}$ are learnable parameters that adaptively fuse the three sub-segment outputs into the final prediction $Y = (V_1^{T:T+S}, \cdots, V_D^{T:T+S}) \in \mathbb{R}^{S \times D}$.

Table 1: MTSF results with prediction lengths $S \in \{96, 192, 336, 720\}$ and fixed lookback length $T = 96$. The best results in **bold**, the second underlined, and "–" denote metrics not reported in the original papers. The lower MSE/MAE indicates a more accurate prediction result. The full results in Table 18.

| Models | CAIFormer (Ours) | | TimePro (2025) | | SEMPO (2025) | | TFPS (2025) | | iTransformer (2024) | | PatchTST (2023) | | RLinear (2023) | | Crossformer (2023) | | TiDE (2023) | | TimesNet (2023) | | DLinear (2023) | | FEDformer (2022) | | Autoformer (2021) | |
|---|---|---|---|---|---|---|---|---|---|---|---|---|---|---|---|---|---|---|---|---|---|---|---|---|---|---|
| Metric | MSE | MAE | MSE | MAE | MSE | MAE | MSE | MAE | MSE | MAE | MSE | MAE | MSE | MAE | MSE | MAE | MSE | MAE | MSE | MAE | MSE | MAE | MSE | MAE | MSE | MAE |
| ETTm1 | **0.382** | **0.395** | 0.391 | 0.400 | 0.503 | 0.466 | 0.395 | 0.406 | 0.407 | 0.410 | 0.387 | 0.400 | 0.414 | 0.407 | 0.513 | 0.496 | 0.419 | 0.419 | 0.400 | 0.406 | 0.403 | 0.407 | 0.448 | 0.452 | 0.588 | 0.517 |
| ETTm2 | 0.276 | 0.323 | 0.281 | 0.326 | 0.286 | 0.341 | 0.276 | 0.321 | 0.288 | 0.332 | 0.281 | 0.326 | 0.286 | 0.327 | 0.757 | 0.610 | 0.358 | 0.404 | 0.291 | 0.333 | 0.350 | 0.401 | 0.305 | 0.349 | 0.327 | 0.371 |
| ETTh1 | 0.439 | 0.439 | 0.438 | 0.438 | 0.410 | 0.430 | 0.448 | 0.443 | 0.454 | 0.447 | 0.469 | 0.454 | 0.446 | 0.434 | 0.529 | 0.522 | 0.541 | 0.507 | 0.458 | 0.450 | 0.456 | 0.452 | 0.440 | 0.460 | 0.496 | 0.487 |
| ETTh2 | 0.380 | 0.403 | 0.377 | 0.403 | 0.341 | 0.391 | 0.380 | 0.403 | 0.383 | 0.407 | 0.387 | 0.407 | 0.374 | 0.398 | 0.942 | 0.684 | 0.611 | 0.550 | 0.414 | 0.427 | 0.559 | 0.515 | 0.437 | 0.449 | 0.450 | 0.459 |
| Exchange | 0.345 | 0.395 | 0.352 | 0.399 | - | - | 0.395 | 0.414 | 0.360 | 0.403 | 0.367 | 0.404 | 0.378 | 0.417 | 0.940 | 0.707 | 0.370 | 0.413 | 0.416 | 0.443 | 0.354 | 0.414 | 0.519 | 0.429 | 0.613 | 0.539 |
| Weather | 0.239 | 0.268 | 0.251 | 0.276 | 0.248 | 0.287 | 0.241 | 0.271 | 0.258 | 0.279 | 0.259 | 0.272 | 0.291 | 0.259 | 0.315 | 0.271 | 0.320 | 0.259 | 0.287 | 0.265 | 0.317 | 0.309 | 0.360 | 0.338 | 0.338 | 0.382 |
| ECL | **0.168** | **0.261** | 0.169 | 0.262 | 0.196 | 0.295 | 0.183 | 0.280 | 0.178 | 0.270 | 0.205 | 0.290 | 0.219 | 0.298 | 0.244 | 0.334 | 0.251 | 0.344 | 0.192 | 0.295 | 0.212 | 0.300 | 0.214 | 0.327 | 0.227 | 0.338 |
| Traffic | **0.421** | **0.275** | - | - | 0.466 | 0.344 | - | - | 0.428 | 0.282 | 0.481 | 0.304 | 0.626 | 0.378 | 0.550 | 0.304 | 0.760 | 0.473 | 0.620 | 0.336 | 0.625 | 0.383 | 0.610 | 0.376 | 0.628 | 0.379 |

In summary, our decoupled forecasting approach offers several advantages. First, by explicitly excluding irrelevant variables, the model eliminates the risk of learning misleading patterns. Second, by separating intrinsic temporal dynamics from direct causal influences, the model achieves more accurate interaction modeling. Third, systematically incorporating collider structure variables leverages conditional dependencies, enhancing the forecasting context.

## 5 EXPERIMENTAL RESULTS

In this section, we present results on eight benchmark datasets, followed by ablation studies on each module and the projection $\Psi$, and an evaluation of CAIFormer's efficiency. Additional analyses, including interpretability, robustness, lookback length, and comparisons with causal and channel-independent TSF methods, are provided in Appendix H-L.

**Experimental Setup.** We evaluate CAIFormer on eight real-world multivariate time series datasets. Experiments are implemented in PyTorch and trained on NVIDIA V100 GPUs. For fair comparison, we follow the standard train-validation-test splits and preprocessing protocols of (Liu et al., 2024). Hyperparameters (learning rate, batch size, depth, hidden size) are tuned via grid search on validation performance. We report MSE and MAE, averaged over five runs with different random seeds. Details on datasets, PC implementation, and model configurations are given in Appendix F, N, and G, and we also analyze the nonlinear PC in Appendix O.

**Comparison Results.** We evaluate CAIFormer on eight benchmark datasets following the settings of iTransformer (Liu et al., 2024), with lookback length fixed at 96 and prediction lengths in $\{96, 192, 336, 720\}$. We compare against nine representative baselines, including Transformer-based (iTransformer, PatchTST, Autoformer, FEDformer, Crossformer, TimePro, SEMPO, TFPS), linear (DLinear, TiDE, RLinear), and TCN-based (TimesNet) models. As shown in Table 1, CAIFormer achieves the best or second-best performance across all datasets, consistently outperforming strong baselines such as iTransformer and PatchTST, demonstrating its superior forecasting accuracy. Additionally, Appendix K provides a comparison with causal inference-based forecasting methods (Gong et al., 2025), and Appendix L provides a comparison with channel-independent forecasting methods (Wang et al., 2024).

**Ablation Study About Each Block.** We assess the contribution of each module (ESPB, DCSPB, CCSPB) on the Weather, ETTh1, and Exchange datasets. All experiments use a fixed lookback window of $T = 96$ and prediction horizons $S \in 96, 192, 336, 720$, with averaged MSE/MAE results reported in Table 2. The full model with all three modules consistently achieves the best performance. Removing any one block leads to a clear increase in error, indicating that each captures unique and complementary aspects of the temporal–causal structure. To further validate the role of causal masks, we introduce a Shuffle Mask setting, where $D_{\text{mask}}$ and $CS_{\text{mask}}$ are randomly permuted on 10% of variables. In this case, even when only DCSPB or CCSPB is retained, performance drops substantially compared to the unshuffled version. This demonstrates that the masks encode non-trivial causal constraints from the DAG, and disrupting them introduces spurious connections that degrade prediction accuracy. Appendix M presents an explicit comparison between hard discarding of SCS variables and a soft alternative with an additional SCS block, showing that the latter brings no performance benefit.

Table 2: The average performance of lookback length $T = 96$ and prediction lengths $S \in \{96, 192, 336, 720\}$ in weather and ETTh1 datasets.

| ESPB | DCSPB | CCSPB | Shuffle Mask | Weather | | ETTh1 | | Exchange | |
|---|---|---|---|---|---|---|---|---|---|
| | | | | MSE | MAE | MSE | MAE | MSE | MAE |
| w | w | w | No | 0.239 | 0.268 | 0.439 | 0.439 | 0.345 | 0.395 |
| w/o | w/o | w | No | 0.354 | 0.345 | 0.533 | 0.527 | 0.448 | 0.497 |
| w | w/o | w/o | No | 0.259 | 0.281 | 0.469 | 0.454 | 0.367 | 0.404 |
| w/o | w | w/o | No | 0.282 | 0.326 | 0.491 | 0.483 | 0.415 | 0.463 |
| w/o | w/o | w | Yes | 0.378 | 0.369 | 0.551 | 0.548 | 0.471 | 0.522 |
| w/o | w | w/o | Yes | 0.331 | 0.370 | 0.512 | 0.509 | 0.439 | 0.487 |

**Ablation Study on the Projection Operator $\Psi$.** To empirically validate the generalization benefits of the projection operator $\Psi$ introduced in Section 3.5, we conduct an ablation experiment on the Weather dataset. We fix both the lookback window and the prediction horizon to 96, and compare two model variants: one with the projection $\Psi$ applied to the collider sub-module's output, and one without it. Figure 3 plots the training and test MSELoss over epochs for both variants. The results show that applying $\Psi$ substantially reduces the generalization gap, that is, the difference between training and test losses, indicating improved robustness and better generalization. This aligns with the theoretical insight that $\Psi$ eliminates components of the predictor that are spuriously correlated with collider spouse variables.

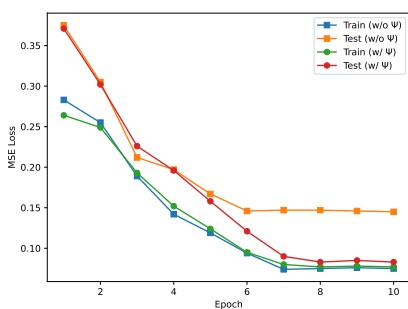

Figure 3: Effect of $\Psi$ Projection on Train/Test Loss

**Complexity and Efficiency.** To assess the practical applicability of CAIFormer, we compare its training and inference times with representative baseline models (PatchTST and iTransformer) under the same setting (input-96, predict-96 on the Weather dataset). Although CAIFormer adopts an all-to-one strategy that conceptually treats each variable separately, the computations are implemented in parallel, so inference time

Table 3: Comparison of training/inference time and forecasting performance.

| Model | Train Time (s) | Test Time (s) | MSE |
|---|---|---|---|
| CAIFormer | 242 | 3.2 | 0.147 |
| PatchTST | 253 | 3.3 | 0.177 |
| iTransformer | 150 | 2.2 | 0.174 |

does not scale linearly with the number of variables. As shown in Table 3, CAIFormer requires slightly higher training and inference costs than iTransformer, but remains comparable to PatchTST. More importantly, CAIFormer achieves a notable performance gain (MSE $0.147$ vs. $0.177/0.174$). These results indicate that the additional computational overhead of causal discovery is moderate and justified by the improved forecasting accuracy. We also provide a detailed analysis of the runtime of different causal discovery algorithms, showing that their computational overhead is small for medium-scale datasets and remains a one-time offline cost even for higher-dimensional data; see Appendix S for full results.

## 6 CONCLUSION

In this paper, we rethink the problem of MTSF from a causal perspective. We construct an SCM for variables in time series from observation data. For each target variable, we treat it independently and partition the history sequence into four sub-segments: endogenous, direct causal, collider causal, and spurious correlation. We introduce an all-to-one decoupled forecasting strategy, where the prediction for each variable relies solely on the first three causally relevant sub-segments, and these three sub-segments have different roles. Based on this strategy, we develop a novel model, CAIFormer. CAIFormer employs three blocks to model the first three causal sub-segments in isolation, then combines their predictions at the output layer. This design effectively removes irrelevant variables, captures the target's intrinsic pattern, and identifies related variables according to their distinct causal roles. Extensive experiments validate the effectiveness, interpretability, and robustness of CAIFormer.

## ETHICS STATEMENT

All authors have carefully reviewed and fully comply with the ICLR Code of Ethics. This research was conducted in accordance with the principles of integrity, fairness, and transparency. The work does not involve human subjects, personally identifiable information, or other sensitive data, and it poses no foreseeable risks of harm, misuse, or ethical concerns. All experiments were performed on publicly available datasets or synthetic data with proper documentation to ensure reproducibility. The authors declare that there are no conflicts of interest or violations of the ICLR Code of Ethics associated with this submission.

## REPRODUCIBILITY STATEMENT

We ensure reproducibility by detailing experimental settings, datasets, and hyperparameters in both Section 5 and Appendices F-G, and by providing full proofs of theoretical results in Appendix E. The source code is included in the supplementary materials to reproduce the proposed method.

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

APPENDIX

The appendix is organized into several sections:

- Appendix A states the use of LLM in the paper;
- Appendix B provides the notation;
- Appendix C provides the background in causality;
- Appendix D presents the proof of analysis in Section 3.4;
- Appendix E presents the proof of Theorem 3.1;
- Appendix F provides the dataset descriptions;
- Appendix G provides the details of the implementation;
- Appendix H demonstrates that CAIFormer uncovers meaningful causal structures and improves interpretability;
- Appendix I comprehensively evaluates the robustness of CAIFormer from three perspectives: random seed sensitivity, variable ordering, DAG perturbations, and causal sufficiency and consistency;
- Appendix J provides supplementary experiments to examine how CAIFormer behaves under different input window lengths;
- Appendix K provides a comparison with causal inference-based forecasting methods;
- Appendix L provides a comparison with channel-independent forecasting methods;
- Appendix M provides a detailed comparison between the hard removal of SCS variables and a soft alternative that explicitly models them;
- Appendix N details the implementation of the causal discovery algorithm PC;
- Appendix O provides a comparison with the nonlinear PC algorithm;
- Appendix P visualizes the DAGs extracted from datasets and the mask form of DAGs;
- Appendix Q prevents the comparison of the results among CAIFormer, iTransformer, and PatchTST;
- Appendix R provides the detailed definitions and causal masks;
- Appendix S provides the runtime analysis of causal discovery methods.

## A   USE OF LARGE LANGUAGE MODELS

Large language models (LLMs) were used solely as a general-purpose tool for grammar and language polishing. They were not involved in research ideation, methodology, analysis, or substantive writing. The authors take full responsibility for all contents of the paper.

## B   LIST OF NOTATIONS

We list the definitions of all notations from the main text as follows:

- ☐ **Time Series Symbols (Basic Class)**
  - $X = [x_1, \cdots, x_T] \in \mathbb{R}^{T \times D}$: Multivariate time series history(T time steps, D variables).
  - $Y = [x_{T+1}, \cdots, x_{T+S}] \in \mathbb{R}^{S \times D}$: Multivariate time series future(S time steps).
  - $x_t = [V_1^t, \cdots, V_D^t] \in \mathbb{R}^D$: the value of D variables in $t$-th time step.
  - $V_i^t \in \mathbb{R}$: the value of $i$-th variable in $t$-th time step.
  - $V_i^{t:t+\tau} \in \mathbb{R}^\tau$: the value of $i$-th variables in $\tau$ timesteps.
  - $T$: the lookback length.
  - $S$: the prediction length.

- $D_{\text{train}} = \{(X^i, Y^i)\}_{i=1}^{K}$: the set of train samples.
- $f^*$: an optimal predictor that $f^*(X) = Y$.
- $\mathcal{F}$: a hypothesis space that $f^* \in \mathcal{F}$.

□ **Causal Graph and Variable Categorization (Causal Class)**

- $G = (V, E)$: Causal DAG among variables.
- $V$: Set of variable nodes $\{V_1, V_2, ..., V_D\}$.
- $W_{\text{adjm}} \in \mathbb{R}^{D \times D}$: Adjacency matrix from PC algorithm.
- $V_i$: Target variable.
- $V_p$: Variables that have a direct causal influence on $V_i$.
- $V_k$: Variables that are directly influenced by $V_i$, denoted as $V_i \to V_k$ and each $V_k$ is not a collider.
- $V_c$: Variables that are directly influenced by $V_i$, denoted as $V_i \to V_c$ and each $V_c$ is a collider.
- $V_s$: The spouse variables of $V_i$.
- $\mathcal{S}_i^P$: Direct parents of variable $V_i$.
- $\mathcal{S}_i^K$: Direct children of variable $V_i$ exclude collider.
- $\mathcal{S}_i^C$: Direct children of variable $V_i$ and is a collider.
- $\mathcal{S}_i^S$: Spouse of variable $V_i$.
- $\mathcal{S}_{V_c}$: The set of collider.
- $\mathcal{S}_{V_s}$: The set of spouse variable.
- $Xi$: The node in DAG, denotes the $i$-th variable in dataset.

□ **Model Structure Variables (Module Class)**

- $H$: Number of patches.
- $P$: Length of patch.
- $d_E$: Embedding dimension of ESPB.
- $d_D$: Embedding dimension of DCSPB.
- $d_C$: Embedding dimension of CCSPB.
- $E_e$: Number of Encoder layers in ESPB.
- $E_d$: Number of Encoder layers in DCSPB.
- $E_c$: Number of Encoder layers in CCSPB.
- $\delta$: A normalization factor.
- $Y_e$: Output from Endogenous Sub-segment Prediction Block (ESPB).
- $Y_d$: Output from Direct Causal Sub-segment Prediction Block (DCSPB).
- $Y_c$: Output from Collider Causal Sub-segment Prediction Block (CCSPB).
- $Y_{\text{cat}}$: Concat of $Y_e$, $Y_d$, and $Y_c$.
- $D_{mask}$: Attention mask for direct causal variables.
- $CS_{mask}$: Attention mask for collider structure.
- $S_{mask}$: Attention mask for spouse variables.

□ **Theoretical Analysis Variables (Generalization and Projection Class)**

- $f_{\text{IP}}^*$: Optimal predictor under MSELoss for collider structure inputs.
- $\mathbb{E}[\cdot]$: Expectation operator.
- $\Phi$: Conditional expectation operator.
- $\Psi f = f - \Phi f$: Projection operator into kernel space.
- $\mathcal{F}_\Psi$: Function space satisfying collider constraints.
- $L^2(V)$: Space of square-integrable functions.
- $\Delta(f, \Psi f)$: Generalization gap between original and projected function.

**Algorithm 1** Pseudo-Code of CAIFormer

---

**Dataset**: Multivariate time series dataset $D_{\text{train}} = \{(X^i, Y^i)\}_{i=1}^K$, where $X^i = \{V_1^{0:T}, \cdots, V_D^{0:T}\} \in \mathbb{R}^{T \times D}$ and $Y^i = \{V_1^{T:T+S}, \cdots, V_D^{T:T+S}\} \in \mathbb{R}^{S \times D}$.

**Preprocessing**:

  1: Apply PC algorithm on $D$ to learn DAG and construct masks $D_{\text{mask}}, CS_{\text{mask}}, S_{\text{mask}}$.

**Input**: $X = (V_1^{0:T}, V_2^{0:T}, \cdots, V_D^{0:T})$

**Endogenous Sub-segment Prediction Block (ESPB):**

  1: $X_{\text{Patch}} = f_{\text{Patch}}(X)$                                    $\triangleright X_{\text{Patch}} \in \mathbb{R}^{H \times P \times D}$

  2: $X_{\text{Enc}}^0 = f_{\text{Emb}}^t(X_{\text{Patch}})$                               $\triangleright X_{\text{Enc}}^0 \in \mathbb{R}^{H \times D \times d_E}$

  3: **for** $e = 1$ to $E_e$ **do**

  4:     $X_{\text{Enc}}^e = \text{Encoder}(X_{\text{Enc}}^{e-1})$

  5: **end for**

  6: $Y_e = f_{\text{Projection}}^e(X_{\text{Enc}}^{E_e})$                            $\triangleright Y_e \in \mathbb{R}^{S \times D}$

**Direct Causal Sub-segment Prediction Block (DCSPB):**

  1: $X_{\text{Enc}}^0 = f_{\text{Emb}}^v(X)$                                 $\triangleright X_{\text{Enc}}^0 \in \mathbb{R}^{D \times d_D}$

  2: **for** $e = 1$ to $E_d$ **do**

  3:     $X_{\text{Enc}}^e = \text{Encoder}(X_{\text{Enc}}^{e-1}, D_{\text{mask}})$

  4: **end for**

  5: $Y_d = f_{\text{Projection}}^d(X_{\text{Enc}}^{E_d})$                           $\triangleright Y_d \in \mathbb{R}^{S \times D}$

**Collider Causal Sub-segment Prediction Block (CCSPB):**

  1: $X_{\text{Enc}}^0 = f_{\text{Emb}}^v(X)$                                 $\triangleright X_{\text{Enc}}^0 \in \mathbb{R}^{D \times d_C}$

  2: **for** $e = 1$ to $E_c$ **do**

  3:     $X_{\text{Enc}}^e = \text{Encoder}(X_{\text{Enc}}^{e-1}, CS_{\text{mask}})$

  4: **end for**

  5: $Z = f_{\text{Projection}}^c(X_{\text{Enc}}^{E_c})$                             $\triangleright Z \in \mathbb{R}^{S \times D}$

  6: $X_{\text{collider}} = X \odot S_{\text{mask}}$

  7: $EZ = \mathbb{E}[Z \mid X_{\text{collider}}] - C$

  8: $Y_c = Z - EZ$                                      $\triangleright Y_c \in \mathbb{R}^{S \times D}$

**Output Linear Layer:**

  1: $Y_{\text{cat}} = \text{Concat}(Y_e, Y_d, Y_c)$

  2: $Y = f_o(Y_{\text{cat}})$                                     $\triangleright Y \in \mathbb{R}^{S \times D}$

  3: **Output:** Predicted future values $Y = (V_1^{T:T+S}, \cdots, V_D^{T:T+S})$

---

## C BACKGROUND IN CAUSALITY

Causal relationships among variables play a crucial role in MTSF. By constructing a structural causal model, we can better understand the dependencies and independencies among variables, enabling us to build more accurate forecasting models (Pearl, 2009b). One core concept of causality is conditional independence, which is defined as:

**Definition C.1 (Conditional Independence (Dawid, 1979))** *Let $V = \{V_1, V_2, \cdots, V_D\}$ be a finite set of variables, $V_i$ is the $i$-th variable and $D$ is the number of variable, $P(\cdot)$ be a joint probability function over the variables in $V$, and $\mathcal{S}_X, \mathcal{S}_Y, \mathcal{S}_Z$ stand for three subsets of variables in $V$. Then, $\mathcal{S}_X$ and $\mathcal{S}_Y$ are said to be conditionally independent given $\mathcal{S}_Z$ if*

$$P(\mathcal{S}_X \mid \mathcal{S}_Y, \mathcal{S}_Z) = P(\mathcal{S}_X \mid \mathcal{S}_Z), \forall P(\mathcal{S}_Y, \mathcal{S}_Z) > 0. \tag{8}$$

*That is, $\mathcal{S}_Y$ does not provide any additional information for predicting $\mathcal{S}_X$, once given $\mathcal{S}_Z$. $\mathcal{S}_X \perp\!\!\!\perp \mathcal{S}_Y \mid \mathcal{S}_Z$ denotes the conditional independence of $\mathcal{S}_X$ and $\mathcal{S}_Y$ given $\mathcal{S}_Z$.*

Conditional independence relationships among variables form the basis of the SCM. In these models, a DAG, denoted as $G = (V, E)$, is typically used to represent the relationships among variables, where the node set $V = \{V_1, V_2, \cdots, V_D\}$ corresponds to random variables, and the edge set $E = \{(V_1, V_2), (V_2, V_3), \cdots\}$ represents causal relationships between variables. An SCM is built upon three fundamental structures: Chain, Fork and Collider. Any model containing at least three variables incorporates these key structures.

**Definition C.2 (Chain)** *A chain $V_p \rightarrow V_i \rightarrow V_c$ is a graphical structure involving three variables $V_p$, $V_i$, and $V_c$ in graph $G$, where $V_p$ has a directed edge to $V_i$ and $V_i$ has a directed edge to $V_c$. Here, $V_p$ causally influences $V_i$, and $V_i$ causally influences $V_c$, making $V_i$ a mediator.*

**Definition C.3 (Fork)** *A fork $V_b \leftarrow V_p \rightarrow V_i$ is a graphical structure involving $V_b$, $V_p$, and $V_i$, where $V_p$ is a common parent of both $V_b$ and $V_i$. $V_p$ causally influences $V_b$ and $V_i$.*

**Definition C.4 (Collider)** *A collider, also known as a V-structure, $V_i \rightarrow V_c \leftarrow V_s$, is a graphical structure involving three variables $V_i$, $V_c$, and $V_s$, where $V_c$ is a common child of both $V_i$ and $V_s$, $V_i$ and $V_s$ are not directly connected. Here, $V_i$ and $V_s$ causally influence $V_c$.*

In a chain structure, $V_p$ and $V_c$ are conditionally independent given $V_i$, formally, $V_p \perp\!\!\!\perp V_c \mid V_i$. In a fork structure, $V_b$ and $V_i$ are independent given $V_p$, $V_b$ provides no additional information about $V_i$, and vice versa, i.e., $V_b \perp\!\!\!\perp V_i \mid V_p$. In a collider structure, $V_i$ and $V_s$ are marginally independent, knowing $V_i$ does not provide information about $V_s$ and vice versa. However, when conditioning on the collider $V_c$, this independence is broken, making $V_i$ and $V_s$ dependent. Formally, $V_i \perp\!\!\!\perp V_s$ and $V_i \not\perp\!\!\!\perp V_s \mid V_c$. The related proofs are presented in Chapter Two of (Pearl, 2009a). The above independence relationships are fundamental for understanding the dependencies implied by an SCM, thereby facilitating tasks such as causal discovery and causal inference in MTSF.

## D PROOFS OF CONDITIONAL INDEPENDENCE FOR PATHS

Based on the analysis detailed in Appendix C, we exhaust all categories of relationships among other variables and the target variable in Figure 1d, obtaining: 1) **Path a:** $V_1 \rightarrow V_i$. Contains only two variables, $V_1$ points to $V_i$, and there are no other variables connected to the left of $V_1$ forming a chain ($V_2 \rightarrow V_1 \rightarrow V_i$) or a fork ($V_2 \leftarrow V_1 \rightarrow V_i$). In this case, $V_1 \not\perp\!\!\!\perp V_i$; 2) **Path b:** $V_D - \cdots - V_{i+1} - V_{i-1} - \cdots - V_3 - V_2 \rightarrow V_1 \rightarrow V_i$. Here, "$-$" represents that the casual relationship between two variables is unclear, e.g., "$-$" can be either "$\rightarrow$" or "$\leftarrow$", but we are not sure whether it is "$\rightarrow$" or "$\leftarrow$". In this case, when given $\mathcal{Z} = \{V_1, \cdots, V_{i-1}, V_{i+1}, \cdots V_D\}$, we can obtain that $\{V_1 \not\perp\!\!\!\perp V_i, V_j \perp\!\!\!\perp V_i\} \mid \mathcal{Z}$, where $V_j \in \mathcal{Z} \setminus V_1$. Thus, **Path b** equals to $V_1 \rightarrow V_i$; 3) **Path c:** $V_D - \cdots - V_{i+1} - V_{i-1} - \cdots - V_3 - V_2 \leftarrow V_1 \rightarrow V_i$. In this case, when given $\mathcal{Z} = \{V_1, \cdots, V_{i-1}, V_{i+1}, \cdots V_D\}$, we can obtain that $\{V_1 \not\perp\!\!\!\perp V_i, V_j \perp\!\!\!\perp V_i\} \mid \mathcal{Z}$, where $V_j \in \mathcal{Z} \setminus V_1$. Thus, **Path c** equals to $V_1 \rightarrow V_i$; 4) **Path d:** $V_i \rightarrow V_1$. Contains only two variables, $V_i$ points to $V_1$, and there are no other variables to the right of $V_1$ forming a chain ($V_i \rightarrow V_1 \rightarrow V_2$) or a collider ($V_i \rightarrow V_1 \leftarrow V_2$). In this case, $V_1 \not\perp\!\!\!\perp V_i$; 5) **Path e:** $V_i \rightarrow V_1 \rightarrow V_2 - V_3 - \cdots - V_{i-1} - V_{i+1} - \cdots - V_D$.

In this case, when given $\mathcal{Z} = \{V_1, \cdots, V_{i-1}, V_{i+1}, \cdots V_D\}$, we can obtain that $\{V_1 \not\perp\!\!\!\perp V_i, V_j \perp\!\!\!\perp V_i\} \mid \mathcal{Z}$, where $V_j \in \mathcal{Z} \setminus V_1$. Thus, **Path e** equals to $V_i \to V_1$; 6) **Path f:** $V_i \to V_1 \leftarrow V_2 - V_3 - \cdots - V_{i-1} - V_{i+1} - \cdots - V_D$. In this case, when given $\mathcal{Z} = \{V_1, \cdots, V_{i-1}, V_{i+1}, \cdots V_D\}$, we can obtain that $\{V_1 \not\perp\!\!\!\perp V_i, V_2 \not\perp\!\!\!\perp V_i, V_j \perp\!\!\!\perp V_i\} \mid \mathcal{Z}$, where $V_j \in \mathcal{Z} \setminus \{V_1, V_2\}$. Thus, **Path f** equals to $V_i \to V_1 \leftarrow V_2$.

Let $\mathcal{V} = \{V_1, V_2, \cdots, V_D\}$ denote the full set of variables. We focus on a target variable $V_i$ and consider its interactions with other variables via six typical causal paths (Path a to Path f). We use the notation $\mathcal{S}_A \perp\!\!\!\perp \mathcal{S}_B \mid \mathcal{S}_C$ to represent that $\mathcal{S}_A$ is conditionally independent of $\mathcal{S}_B$ given $\mathcal{S}_C$.

**Path a:** $V_1 \to V_i$. This is a direct causal link from $V_1$ to $V_i$. The joint distribution factorizes as $P(V_1, V_i) = P(V_1)P(V_i \mid V_1)$. Then the marginal and conditional probabilities are $P(V_i) = \int P(V_i \mid V_1)P(V_1)dV_1$. $P(V_i \mid V_1) \neq P(V_i)$ for any value of $V_1$, then $P(V_i \mid V_1) \neq P(V_i) \Rightarrow V_1 \not\perp\!\!\!\perp V_i$.

**Path b:** $V_D - \cdots - V_{i+1} - V_{i-1} - \cdots - V_3 - V_2 \to V_1 \to V_i$. This structure represents a causal chain beginning at $V_2$ and ending at $V_i$, where $V_3, \cdots, V_{i-1}, V_{i+1}, \cdots, V_D$ are unclear path direction variables in the left of $V_2$.

Let $\mathcal{Z} = \mathcal{V} \setminus \{V_1, V_i\}$. By d-separation, $V_1$ blocks all paths from $V_j$ to $V_i$, so $P(V_i \mid V_1, V_j, \mathcal{Z} \setminus \{V_j\}) = P(V_i \mid V_1)$. Hence $V_i \perp\!\!\!\perp V_j \mid \{V_1\} \cup (\mathcal{Z} \setminus \{V_j\})$ and $V_i \not\perp\!\!\!\perp V_1 \mid \mathcal{Z}$. Thus the entire structure simplifies to the direct influence $V_1 \to V_i$.

**Path c:** $V_D - \cdots - V_3 - V_2 \leftarrow V_1 \to V_i$. This is a fork structure where $V_1$ is a common cause of both $V_2$ and $V_i$, and the path may include $V_3, \cdots, V_{i-1}, V_{i+1}, \cdots, V_D$ connected to $V_2$ from the left with unclear direction paths.

Let $\mathcal{Z} = \mathcal{V} \setminus \{V_1, V_i\}$. Conditioning on $V_1$ d-separates $V_2$ (and all its upstream nodes) from $V_i$, so $P(V_i \mid V_1, V_2, \mathcal{Z} \setminus \{V_2\}) = P(V_i \mid V_1)$. Thus $V_i \perp\!\!\!\perp V_2 \mid \{V_1\} \cup (\mathcal{Z} \setminus \{V_2\})$ and $V_i \not\perp\!\!\!\perp V_1 \mid \mathcal{Z}$. Hence Path c also reduces to $V_1 \to V_i$.

**Path d:** $V_i \to V_1$. This is a direct causal link from $V_i$ to $V_1$. The joint distribution factorizes as $P(V_1, V_i) = P(V_i)P(V_1 \mid V_i)$. Then the marginal and conditional probabilities are $P(V_i) = \int P(V_i \mid V_1)P(V_1)dV_1$. $P(V_i \mid V_1) \neq P(V_i)$ for any value of $V_1$, then $P(V_i \mid V_1) \neq P(V_i) \Rightarrow V_1 \not\perp\!\!\!\perp V_i$.

**Path e:** $V_i \to V_1 \to V_2 - V_3 - \cdots - V_D$ This is a chain structure originating from $V_i$ and propagating through intermediate nodes. let $\mathcal{Z} = \mathcal{V} \setminus \{V_i\}$. $P(V_1 \mid V_i, \mathcal{Z} \setminus \{V_1\}) \neq P(V_1 \mid \mathcal{Z} \setminus \{V_1\}) \Rightarrow V_1 \not\perp\!\!\!\perp V_i \mid \mathcal{Z}$. For $j \geq 2$, the dependency is blocked by known $V_1$: $P(V_j \mid V_i, \mathcal{Z} \setminus \{V_j\}) = P(V_j \mid \mathcal{Z} \setminus \{V_j\}) \Rightarrow V_j \perp\!\!\!\perp V_i \mid \mathcal{Z}$. Thus, the path reduces to $V_i \to V_1$.

**Path f:** $V_i \to V_1 \leftarrow V_2 - V_3 - \cdots - V_D$. This is a collider structure where $V_1$ is a common result of $V_i$ and $V_2$, and $V_2$ may be influenced by the right variables with unclear direction paths. let $\mathcal{Z} = \mathcal{V} \setminus \{V_i\}$.

Since $V_1$ is observed, the dependency between $V_i$ and $V_2$ is activated: $P(V_i \mid \mathcal{Z}) \neq P(V_i \mid \mathcal{Z} \setminus \{V_1, V_2\}) \Rightarrow V_2 \not\perp\!\!\!\perp V_i \mid \mathcal{Z}$. Meanwhile, $P(V_i \mid \mathcal{Z}) = P(V_i \mid \mathcal{Z} \setminus \{V_j\}) \Rightarrow V_j \perp\!\!\!\perp V_i \mid \mathcal{Z} \forall j \in \{3, \cdots, i-1, i+1, \cdots, D\}$. Hence both $V_1$ and $V_2$ are relevant under conditioning, confirming the collider structure $V_i \to V_1 \leftarrow V_2$.

# E PROOF OF THEOREM 3.1

**Proof E.1** *The conditional expectation* $\Pi : Z \in L^2(\Omega) \mapsto \mathbb{E}[Z \mid \mathcal{S}_{V_s}]$ *defines an orthogonal projection onto the space of $\mathcal{S}_{V_s}$-measurable random variables with finite variance $L^2(\Omega, \sigma(\mathcal{S}_{V_s}), P)$. Thus, its range and null space are orthogonal in $L^2(\Omega)$. Let $f \in L^2(V)$. We have $\Phi f(\mathcal{S}_{V_c}, \mathcal{S}_{V_s}) = \mathbb{E}[f(\mathcal{S}_{V_c}, \mathcal{S}_{V_s}) \mid \mathcal{S}_{V_s}] - C = \Pi f(\mathcal{S}_{V_c}, \mathcal{S}_{V_s}) - C$ hence $\Phi f(\mathcal{S}_{V_c}, \mathcal{S}_{V_s})$ is in the range of $\Pi$. On the other hand,*

$$\mathbb{E}[\Psi f(\mathcal{S}_{V_c}, \mathcal{S}_{V_s}) \mid \mathcal{S}_{V_s}] = \mathbb{E}[f(\mathcal{S}_{V_c}, \mathcal{S}_{V_s}) \mid \mathcal{S}_{V_s}] - \mathbb{E}[\Phi f(\mathcal{S}_{V_c}, \mathcal{S}_{V_s}) \mid \mathcal{S}_{V_s}]$$
$$= \mathbb{E}[f(\mathcal{S}_{V_c}, \mathcal{S}_{V_s}) \mid \mathcal{S}_{V_s}] - \mathbb{E}[f(\mathcal{S}_{V_c}, \mathcal{S}_{V_s}) \mid \mathcal{S}_{V_s}] = 0. \tag{9}$$

*Therefore, $\Psi f(\mathcal{S}_{V_c}, \mathcal{S}_{V_s})$ is in the null space of $\Pi$. Finally, because $V_i \perp\!\!\!\perp \mathcal{S}_{V_s}$ we have $\mathbb{E}[V_i \mid \mathcal{S}_{V_s}] = \mathbb{E}[V_i] = C$ by assumption, therefore $V_i$ is also in the null space of $\Pi$.*

*Hence, adopting this random variable view, the desired result simply follows from $L^2(\Omega)$ orthogonality:*

$$
\begin{aligned}
\Delta(f, \Psi f) &= \mathbb{E}[(V_i - f(\mathcal{S}_{V_c}, \mathcal{S}_{V_s}))^2] - \mathbb{E}[(V_i - \Psi f(\mathcal{S}_{V_c}, \mathcal{S}_{V_s}))^2] \\
&= \|V_i - f(\mathcal{S}_{V_c}, \mathcal{S}_{V_s})\|^2_{L^2(\Omega)} - \|V_i - \Psi f(\mathcal{S}_{V_c}, \mathcal{S}_{V_s})\|^2_{L^2(\Omega)} \\
&= \|V_i - \Psi f(\mathcal{S}_{V_c}, \mathcal{S}_{V_s}) - \Phi f(\mathcal{S}_{V_c}, \mathcal{S}_{V_s})\|^2_{L^2(\Omega)} - \|V_i - \Psi f(\mathcal{S}_{V_c}, \mathcal{S}_{V_s})\|^2_{L^2(\Omega)} \\
&= \|V_i - \Psi f(\mathcal{S}_{V_c}, \mathcal{S}_{V_s})\|^2_{L^2(\Omega)} + \|\Phi f(\mathcal{S}_{V_c}, \mathcal{S}_{V_s})\|^2_{L^2(\Omega)} - \|V_i - \Psi f((\mathcal{S}_{V_c}, \mathcal{S}_{V_s})\|^2_{L^2(\Omega)} \\
&= \mathbb{E}[\Phi f(\mathcal{S}_{V_c}, \mathcal{S}_{V_s})^2] = \|\Phi f\|^2_{L^2(\Omega)}.
\end{aligned}
\tag{10}
$$

# F  DATASET DESCRIPTIONS

In this paper, we conduct tests using six real-world datasets. These datasets include:

(1) The ETT dataset contains 7 factors of electricity transformer from July 2016 to July 2018, consists of two sub-datasets, ETT1 and ETT2, collected from electricity transformers at two different stations. Each sub-dataset is available in two resolutions (15 minutes and 1 hour), containing multiple load series and a single oil temperature series. (2) Weather covers 21 meteorological variables recorded at 10-minute intervals throughout the year 2020. The data was collected by the Max Planck Institute for Biogeochemistry's Weather Station, providing valuable meteorological insights. (3) Exchange Rate contains daily currency exchange rates for eight countries, spanning from 1990 to 2016. (4) ECL contains the electricity consumption of 370 clients for short-term forecasting while it contains the electricity consumption of 321 clients for long-term forecasting. It is collected since 01/01/2011. The data sampling interval is every 15 minutes; (5) Traffic gathers hourly road occupancy rates from 862 sensors on San Francisco Bay area freeways, covering the period from January 2015 to December 2016.

We follow the same data processing and chronological train-validation-test split protocol as used in iTransformer (Liu et al., 2024) to avoid data leakage issues. The details of the datasets are provided in Table 4.

Table 4: Detailed descriptions of datasets. Dim denotes the number of variables in each dataset. Prediction Length denotes the number of future time points to predict; each dataset includes four different forecasting horizons. Time steps represents the number of time points. Percentage indicates the proportions of the dataset allocated to Train, Validation, and Test splits. Frequency specifies the sampling interval between consecutive time points.

| Dataset | Dim | Prediction Length | Time steps | Percentage | Frequency | Information |
|---------|-----|-------------------|------------|------------|-----------|-------------|
| ETTh1,ETTh2 | 7 | {96, 192, 336, 720} | 17420 | (60%, 20%, 20%) | Hourly | Electricity |
| ETTm1,ETTm2 | 7 | {96, 192, 336, 720} | 69680 | (60%, 20%, 20%) | 15min | Electricity |
| Exchange | 8 | {96, 192, 336, 720} | 7588 | (70%, 10%, 20%) | Daily | Economy |
| Weather | 21 | {96, 192, 336, 720} | 52560 | (70%, 10%, 20%) | 10min | Weather |
| ECL | 321 | {96, 192, 336, 720} | 26304 | (70%, 10%, 20%) | Hourly | Electricity |
| Traffic | 862 | {96, 192, 336, 720} | 17451 | (70%, 10%, 20%) | Hourly | Transportation |

# G  IMPLEMENTATION DETAILS

All experiments are implemented in PyTorch (Paszke et al., 2019) and trained on NVIDIA V100 32GB GPUs. The model is optimized using the Adam optimizer (Kingma & Ba, 2015). The initial learning rate is selected from $\{10^{-3}, 10^{-4}\}$, and the batch size from $\{16, 32, 64\}$, based on test set performance. Each prediction block in CAIFormer, ESPB, DCSPB, and CCSPB contains a stack of Transformer encoder layers, where the number of layers is selected from $\{1, 2, 3\}$. The hidden dimension for each block is selected independently from $\{64, 128, 256, 512\}$. Training is performed for up to 10 epochs using Mean Squared Error (MSE) as the loss function. An early stopping strategy is employed: if validation loss does not improve for three consecutive epochs, training is terminated. After training, the model checkpoint with the best performance on the test set is selected and used

for final evaluation. For evaluation, we report both MSE and Mean Absolute Error (MAE). All experiments are repeated five times with different random seeds, and the average results are reported.

# H  ADDITIONAL INTERPRETABILITY EXPERIMENTS

To further demonstrate that CAIFormer uncovers meaningful causal structures and improves interpretability, we conduct two complementary experiments.

## H.1  ABLATION ON ALL-TO-ONE FORECASTING PARADIGM.

To isolate the effect of the proposed all-to-one decomposition from architectural choices, we replace CAIFormer with a minimal MLP forecaster and compare two training paradigms on ETTh1 (input-96, predict-96): (a) All-to-all MLP: each target is predicted from the histories of all variables; (b) All-to-one MLP(Ours): for each target, an independent MLP consumes only its ES, DCS, and CCS sub-segments defined by the discovered DAG. As shown in Table 5, the all-to-one MLP achieves equal or better performance across all targets (strictly better on 6/7, tied on 1/7), reducing average MSE from $0.395$ to $0.392$. This indicates that causal decomposition effectively reduces spurious interference and benefits prediction even with a lightweight architecture.

Table 5: MLP on ETTh1 (input-96, predict-96): all-to-all vs. all-to-one (ES+DCS+CCS).

| Target | X1 | X2 | X3 | X4 | X5 | X6 | X7 |
|---|---|---|---|---|---|---|---|
| All-to-all (All vars) | 0.392 | 0.403 | 0.382 | 0.386 | 0.388 | 0.391 | 0.420 |
| All-to-one (ES+DCS+CCS) | 0.390 | 0.403 | 0.380 | 0.383 | 0.385 | 0.386 | 0.419 |

## H.2  ATTRIBUTION-BASED VARIABLE REMOVAL.

We further evaluate whether CAIFormer's forecasts align with causal insights using the Weather dataset with OT as target. We adopt the Integrated Gradients (IG) algorithm (Sundararajan et al., 2017) to measure variable contributions and select the top-5 most important inputs for both CAIFormer and iTransformer. These are grouped into: (a) variables shared by both models, (b) unique to CAIFormer, and (c) unique to iTransformer. We then remove sets (b) and (c) respectively and retrain both models. Results in Table 6 show that removing CAIFormer-specific variables (set b) causes a large performance drop for CAIFormer (MSE $0.105 \rightarrow 0.217$) and a smaller decline for iTransformer, whereas removing iTransformer-specific variables (set c) has little effect on CAIFormer but a comparable drop for iTransformer. These results suggest that CAIFormer relies more on genuinely causal variables, while iTransformer is more dependent on substitutable correlations.

We then remove the variables in set (b) or set (c) and retrain both models. Table 6 shows the results. Removing CAIFormer-specific variables (set b) causes a large drop in CAIFormer's performance (MSE $0.105 \rightarrow 0.217$) and a smaller decline for iTransformer. In contrast, removing iTransformer-specific variables (set c) has little effect on CAIFormer ($0.105 \rightarrow 0.114$), while iTransformer suffers a comparable drop to the removal of set (b).

Table 6: MSE on Weather dataset under variable removal.

| Model | CAIFormer | iTransformer |
|---|---|---|
| w/ all | 0.105 | 0.132 |
| w/o set (b) | 0.217 | 0.152 |
| w/o set (c) | 0.114 | 0.147 |

These findings suggest that CAIFormer relies more on variables with genuine causal relevance, whereas iTransformer is more sensitive to variables that can be substituted by correlations. This demonstrates that CAIFormer not only improves forecasting accuracy but also enhances interpretability by uncovering meaningful causal links.

## I ROBUSTNESS EVALUATION

We comprehensively evaluate the robustness of CAIFormer from three perspectives: random seed sensitivity, variable ordering, DAG perturbations, and causal sufficiency.

### I.1 RANDOM SEED SENSITIVITY.

We repeat experiments five times with different random seeds $\{2021, 2022, 2023, 2024, 2025\}$ on the ETTm1, ETTh1, and Exchange-rate datasets. Table 7 reports the standard deviations of CAIFormer's performance, which are consistently low, demonstrating stable and reproducible results.

Table 7: Robustness evaluation of CAIFormer. The reported results (MSE and MAE) reflect the mean and standard deviation computed over five independent runs with different random seeds.

| Dataset | ETTm1 | | ETTh1 | | Exchange | |
|---|---|---|---|---|---|---|
| Horizon | MSE | MAE | MSE | MAE | MSE | MAE |
| 96 | 0.327±0.002 | 0.364±0.001 | 0.382±0.002 | 0.399±0.003 | 0.083±0.000 | 0.201±0.002 |
| 192 | 0.361±0.002 | 0.377±0.003 | 0.429±0.003 | 0.426±0.001 | 0.173±0.001 | 0.295±0.001 |
| 336 | 0.391±0.003 | 0.402±0.001 | 0.474±0.002 | 0.449±0.002 | 0.302±0.002 | 0.395±0.001 |
| 720 | 0.449±0.002 | 0.437±0.004 | 0.495±0.003 | 0.483±0.004 | 0.842±0.005 | 0.688±0.003 |

### I.2 VARIABLE ORDERING SENSITIVITY.

Variables in multivariate time series (e.g., temperature, pressure) have no inherent order, and altering their positions should not affect causal semantics. To verify this, we randomly shuffled all variables three times on the Weather dataset. The PC algorithm consistently produced the same causal graph, and retraining CAIFormer (input-96, predict-96) resulted in negligible performance changes ($\Delta$MSE < 0.001, $\Delta$MAE < 0.001). This confirms that CAIFormer is robust to variable ordering.

### I.3 ROBUSTNESS TO DAG PERTURBATIONS.

To quantify the impact of imperfect causal discovery, we inject synthetic noise into the learned DAG on the Weather dataset ($D = 21$, lookback/prediction = 96). For each perturbation ratio $p \in \{0\%, 5\%, 10\%, 15\%, 20\%, 25\%, 30\%\}$, we consider: (i) False-Negative (FN): delete $p\%$ of true edges; (ii) False-Positive (FP): add $p\%$ spurious edges; (iii) Shuffle (FN+FP): delete $p/2\%$ true edges and add the same number of spurious edges. We retrain CAIFormer with identical hyperparameters and report MSE/MAE. Results in Fig. 4 show a clear monotonic degradation as $p$ increases. Across perturbation types, FN causes the largest error, Shuffle is intermediate, and FP is the least harmful at the same $p$ (i.e., FN > Shuffle > FP), indicating that removing true causal links is more detrimental than introducing extra distractors.

### I.4 ROBUSTNESS TO CAUSAL DISCOVERY ALGORITHMS.

We cross-validate the PC estimated structures using multiple causal discovery methods. On the ETTh1 dataset, we apply PC, FCI, and PCMCI to estimate causal graphs, and list their corresponding adjacency matrices (encoded with $-1/0/1$ for edge directions; $-1$ indicates a directed edge from the row variable to the column variable, 0 indicates no edge, and 1 indicates the opposite direction) in Table 8. Based on these matrices, we compute the Jaccard similarity of the edge sets and obtain 95% between PC and FCI, and 90% between PC and PCMCI. This shows that even when using FCI or PCMCI, the resulting structures remain highly consistent with the PC graph at the level of edges.

We further replace the PC-based DAG in CAIFormer with the FCI-based and PCMCI-based DAGs, and run forecasting experiments on ETTh1 with an input length of 96. As reported in Table 9, the MSE/MAE of the PC-based, FCI-based, and PCMCI-based variants differ by less than 0.01 across all four horizons (96/192/336/720). Overall, these results indicate that, under the same model architecture, switching between different mainstream causal discovery algorithms has almost no effect on the forecasting performance of CAIFormer.

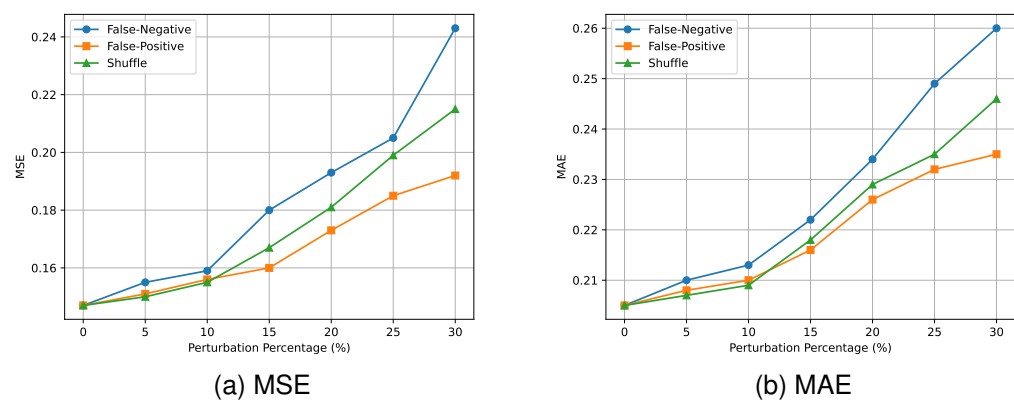

(a) MSE                              (b) MAE

Figure 4: MSE/MAE vs. Perturbation Percentage on Weather Dataset

Table 8: Adjacency matrices estimated by PC, FCI, and PCMCI on the ETTh1 dataset. Here $-1$ denotes a directed edge from the row variable to the column variable, $0$ indicates no edge, and $1$ denotes the opposite direction.

| PC/FCI/PCMCI | 1 | 2 | 3 | 4 | 5 | 6 | 7 |
|---|---|---|---|---|---|---|---|
| 1 | 0/0/0 | -1/1/1 | 0/-1/1 | -1/-1/-1 | 1/1/-1 | 1/-1/1 | 0/0/0 |
| 2 | 1/-1/-1 | 0/0/0 | 0/-1/0 | -1/-1/1 | 1/1/1 | 1/0/1 | 1/1/-1 |
| 3 | 0/1/-1 | 0/1/0 | 0/0/0 | -1/-1/1 | 1/1/1 | 1/1/1 | -1/-1/0 |
| 4 | 1/1/1 | 1/1/-1 | 1/1/-1 | 0/0/0 | 1/0/0 | 1/1/0 | 1/0/1 |
| 5 | -1/-1/1 | -1/-1/-1 | -1/-1/-1 | -1/0/0 | 0/0/0 | 0/-1/1 | -1/-1/-1 |
| 6 | -1/1/-1 | -1/0/-1 | -1/-1/-1 | -1/-1/0 | 0/-1/-1 | 0/0/0 | 0/0/-1 |
| 7 | 0/0/0 | -1/-1/1 | 1/1/0 | -1/0/-1 | 1/1/1 | 0/0/1 | 0/0/0 |

## I.5 CAUSAL SUFFICIENCY STABILITY.

The PC algorithm operates under the assumption of causal sufficiency, i.e., that all relevant confounders are observed. When this assumption is violated, PC may introduce spurious edges due to unobserved common causes. Nevertheless, our framework is general and does not rely on PC in particular: in principle, it can incorporate alternative causal discovery methods that explicitly handle latent confounders, such as FCI or tsFCI. Although causal sufficiency cannot be directly verified on the available datasets, we conduct an empirical stress test by examining the stability of the learned causal structures across different data splits. Specifically, on the ETTh1 dataset, we extract DAGs separately from the training, test, and full datasets and compute their pairwise Jaccard similarity. The resulting scores are 80.9% (train vs. test) and 90.4% (train vs. full), indicating that the inferred causal graphs are highly consistent across splits. This consistency indirectly supports the plausibility of the causal sufficiency assumption in our setting. The corresponding adjacency matrices are reported in Table 10.

## I.6 ENSEMBLE AND BOOTSTRAPPING OF DAGS

To further examine how uncertainties in the causal discovery phase influence CAIFormer, we additionally explore ensemble and bootstrapping schemes for constructing potentially more robust DAGs. On the ETTh1 dataset, we first consider an ensemble design in which we run PC, FCI, and PCMCI on the full training set and obtain an aggregated DAG via majority voting across the three estimated edge sets. This approach aims to reduce algorithm-specific variability by capturing structural information that remains stable across different causal discovery procedures. We also investigate a bootstrapping alternative designed to assess sampling variability: specifically, we generate ten bootstrap resamples of the training set, each containing 10% of the original data sampled with replacement, run the PC algorithm on each resample, and obtain a bootstrap-aggregated DAG

Table 9: Forecasting performance of CAIFormer on ETTh1 (input-96) using DAGs estimated by PC, FCI, and PCMCI. The differences across causal discovery algorithms are within 0.01 for all horizons.

| Horizon | PC-based | | FCI-based | | PCMCI-based | |
|---|---|---|---|---|---|---|
| | MSE | MAE | MSE | MAE | MSE | MAE |
| 96 | 0.372 | 0.399 | 0.374 | 0.399 | 0.372 | 0.400 |
| 192 | 0.429 | 0.426 | 0.428 | 0.425 | 0.430 | 0.426 |
| 336 | 0.464 | 0.449 | 0.464 | 0.450 | 0.463 | 0.447 |
| 720 | 0.495 | 0.483 | 0.496 | 0.485 | 0.494 | 0.483 |

Table 10: Adjacency matrices extracted by PC from training, test, and full ETTh1 datasets. Here $-1$ denotes a directed edge from the row variable to the column variable, $0$ indicates no edge, and $1$ denotes the opposite direction.

| Train/Test/All | 1 | 2 | 3 | 4 | 5 | 6 | 7 |
|---|---|---|---|---|---|---|---|
| 1 | 0/0/0 | -1/0/1 | 0/-1/1 | -1/-1/-1 | 1/-1/-1 | 1/-1/0 | 0/0/0 |
| 2 | 1/0/-1 | 0/0/0 | 0/-1/0 | -1/-1/1 | 1/1/1 | 1/0/1 | 1/1/1 |
| 3 | 0/1/-1 | 0/1/0 | 0/0/0 | -1/-1/1 | 1/-1/1 | 1/-1/1 | -1/0/0 |
| 4 | 1/1/1 | 1/1/-1 | 1/1/-1 | 0/0/0 | 1/0/0 | 1/1/0 | 1/0/1 |
| 5 | -1/1/1 | -1/-1/-1 | -1/1/-1 | -1/0/0 | 0/0/0 | 0/-1/1 | -1/-1/-1 |
| 6 | -1/1/0 | -1/0/-1 | -1/1/-1 | -1/-1/0 | 0/-1/-1 | 0/0/0 | 0/-1/0 |
| 7 | 0/0/0 | -1/-1/-1 | 1/0/0 | -1/0/-1 | 1/1/1 | 0/1/0 | 0/0/0 |

by retaining only those edges that appear in at least six of the ten estimated graphs. Appendix Tables 11 and 12 report the adjacency matrices of the single PC graph, the ensemble graph, and the bootstrap graph, as well as the corresponding forecasting results when each serves as the structural prior for CAIFormer. Compared with the single PC-based graph, the ensemble DAG leads to a slight degradation in forecasting performance across all horizons, while the bootstrap-based DAG produces a more marked degradation. We attribute this behavior to the fact that bootstrapping substantially reduces the effective sample size for each causal discovery run, injecting additional statistical noise into the estimated structures; although majority voting mitigates some of this variability, the resulting graph is still less reliable than a single DAG learned from the full dataset.

Table 11: Adjacency matrices estimated by PC, ensemble voting, and bootstrap aggregation on ETTh1. Entries $-1/0/1$ denote a directed edge from row to column, no edge, and the opposite direction, respectively.

| PC / Ensemble / Bootstrap | 1 | 2 | 3 | 4 | 5 | 6 | 7 |
|---|---|---|---|---|---|---|---|
| 1 | 0/0/0 | -1/1/0 | 0/0/1 | -1/-1/-1 | 1/1/-1 | 1/1/-1 | 0/0/0 |
| 2 | 1/-1/0 | 0/0/0 | 0/0/0 | -1/-1/1 | 1/1/0 | 1/1/0 | 1/1/-1 |
| 3 | 0/0/-1 | 0/0/0 | 0/0/0 | -1/-1/1 | 1/1/1 | 1/1/0 | -1/-1/0 |
| 4 | 1/1/1 | 1/1/-1 | 1/1/-1 | 0/0/0 | 1/0/0 | 1/1/0 | 1/1/1 |
| 5 | -1/-1/1 | -1/-1/0 | -1/-1/-1 | -1/0/0 | 0/0/0 | 0/0/1 | -1/-1/0 |
| 6 | -1/-1/1 | -1/-1/0 | -1/-1/0 | -1/-1/0 | 0/-1/-1 | 0/0/0 | 0/0/-1 |
| 7 | 0/0/0 | -1/-1/1 | 1/1/0 | -1/-1/-1 | 1/1/0 | 0/0/1 | 0/0/0 |

## J  EFFECT OF LOOK-BACK WINDOW LENGTH

Prior studies (Liu et al., 2024; Nie et al., 2023) demonstrate that longer historical windows can improve forecasting accuracy. To verify this trend for CAIFormer, we fix the prediction horizon to 96 and vary the input window length from $\{96, 192, 336, 512\}$ on the Traffic dataset. The results are illustrated in Fig. 5. As the look-back window increases, the MSE consistently decreases, confirming that CAIFormer benefits from extended historical information in a manner similar to prior Transformer-based methods.

Table 12: Forecasting performance of CAIFormer on ETTh1 when using PC-based, ensemble-based, or bootstrap-based DAGs (input length = 96). Ensemble and bootstrap aggregation do not improve performance and instead introduce noticeable degradation.

| Horizon | PC-based MSE | PC-based MAE | Ensemble-based MSE | Ensemble-based MAE | Bootstrap-based MSE | Bootstrap-based MAE |
|---|---|---|---|---|---|---|
| 96 | 0.372 | 0.399 | 0.382 | 0.405 | 0.402 | 0.421 |
| 192 | 0.429 | 0.426 | 0.437 | 0.434 | 0.465 | 0.459 |
| 336 | 0.464 | 0.449 | 0.480 | 0.461 | 0.510 | 0.492 |
| 720 | 0.495 | 0.483 | 0.511 | 0.493 | 0.559 | 0.531 |

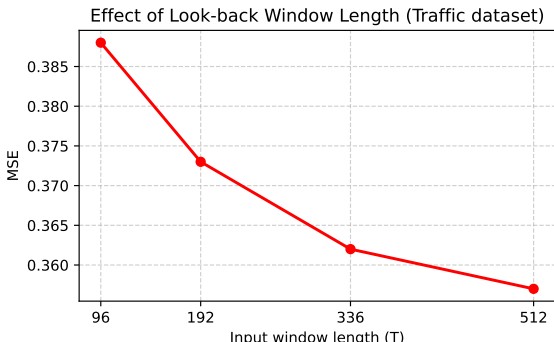

Figure 5: Effect of look-back window length on the Traffic dataset (prediction window = 96).

## K  COMPARE WITH CAUSAL TSF METHOD

We also compare with recent causal inference-based forecasting methods. In particular, we include Causal-TSF (Gong et al., 2025) as a representative baseline, and follow its experimental setting and evaluation metrics to ensure consistency. On the Traffic dataset, CAIFormer achieves consistently lower RSE and higher CORR across forecasting horizons (Table 13), highlighting the benefit of incorporating causal structures into forecasting.

Table 13: Comparison with a causal inference-based TSF method (Traffic dataset).

| Methods | Metric | 3 | 6 | 12 | 24 |
|---|---|---|---|---|---|
| Causal-TSF | RSE↓ | 0.4012 | 0.4181 | 0.4290 | 0.4349 |
| | CORR↑ | 0.9033 | 0.8951 | 0.8892 | 0.8847 |
| CAIFormer | RSE↓ | 0.3596 | 0.3711 | 0.3822 | 0.3885 |
| | CORR↑ | 0.9422 | 0.9175 | 0.9018 | 0.8984 |

## L  COMPARE WITH CI TSF METHOD

Multivariate time series forecasting methods can be divided into channel-independent (CI) and channel-dependent (CD) approaches. CI methods (e.g., DLinear, PatchTST) treat each variable independently, while CD methods explicitly model inter-variable dependencies. Following prior literature, we include strong CI baselines, and also three representative CD models: iTransformer, Crossformer, and TiDE.

In addition, we compare with TimeXer (Wang et al., 2024), a recent CD method for large-scale multivariate forecasting. Table 14 reports results on the ECL and Traffic datasets, both with a large

number of variables. CAIFormer consistently outperforms TimeXer across all horizons in both MSE and MAE, highlighting the effectiveness of causal masks for capturing inter-variable relationships.

Table 14: Comparison with CD baseline TimeXer on ECL and Traffic datasets.

| Dataset | Horizon | CAIFormer | | TimeXer | |
|---|---|---|---|---|---|
| | | MSE | MAE | MSE | MAE |
| ECL | 96 | 0.139 | 0.235 | 0.140 | 0.242 |
| | 192 | 0.155 | 0.245 | 0.157 | 0.256 |
| | 336 | 0.171 | 0.262 | 0.176 | 0.275 |
| | 720 | 0.209 | 0.303 | 0.211 | 0.306 |
| | Avg | 0.168 | 0.261 | 0.171 | 0.270 |
| Traffic | 96 | 0.388 | 0.261 | 0.428 | 0.271 |
| | 192 | 0.411 | 0.269 | 0.448 | 0.282 |
| | 336 | 0.424 | 0.276 | 0.473 | 0.289 |
| | 720 | 0.461 | 0.297 | 0.516 | 0.307 |
| | Avg | 0.421 | 0.280 | 0.466 | 0.287 |

## M  HARD PROCESSING OF ABLATION SCS VARIABLES

In our framework, variables in the SCS (Spurious Correlation Set) are defined under structural causal models as having no direct causal effect on the target variable. Therefore, SCS variables, as unnecessary signals, may introduce additional spurious correlations if included in the predictive model. Given this, this paper employs a "hard removal" strategy to completely remove SCS variables from the model input. From a causal perspective, excluding SCS variables is the most direct way to enforce the causal semantics implied by the discovered DAG. If a variable is determined to be irrelevant to the target variable, providing it to the model, even with strong regularization, may lead to the model exploiting spurious temporal correlations. This hard removal design reflects the intention to ensure the model strictly conforms to the causal structure and avoids unnecessary confounding signals.

To validate the rationale for this setup, we designed an explicit test. In addition to the existing three modules ESPB, DCSPB, and CCSPB, we added an extra module, SCSPB, which only accepts SCS variables. We then fused the predictions from all four modules to obtain the final output. Unlike hard removal strategies, this "mild" approach allows the model to access SCS variables and relies on the model architecture to learn to ignore or reduce their weights. We evaluated CAIFormer and CAIFormer+SCSPB on ETTh1 and ETTm1 using prediction periods $\{96, 192, 336, 720\}$. The results are summarized in Table 15.

Table 15: Comparison between CAIFormer (hard removal of SCS variables) and CAIFormer+SCSPB (soft handling). Adding SCSPB consistently leads to slightly worse MSE/MAE across datasets and horizons.

| Model | Metric | ETTh1 | | | | ETTm1 | | | |
|---|---|---|---|---|---|---|---|---|---|
| | | 96 | 192 | 336 | 720 | 96 | 192 | 336 | 720 |
| CAIFormer | MSE | 0.372 | 0.429 | 0.464 | 0.495 | 0.327 | 0.361 | 0.391 | 0.449 |
| | MAE | 0.399 | 0.426 | 0.449 | 0.483 | 0.364 | 0.377 | 0.402 | 0.437 |
| CAIFormer+SCSPB | MSE | 0.380 | 0.433 | 0.469 | 0.499 | 0.332 | 0.369 | 0.401 | 0.458 |
| | MAE | 0.401 | 0.433 | 0.452 | 0.487 | 0.367 | 0.381 | 0.408 | 0.444 |

## N  CAUSAL DISCOVERY ALGORITHM

In this section, we detail the Peter-Clark (PC) algorithm (Algorithm 2) used for causal discovery. The PC algorithm assumes that the data are sampled from a faithful joint distribution $\hat{P}$ over a variable

set $V$; that is, every conditional independence in $\hat{P}$ corresponds to d-separation in the true causal DAG, and vice-versa. The outcome is a completed partially directed acyclic graph (CPDAG), denoted $H(\hat{P})$. A CPDAG represents the entire Markov-equivalence class of the true (but unobserved) causal DAG: edges that are compelled in every member of the class appear directed, whereas reversible edges remain undirected. In this mixed graph, oriented edges encode compelled causal directions, whereas undirected edges denote Markov-equivalent ambiguities.

---

**Algorithm 2** Causal Discovery Algorithm-PC

---

**Input**: $\hat{P}$, a stable distribution on a set $V$ of variables;
**Output**: A pattern $H(\hat{P})$ compatible with $\hat{P}$.

1: Initialize complete undirected graph $G$ on $V$
2: depth $\leftarrow 0$
3: **repeat**
4:     **for** each ordered pair $(a, b)$ adjacent in $G$ **do**
5:         **for** each $S \subseteq \mathrm{Adj}(a) \setminus \{b\}$ with $|S| =$depth **do**
6:             **if** $a \perp\!\!\!\perp b \mid S$ in $\hat{P}$ **then**
7:                 Remove edge $a-b$ from $G$;   $\mathrm{Sepset}[a][b] \leftarrow S$
8:                 **break**
9:             **end if**
10:         **end for**
11:     **end for**
12:     depth $\leftarrow$ depth+1
13: **until** no edge removed at current depth
14: **for** each non-adjacent $a, b$ with common neighbor $c$ **do**
15:     **if** $c \notin \mathrm{Sepset}[a][b]$ **then**
16:         Orient $a \rightarrow c \leftarrow b$                             (v-structure)
17:     **end if**
18: **end for**
19: **while** any Meek rule applies without creating a cycle **do**
20:     Orient the corresponding edge
21: **end while**
22: **return** $G$ as CPDAG $H(\hat{P})$

---

# O   VALIDATION OF LINEAR VS. NONLINEAR PC

The PC algorithm is not restricted to linear dependence measures; its conditional independence tests can be replaced by nonlinear alternatives, such as kernel-based tests. To assess whether this choice affects CAIFormer's performance, we additionally ran experiments with a nonlinear-PC variant on six benchmark datasets. Table 16 reports the MSE and MAE results. Across all datasets and horizons, the differences between the linear and nonlinear variants are within 0.01 on average, confirming that the choice of linear-PC in our main experiments is sufficient and does not affect the conclusions.

Table 16: CAIFormer results with nonlinear-PC algorithm (MSE/MAE).

| Dataset | Horizon | MSE | MAE |
|---------|---------|-------|-------|
| ETTm1 | Avg | 0.379 | 0.398 |
| ETTm2 | Avg | 0.276 | 0.319 |
| ETTh1 | Avg | 0.440 | 0.438 |
| ETTh2 | Avg | 0.384 | 0.408 |
| Exchange | Avg | 0.339 | 0.395 |
| Weather | Avg | 0.239 | 0.265 |
| ECL | Avg | 0.161 | 0.255 |
| Traffic | Avg | 0.415 | 0.264 |

# P CAUSAL DISCOVERY VISUALIZATION

Figure 6 displays the DAGs discovered by the PC algorithm on six datasets (ETTh1, ETTh2, ETTm1, ETTm2, Exchange, and Weather). Directed edges denote compelled causal relations, whereas undirected edges mark orientational ambiguity. We use ETTh1, ETTh2, and Weather datasets as illustrating examples. Beyond the statistical visualization in Figure 6, the discovered structures also align with variable semantics. For example, in the ETT datasets we consistently observe $X_4 \to X_2 \leftarrow X_6$, where medium and low loads jointly influence high load. This reflects the known interactions among load levels in power systems, which are critical for grid stability. In the Exchange dataset, a structure $X_4 \to X_3 \leftarrow X_5$ indicates that two USD–currency exchange rates jointly affect a third one, capturing triangular relations commonly seen in foreign exchange markets. These examples demonstrate that the extracted DAGs are not only statistically reliable but also consistent with practical domain knowledge.

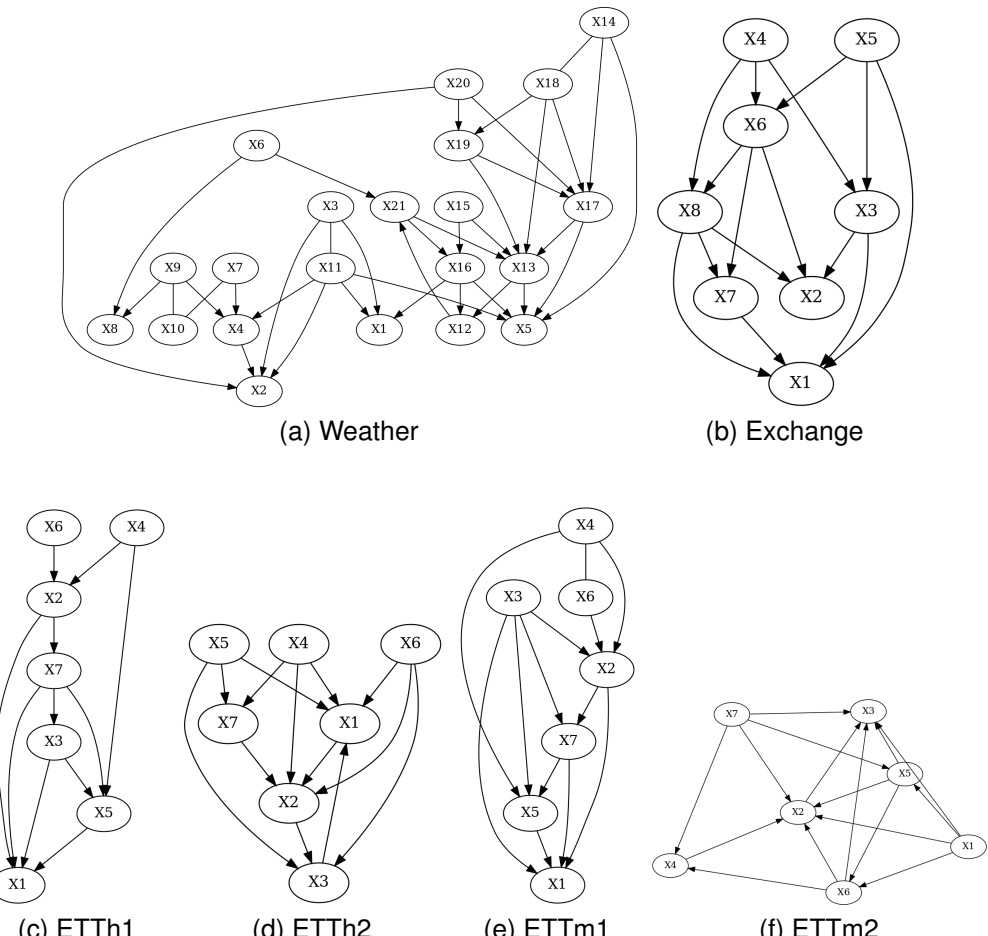

Figure 6: Visualization of causal DAGs discovered by the PC algorithm across different datasets. Directed edges indicate inferred causal relationships between variables, while undirected edges indicate uncertainty regarding causal direction. The results cover six datasets: (a) Weather,(b) Exchange, (c) ETTh1, (d) ETTh2, (e) ETTm1, and (f) ETTm2.

Building on these causal graphs, we derive the three masks required by CAIFormer ($D_{\text{mask}}$, $CS_{\text{mask}}$, and $S_{\text{mask}}$). Their heatmap visualizations in Figure 7 show clear structural patterns: some variables lack direct causal parents, while others do not have indirect auxiliary variables. This explains why, as reported in Table 2, disabling any single module causes a noticeable drop in forecasting accuracy.

To further support interpretability, we also present the masks and attention scores as Figure 8, where darker or warmer colors highlight stronger causal relevance. These visualizations make the DCS/CCS

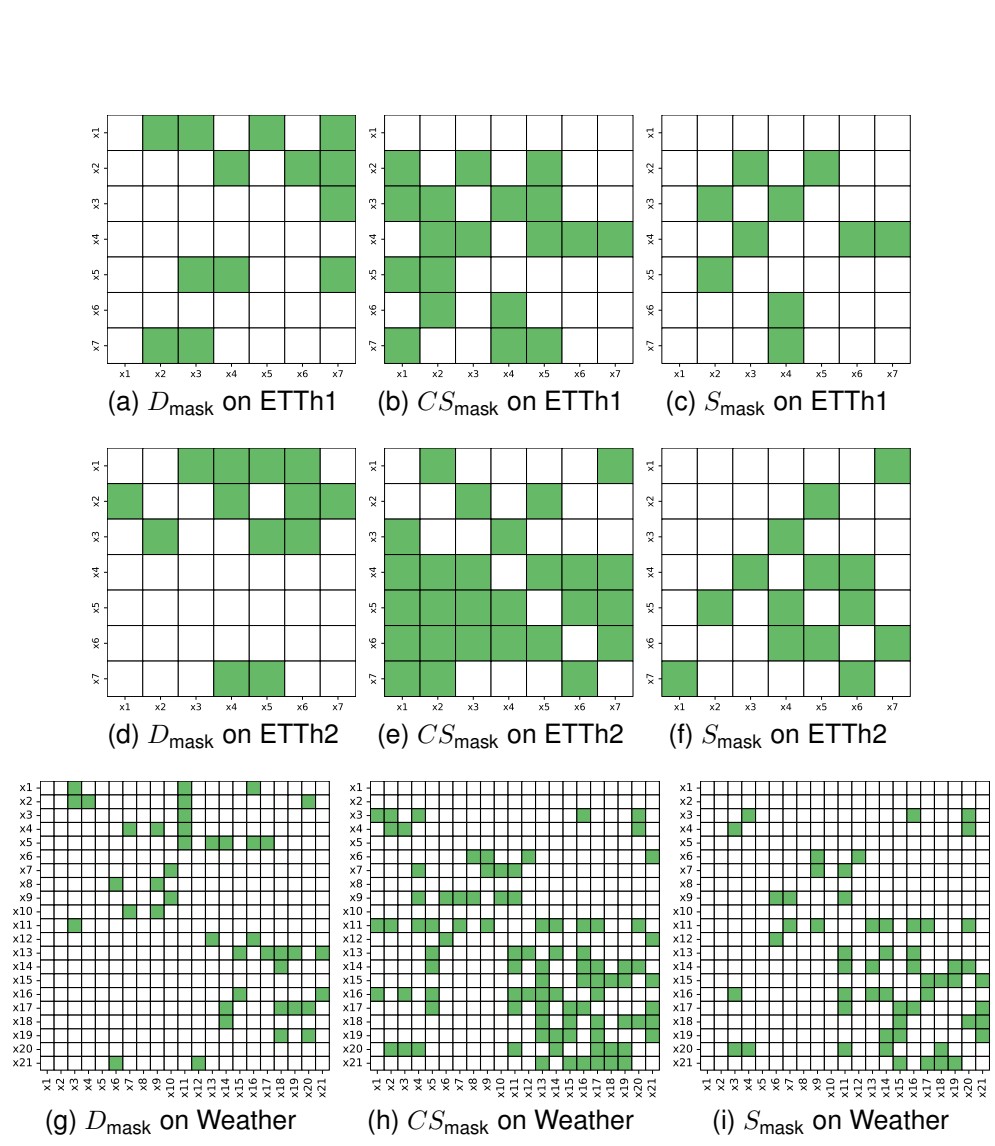

Figure 7: Visualization of the masks constructed from the DAG discovered by the PC algorithm on the ETTh1, ETTh2, and weather datasets.

separation more transparent, showing that CAIFormer assigns higher weights to variables with plausible causal impact while suppressing irrelevant ones, in line with domain knowledge.

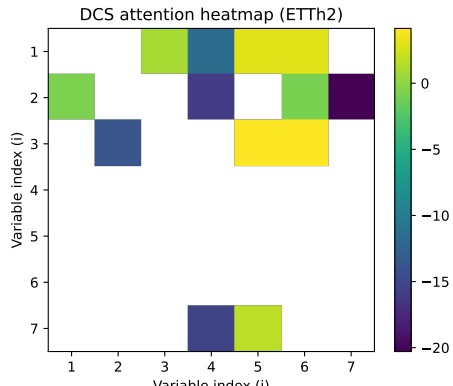
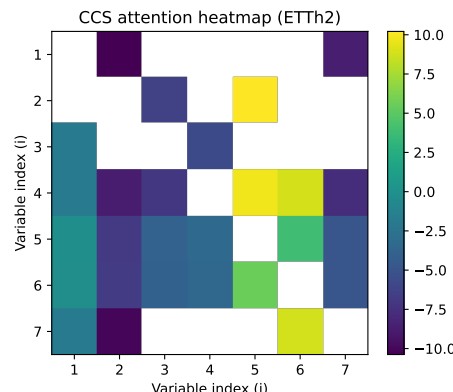

(a) DCS attention heatmap on ETTh2    (b) CCS attention heatmap on ETTh2

Figure 8: Visualization of DCS and CCS attention maps on ETTh2. Warmer colors indicate higher attention, while blank cells denote masked values.

## Q   FORECASTING RESULTS COMPARISON

Figure 9 compares CAIFormer (input window 96, prediction 96) with iTransformer and PatchTST on ETTh1, ETTh2, and Weather.

## R   DETAILED DEFINITIONS AND CAUSAL MASKS

We provide detailed definitions and the formulas for the causal relationships and masks used in the main method. Specifically, we define the sets and masks for each target variable $V_i$ based on the adjacency matrix $W_{\mathrm{adjm}} \in \{-1, 0, 1\}^{D \times D}$, as follows:

- **Direct Parents:** $\mathcal{S}_i^P = \{V_p \mid W_{\mathrm{adjm}}[p][i] = -1\}$, the set of variables that have edges pointing to $V_i$.

- **Direct Children:** $\mathcal{S}_i^K = \{V_k \mid W_{\mathrm{adjm}}[i][k] = -1, W_{\mathrm{adjm}}[s][k] \neq -1, \forall s \neq i\}$, the set of variables that are children of $V_i$ but have no other parents.

- **Colliders:** $\mathcal{S}_i^C = \{V_c \mid W_{\mathrm{adjm}}[i][c] = -1, \exists s \neq i, W_{\mathrm{adjm}}[s][c] = -1, D_{\mathrm{mask}}[i][s] \neq 1\}$, the set of variables that are colliders, receiving edges from both $V_i$ and at least one other variable.

- **Spouse Variables:** $\mathcal{S}_i^S = \{V_s \mid W_{\mathrm{adjm}}[s][c] = -1, s \neq i, D_{\mathrm{mask}}[i][s] \neq 1\}$, the set of spouse variables for each collider.

The masks for each causal relationship are defined as follows:

- **Direct Causal Mask:** $D_{\mathrm{mask}}[i][j] = 1$ if $V_j \in \mathcal{S}_i^P$ or $V_j \in \mathcal{S}_i^K$, and 0 otherwise.

- **Collider Mask:** $CS_{\mathrm{mask}}[i][j] = 1$ if $V_j \in \mathcal{S}_i^C$ or $V_j \in \mathcal{S}_i^S$, and 0 otherwise.

- **Spouse Mask:** $S_{\mathrm{mask}}[i][j] = 1$ if $V_j \in \mathcal{S}_i^S$, and 0 otherwise.

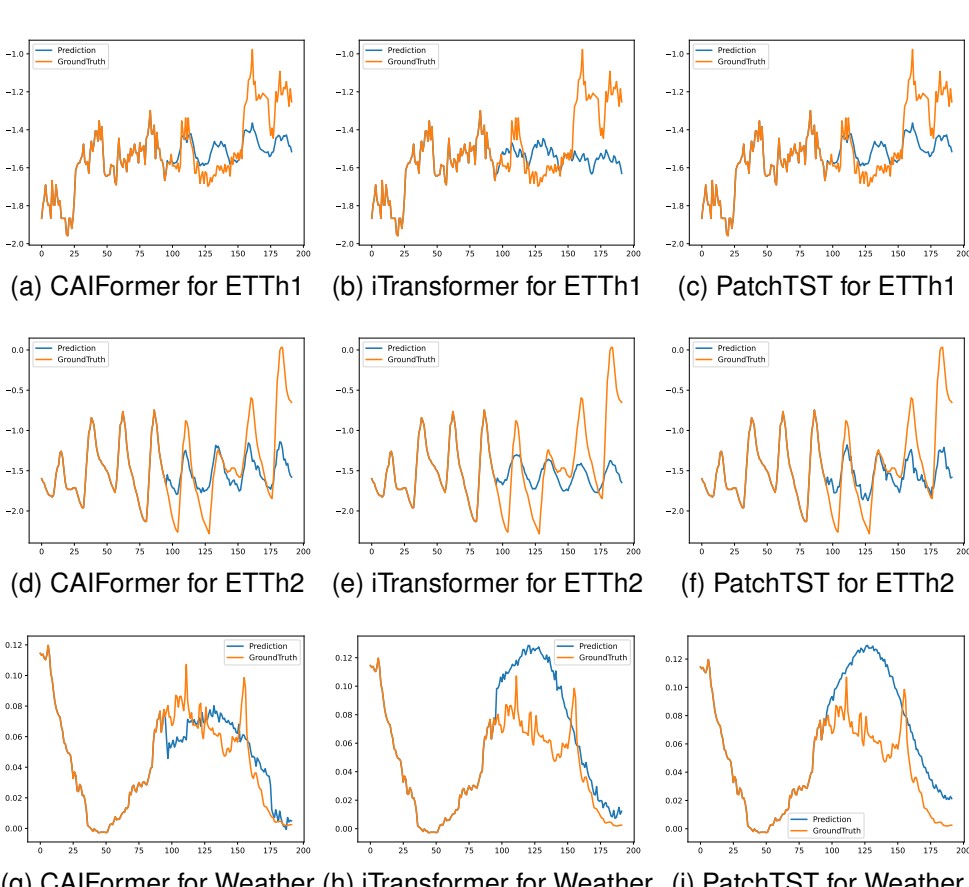

(a) CAIFormer for ETTh1    (b) iTransformer for ETTh1    (c) PatchTST for ETTh1

(d) CAIFormer for ETTh2    (e) iTransformer for ETTh2    (f) PatchTST for ETTh2

(g) CAIFormer for Weather (h) iTransformer for Weather   (i) PatchTST for Weather

Figure 9: Visualization of forecasting results for the ETTh1, ETTh2 and Weather dataset under the input-96-predict-96 setting.

Table 17: Wall-clock time of running different causal discovery algorithms on three representative datasets.

| Dataset | PC | FCI | PCMCI |
|---|---|---|---|
| ETTh1 | 10s | 8s | 12s |
| Weather | 34s | 19s | 41s |
| ECL | 35min | 3min | 22min |

## S  RUNTIME ANALYSIS OF CAUSAL DISCOVERY METHODS

To complement the end-to-end efficiency results reported in the main text, we also measure the stand-alone runtime of common causal discovery algorithms on three representative datasets. Table 17 reports the wall-clock time of running PC, FCI, and PCMCI on ETTh1, Weather, and ECL using our hardware. For small and medium-scale datasets such as ETTh1 and Weather, all methods finish within a few tens of seconds, making the preprocessing overhead negligible compared to model training. For larger and higher-dimensional datasets such as ECL, the runtime increases to several minutes, most notably for PC and PCMCI. However, this cost is incurred once as an offline preprocessing step and can be amortized over repeated training or deployment. Overall, the causal discovery overhead remains moderate in practice and does not limit the applicability of CAIFormer.

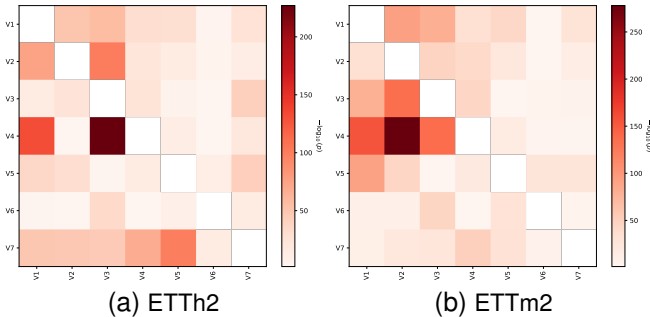

(a) ETTh2          (b) ETTm2

Figure 10: Visualization of Granger causality across variables in ETTh2, ETTm2. Each heatmap shows the transformed causal strength matrix using $-\log(P)$ values, where a darker color indicates a stronger causal influence from the row variable to the column variable. Diagonal entries are masked.

Table 18: Multivariate time series forecasting results with prediction lengths $S \in \{96, 192, 336, 720\}$ and fixed lookback length $T = 96$. The best results in **bold**, the second underlined, and "–" denote metrics not reported in the original papers. The lower MSE/MAE indicates a more accurate prediction result.

| Models | | CAIFormer (Ours) | | TimePro (2025) | | SEMPO (2025) | | TFPS (2025) | | iTransformer (2024) | | PatchTST (2023) | | RLinear (2023) | | Crossformer (2023) | | TiDE (2023) | | TimesNet (2023) | | DLinear (2023) | | FEDformer (2022) | | Autoformer (2021) | |
|---|---|---|---|---|---|---|---|---|---|---|---|---|---|---|---|---|---|---|---|---|---|---|---|---|---|---|---|
| Metric | | MSE | MAE | MSE | MAE | MSE | MAE | MSE | MAE | MSE | MAE | MSE | MAE | MSE | MAE | MSE | MAE | MSE | MAE | MSE | MAE | MSE | MAE | MSE | MAE | MSE | MAE |
| ETTm1 | 96 | 0.327 | 0.364 | 0.326 | 0.364 | 0.466 | 0.443 | 0.327 | 0.367 | 0.334 | 0.368 | 0.329 | 0.367 | 0.355 | 0.376 | 0.404 | 0.426 | 0.364 | 0.387 | 0.338 | 0.375 | 0.345 | 0.372 | 0.379 | 0.419 | 0.505 | 0.475 |
| | 192 | 0.361 | 0.377 | 0.367 | 0.383 | 0.484 | 0.455 | 0.374 | 0.395 | 0.377 | 0.391 | 0.367 | 0.385 | 0.391 | 0.392 | 0.450 | 0.451 | 0.398 | 0.404 | 0.374 | 0.387 | 0.380 | 0.389 | 0.426 | 0.441 | 0.553 | 0.496 |
| | 336 | 0.391 | 0.402 | 0.402 | 0.409 | 0.506 | 0.469 | 0.401 | 0.408 | 0.426 | 0.420 | 0.399 | 0.410 | 0.424 | 0.415 | 0.532 | 0.515 | 0.428 | 0.425 | 0.410 | 0.411 | 0.413 | 0.413 | 0.445 | 0.459 | 0.621 | 0.537 |
| | 720 | 0.449 | 0.437 | 0.469 | 0.446 | 0.557 | 0.498 | 0.479 | 0.456 | 0.491 | 0.459 | 0.454 | 0.439 | 0.487 | 0.450 | 0.666 | 0.589 | 0.487 | 0.461 | 0.478 | 0.450 | 0.474 | 0.453 | 0.543 | 0.490 | 0.671 | 0.561 |
| | Avg | 0.382 | 0.395 | 0.391 | 0.400 | 0.503 | 0.466 | 0.395 | 0.406 | 0.407 | 0.410 | 0.387 | 0.400 | 0.414 | 0.407 | 0.513 | 0.496 | 0.419 | 0.419 | 0.400 | 0.406 | 0.403 | 0.407 | 0.448 | 0.452 | 0.588 | 0.517 |
| ETTm2 | 96 | 0.168 | 0.255 | 0.178 | 0.260 | 0.196 | 0.286 | 0.170 | 0.255 | 0.180 | 0.264 | 0.175 | 0.259 | 0.182 | 0.265 | 0.287 | 0.366 | 0.207 | 0.305 | 0.187 | 0.267 | 0.193 | 0.292 | 0.203 | 0.287 | 0.255 | 0.339 |
| | 192 | 0.240 | 0.302 | 0.242 | 0.303 | 0.252 | 0.323 | 0.235 | 0.296 | 0.250 | 0.309 | 0.241 | 0.302 | 0.246 | 0.304 | 0.414 | 0.492 | 0.290 | 0.364 | 0.249 | 0.309 | 0.284 | 0.362 | 0.269 | 0.328 | 0.281 | 0.340 |
| | 336 | 0.300 | 0.339 | 0.303 | 0.342 | 0.306 | 0.354 | 0.297 | 0.335 | 0.311 | 0.348 | 0.305 | 0.343 | 0.307 | 0.342 | 0.597 | 0.542 | 0.377 | 0.422 | 0.321 | 0.351 | 0.369 | 0.427 | 0.325 | 0.366 | 0.339 | 0.372 |
| | 720 | 0.398 | 0.397 | 0.400 | 0.399 | 0.391 | 0.404 | 0.401 | 0.397 | 0.412 | 0.407 | 0.402 | 0.400 | 0.407 | 0.398 | 1.730 | 1.042 | 0.558 | 0.524 | 0.408 | 0.403 | 0.554 | 0.522 | 0.421 | 0.415 | 0.433 | 0.432 |
| | Avg | 0.276 | 0.323 | 0.281 | 0.326 | 0.286 | 0.341 | 0.276 | 0.321 | 0.288 | 0.332 | 0.281 | 0.326 | 0.286 | 0.327 | 0.757 | 0.610 | 0.358 | 0.404 | 0.291 | 0.333 | 0.350 | 0.401 | 0.305 | 0.349 | 0.327 | 0.371 |
| ETTh1 | 96 | 0.372 | 0.399 | 0.375 | 0.398 | 0.384 | 0.408 | 0.398 | 0.413 | 0.386 | 0.405 | 0.414 | 0.419 | 0.386 | 0.395 | 0.423 | 0.448 | 0.479 | 0.464 | 0.384 | 0.402 | 0.386 | 0.400 | 0.376 | 0.419 | 0.449 | 0.459 |
| | 192 | 0.429 | 0.426 | 0.427 | 0.429 | 0.409 | 0.426 | 0.423 | 0.423 | 0.441 | 0.436 | 0.460 | 0.445 | 0.437 | 0.424 | 0.471 | 0.474 | 0.525 | 0.492 | 0.436 | 0.429 | 0.437 | 0.432 | 0.420 | 0.448 | 0.500 | 0.482 |
| | 336 | 0.464 | 0.449 | 0.472 | 0.450 | 0.417 | 0.433 | 0.484 | 0.461 | 0.487 | 0.458 | 0.501 | 0.466 | 0.479 | 0.446 | 0.570 | 0.546 | 0.565 | 0.515 | 0.491 | 0.469 | 0.481 | 0.459 | 0.459 | 0.465 | 0.521 | 0.496 |
| | 720 | 0.495 | 0.483 | 0.476 | 0.474 | 0.432 | 0.454 | 0.488 | 0.476 | 0.503 | 0.491 | 0.500 | 0.488 | 0.481 | 0.470 | 0.653 | 0.621 | 0.594 | 0.558 | 0.521 | 0.500 | 0.519 | 0.516 | 0.506 | 0.507 | 0.514 | 0.512 |
| | Avg | 0.439 | 0.439 | 0.438 | 0.438 | 0.410 | 0.430 | 0.448 | 0.443 | 0.454 | 0.447 | 0.469 | 0.454 | 0.446 | 0.434 | 0.529 | 0.522 | 0.541 | 0.507 | 0.458 | 0.450 | 0.456 | 0.452 | 0.440 | 0.460 | 0.496 | 0.487 |
| ETTh2 | 96 | 0.294 | 0.344 | 0.293 | 0.345 | 0.282 | 0.342 | 0.313 | 0.355 | 0.297 | 0.349 | 0.302 | 0.348 | 0.288 | 0.338 | 0.745 | 0.584 | 0.400 | 0.440 | 0.340 | 0.374 | 0.333 | 0.387 | 0.358 | 0.397 | 0.346 | 0.388 |
| | 192 | 0.377 | 0.398 | 0.367 | 0.394 | 0.334 | 0.384 | 0.380 | 0.400 | 0.380 | 0.400 | 0.388 | 0.400 | 0.374 | 0.390 | 0.877 | 0.656 | 0.528 | 0.509 | 0.402 | 0.414 | 0.477 | 0.476 | 0.429 | 0.439 | 0.456 | 0.452 |
| | 336 | 0.425 | 0.430 | 0.419 | 0.431 | 0.355 | 0.403 | 0.392 | 0.415 | 0.428 | 0.432 | 0.426 | 0.433 | 0.415 | 0.426 | 1.043 | 0.731 | 0.643 | 0.571 | 0.452 | 0.452 | 0.594 | 0.541 | 0.496 | 0.487 | 0.482 | 0.486 |
| | 720 | 0.424 | 0.442 | 0.427 | 0.445 | 0.410 | 0.433 | 0.410 | 0.433 | 0.427 | 0.445 | 0.431 | 0.446 | 0.420 | 0.440 | 1.104 | 0.763 | 0.874 | 0.679 | 0.462 | 0.468 | 0.831 | 0.657 | 0.463 | 0.474 | 0.515 | 0.511 |
| | Avg | 0.380 | 0.403 | 0.377 | 0.403 | 0.341 | 0.391 | 0.380 | 0.403 | 0.383 | 0.407 | 0.387 | 0.407 | 0.374 | 0.398 | 0.942 | 0.684 | 0.611 | 0.550 | 0.414 | 0.427 | 0.559 | 0.515 | 0.437 | 0.449 | 0.450 | 0.459 |
| Exchange | 96 | 0.083 | 0.201 | 0.085 | 0.204 | - | - | 0.083 | 0.205 | 0.086 | 0.206 | 0.088 | 0.205 | 0.093 | 0.217 | 0.256 | 0.367 | 0.094 | 0.218 | 0.107 | 0.234 | 0.088 | 0.218 | 0.148 | 0.278 | 0.197 | 0.323 |
| | 192 | 0.173 | 0.295 | 0.178 | 0.299 | - | - | 0.174 | 0.297 | 0.177 | 0.299 | 0.176 | 0.299 | 0.184 | 0.307 | 0.470 | 0.509 | 0.184 | 0.307 | 0.226 | 0.344 | 0.176 | 0.315 | 0.271 | 0.315 | 0.300 | 0.369 |
| | 336 | 0.292 | 0.395 | 0.328 | 0.414 | - | - | 0.310 | 0.398 | 0.331 | 0.417 | 0.301 | 0.417 | 0.351 | 0.432 | 1.268 | 0.883 | 0.349 | 0.431 | 0.367 | 0.448 | 0.313 | 0.427 | 0.460 | 0.427 | 0.509 | 0.524 |
| | 720 | 0.832 | 0.688 | 0.817 | 0.679 | - | - | 1.011 | 0.756 | 0.847 | 0.691 | 0.901 | 0.714 | 0.886 | 0.714 | 1.767 | 1.068 | 0.852 | 0.698 | 0.964 | 0.746 | 0.839 | 0.695 | 1.195 | 0.695 | 1.447 | 0.941 |
| | Avg | 0.345 | 0.395 | 0.352 | 0.399 | - | - | 0.395 | 0.414 | 0.360 | 0.403 | 0.367 | 0.404 | 0.378 | 0.417 | 0.940 | 0.707 | 0.370 | 0.413 | 0.416 | 0.443 | 0.354 | 0.414 | 0.519 | 0.429 | 0.613 | 0.539 |
| Weather | 96 | 0.147 | 0.205 | 0.166 | 0.207 | 0.171 | 0.228 | 0.154 | 0.202 | 0.174 | 0.214 | 0.177 | 0.218 | 0.192 | 0.232 | 0.158 | 0.230 | 0.202 | 0.261 | 0.172 | 0.220 | 0.196 | 0.255 | 0.217 | 0.296 | 0.266 | 0.336 |
| | 192 | 0.195 | 0.243 | 0.216 | 0.254 | 0.218 | 0.269 | 0.205 | 0.249 | 0.221 | 0.254 | 0.225 | 0.259 | 0.240 | 0.271 | 0.206 | 0.277 | 0.242 | 0.298 | 0.219 | 0.261 | 0.237 | 0.296 | 0.276 | 0.336 | 0.307 | 0.367 |
| | 336 | 0.269 | 0.285 | 0.273 | 0.296 | 0.267 | 0.304 | 0.262 | 0.289 | 0.278 | 0.296 | 0.278 | 0.297 | 0.292 | 0.307 | 0.272 | 0.335 | 0.287 | 0.335 | 0.280 | 0.306 | 0.283 | 0.335 | 0.339 | 0.380 | 0.359 | 0.395 |
| | 720 | 0.345 | 0.340 | 0.351 | 0.346 | 0.336 | 0.350 | 0.344 | 0.342 | 0.358 | 0.349 | 0.354 | 0.348 | 0.364 | 0.353 | 0.398 | 0.418 | 0.351 | 0.386 | 0.365 | 0.359 | 0.345 | 0.381 | 0.403 | 0.428 | 0.419 | 0.428 |
| | Avg | 0.239 | 0.268 | 0.251 | 0.276 | 0.248 | 0.287 | 0.241 | 0.271 | 0.258 | 0.279 | 0.259 | 0.281 | 0.272 | 0.291 | 0.259 | 0.315 | 0.271 | 0.320 | 0.259 | 0.287 | 0.265 | 0.317 | 0.309 | 0.360 | 0.338 | 0.382 |
| ECL | 96 | 0.139 | 0.235 | 0.139 | 0.234 | 0.168 | 0.271 | 0.149 | 0.236 | 0.148 | 0.240 | 0.181 | 0.270 | 0.201 | 0.281 | 0.219 | 0.314 | 0.237 | 0.329 | 0.168 | 0.272 | 0.197 | 0.282 | 0.193 | 0.308 | 0.201 | 0.317 |
| | 192 | 0.155 | 0.245 | 0.156 | 0.249 | 0.183 | 0.283 | 0.162 | 0.253 | 0.162 | 0.253 | 0.188 | 0.274 | 0.201 | 0.283 | 0.231 | 0.322 | 0.236 | 0.330 | 0.184 | 0.289 | 0.196 | 0.285 | 0.201 | 0.315 | 0.222 | 0.334 |
| | 336 | 0.171 | 0.262 | 0.172 | 0.267 | 0.198 | 0.297 | 0.200 | 0.310 | 0.178 | 0.269 | 0.204 | 0.293 | 0.215 | 0.298 | 0.246 | 0.337 | 0.249 | 0.344 | 0.198 | 0.300 | 0.209 | 0.301 | 0.214 | 0.329 | 0.231 | 0.338 |
| | 720 | 0.209 | 0.303 | 0.209 | 0.299 | 0.238 | 0.329 | 0.220 | 0.320 | 0.225 | 0.317 | 0.246 | 0.324 | 0.257 | 0.331 | 0.280 | 0.363 | 0.284 | 0.373 | 0.220 | 0.320 | 0.245 | 0.333 | 0.246 | 0.355 | 0.254 | 0.361 |
| | Avg | 0.168 | 0.261 | 0.169 | 0.262 | 0.196 | 0.295 | 0.183 | 0.280 | 0.178 | 0.270 | 0.205 | 0.290 | 0.219 | 0.298 | 0.244 | 0.334 | 0.251 | 0.344 | 0.192 | 0.295 | 0.212 | 0.300 | 0.214 | 0.327 | 0.227 | 0.338 |
| Traffic | 96 | 0.388 | 0.261 | - | - | 0.441 | 0.333 | - | - | 0.395 | 0.268 | 0.462 | 0.295 | 0.649 | 0.389 | 0.522 | 0.290 | 0.805 | 0.493 | 0.593 | 0.321 | 0.650 | 0.396 | 0.587 | 0.366 | 0.613 | 0.388 |
| | 192 | 0.411 | 0.269 | - | - | 0.456 | 0.339 | - | - | 0.417 | 0.276 | 0.466 | 0.296 | 0.601 | 0.366 | 0.530 | 0.293 | 0.756 | 0.474 | 0.617 | 0.336 | 0.598 | 0.370 | 0.604 | 0.373 | 0.616 | 0.382 |
| | 336 | 0.424 | 0.276 | - | - | 0.467 | 0.344 | - | - | 0.433 | 0.283 | 0.482 | 0.304 | 0.609 | 0.369 | 0.558 | 0.305 | 0.762 | 0.477 | 0.629 | 0.336 | 0.605 | 0.373 | 0.621 | 0.337 | 0.622 | 0.337 |
| | 720 | 0.461 | 0.297 | - | - | 0.503 | 0.360 | - | - | 0.467 | 0.302 | 0.514 | 0.322 | 0.647 | 0.387 | 0.589 | 0.328 | 0.719 | 0.449 | 0.640 | 0.350 | 0.645 | 0.394 | 0.626 | 0.382 | 0.660 | 0.408 |
| | Avg | 0.421 | 0.275 | - | - | 0.466 | 0.344 | - | - | 0.428 | 0.282 | 0.481 | 0.304 | 0.626 | 0.378 | 0.550 | 0.304 | 0.760 | 0.473 | 0.620 | 0.336 | 0.625 | 0.383 | 0.610 | 0.376 | 0.628 | 0.379 |

