# OpenReview forum: "CAIFormer: A Causal Informed Transformer for Multivariate Time Series Forecasting"
_ICLR.cc/2026/Conference — Submitted to ICLR 2026_

### Official Review · Reviewer_Si3J · 2025-10-21

**Soundness:** 2
**Presentation:** 3
**Contribution:** 2
**Rating:** 4
**Confidence:** 4

**Summary:**

This paper introduces CAIFormer, a novel approach to multivariate time series forecasting that departs from the conventional all-to-all paradigm where all variables' histories are fed into a unified model. The authors propose an all-to-one forecasting strategy that predicts each target variable separately based on a causal decomposition of the input history. The method first constructs a Structural Causal Model using the PC algorithm on observational data, then partitions each target variable's historical influences into four sub-segments: endogenous (the target's own history), direct causal (variables with direct causal relationships), collider causal (variables forming collider structures), and spurious correlation (which is discarded).

**Strengths:**

The extensive experimental validation across eight diverse datasets with multiple baseline comparisons shows thoroughness, and the ablation studies effectively demonstrate the contribution of each component. The paper maintains good clarity through well-structured presentation that logically builds from motivation through theory to implementation.

**Weaknesses:**

The most critical limitation is the reliance on the PC algorithm for causal discovery, which assumes causal sufficiency (no hidden confounders) - an assumption rarely satisfied in real-world time series data. While the authors acknowledge this limitation and mention alternatives like FCI or tsFCI, they do not empirically evaluate these alternatives or discuss how violations of causal sufficiency might impact performance. The computational overhead of running causal discovery as a preprocessing step is mentioned but not thoroughly analyzed, particularly for high-dimensional datasets where causal discovery becomes computationally prohibitive.

**Questions:**

How sensitive is the method to errors in the causal discovery phase, and have the authors considered using ensemble methods or bootstrapping to obtain more robust causal structures?

The paper uses linear PC throughout but mentions nonlinear variants - could the authors provide more insight into when nonlinear causal discovery would be preferred and how it might impact the overall framework?

---

> ### Author Response · Authors · 2025-11-20
> **Response to Weakness of Reviewer Si3J**
>
> We thank the valuable comments of the reviewer Si3J.
>
> > The most critical limitation ... computationally prohibitive.
>
> Response:
> (1) We have already evaluated the stability of PC-estimated DAGs and their impact on forecasting performance (Appendix I). On ETTh1, we run PC on training set, test set, and full dataset, and compute the Jaccard similarity between the resulting DAGs; the train–test and train–full similarities are 80.9% and 90.4%, respectively (Table 10), showing that the learned structures are highly consistent across data splits. We also study how prediction changes when the graph contains errors: on the Weather dataset, we inject 0%–30% perturbations into the PC graph (removing true edges, adding spurious edges, and their mixture), retrain CAIFormer for each perturbed graph, and report the corresponding MSE/MAE. As shown in Fig. 4, performance degrades monotonically and smoothly with the perturbation rate, indicating that CAIFormer clearly benefits from the PC graph while remaining robust to moderate structural noise.
>
> (2) We further analyze whether the PC-guided causal sub-segments coincide with the signals actually used by the models (Appendix H). On the Weather dataset, we perform univariate forecasting with “OT” as target, compute variable attributions for CAIFormer and iTransformer using Integrated Gradients, and form three sets from their top-5 important variables: (a) shared, (b) CAIFormer-only, and (c) iTransformer-only. Removing set (b) leads to a large performance drop for CAIFormer and also hurts iTransformer, whereas removing set (c) has only a minor effect on CAIFormer but degrades iTransformer to a similar extent as removing set (b) (Table 6), suggesting that the variables highlighted by our causal decomposition are indeed the ones both models rely on. In addition, on ETTh1 we conduct an all-to-one ablation with a simple MLP trained either on all variables or only on the causal-related variables (ES, DCS, CCS). Using only the causal sub-segments achieves comparable or slightly better accuracy on all targets, reducing the average MSE from 0.395 to 0.392 (Table 5), which shows that the ES/DCS/CCS decomposition is helpful even in a lightweight architecture.
>
> (3) We cross-validate the PC estimated structures using multiple causal discovery methods (Appendix I.4 of Rebuttal Revision). On the ETTh1 dataset, we apply PC, FCI, and PCMCI to estimate causal graphs, and list their corresponding adjacency matrices (encoded with -1/0/1 for edge directions. Note that -1 indicates a directed edge from the row variable to the column variable, 0 indicates no edge, and 1 indicates the opposite direction.) in table below. Based on these matrices, we compute the Jaccard similarity of the edge sets and obtain 95% between PC and FCI, and 90% between PC and PCMCI. This shows that even when using FCI or PCMCI, the resulting structures remain highly consistent with the PC graph at the level of edges.
> |PC/FCI/PCMCI|1|2|3|4|5|6|7|
> |-|-|-|-|-|-|-|-|
> |1|0/0/0|-1/1/1|0/-1/1|-1/-1/-1|1/1/-1|1/-1/1|0/0/0|
> |2|1/-1/-1|0/0/0|0/-1/0|-1/-1/1|1/1/1|1/0/1|1/1/-1|
> |3|0/1/-1|0/1/0|0/0/0|-1/-1/1|1/1/1|1/1/1|-1/-1/0|
> |4|1/1/1|1/1/-1|1/1/-1|0/0/0|1/0/0|1/1/0|1/0/1|
> |5|-1/-1/1|-1/-1/-1|-1/-1/-1|-1/0/0|0/0/0|0/-1/1|-1/-1/-1|
> |6|-1/1/-1|-1/0/-1|-1/-1/-1|-1/-1/0|0/-1/-1|0/0/0|0/0/-1|
> |7|0/0/0|-1/-1/1|1/1/0|-1/0/-1|1/1/1|0/0/1|0/0/0|
>
> We further replace the PC-based DAG in CAIFormer with the FCI-based and PCMCI-based DAGs, and run forecasting experiments on ETTh1 with an input length of 96. As reported in table below, the MSE/MAE of the PC-based, FCI-based, and PCMCI-based variants differ by less than 0.01 across all four horizons (96/192/336/720). Overall, these results indicate that, under the same model architecture, switching between different mainstream causal discovery algorithms has almost no effect on the forecasting performance of CAIFormer.
> |ETTh1|PC-based||FCI-based||PCMCI-based||
> |-|-|-|-|-|-|-|
> |Metric|MSE|MAE|MSE|MAE|MSE|MAE|
> |96|0.372|0.399|0.374|0.399|0.372|0.400|
> |192|0.429|0.426|0.428|0.425|0.430|0.426|
> |336|0.464|0.449|0.464|0.450|0.463|0.447|
> |720|0.495|0.483|0.496|0.485|0.494|0.483|
>
> (4) Finally, we provide a more explicit analysis of the runtime of different causal discovery methods (Appendix S of Rebuttal Revision). The table below lists the wall-clock time of running PC, FCI, and PCMCI on three representative datasets on our hardware:
> ||PC|FCI|PCMCI|
> |-|-|-|-|
> |ETTh1|10s|8s|12s|
> |Weather|34s|19s|41s|
> |ECL|35min|3min|22min|
>
> For small to medium-scale datasets such as ETTh1 and Weather, the causal discovery step is on the order of seconds, and its overhead is negligible compared to model training. For higher-dimensional data such as ECL, the runtime increases to tens of minutes, but this cost is incurred once as an offline preprocessing step and can be amortized over repeated training and deployment.
>
> [1] Hasan, Uzma, et al. "A Survey on Causal Discovery Methods for I.I.D. and Time Series Data." TMLR, 2023.

---

> ### Author Response · Authors · 2025-11-20
> **Response to Questions of Reviewer Si3J**
>
> We thank the valuable comments of the reviewer Si3J.
>
> > How sensitive is the method to errors in the causal discovery phase, and have the authors considered using ensemble methods or bootstrapping to obtain more robust causal structures?
>
> Response: As the reviewer suggests, we also experiment with ensemble and bootstrapping schemes to obtain potentially more robust DAGs (Appendix I.6 of Rebuttal Revision). On the ETTh1 dataset, we consider two designs: Ensemble method: we run PC, FCI, and PCMCI on the full training set, and construct an ensemble DAG by majority voting over the three edge sets; Bootstrapping method: we perform 10 bootstrap resamples of the training set (each subsample contains 10% of the training data with replacement), run PC on each subsample to obtain 10 graphs, and then take a majority vote over these 10 graphs, retaining edges that appear in at least 6 runs. The adjacency matrices of the three graphs (single PC, ensemble, and bootstrap) are shown below (entries -1/0/1 denote a directed edge from row to column, no edge, and the opposite direction, respectively):
> | PC/Ensemble /Bootstrap | 1       | 2       | 3        | 4        | 5       | 6      | 7       |
> | ---------------------- | ------- | ------- | -------- | -------- | ------- | ------ | ------- |
> | 1                  | 0/0/0   | -1/1/0  | 0/0/1    | -1/-1/-1 | 1/1/-1  | 1/1/-1 | 0/0/0   |
> | 2                 | 1/-1/0  | 0/0/0   | 0/0/0    | -1/-1/1  | 1/1/0   | 1/1/0  | 1/1/-1  |
> | 3                  | 0/0/-1  | 0/0/0   | 0/0/0    | -1/-1/1  | 1/1/1   | 1/1/0  | -1/-1/0 |
> | 4               | 1/1/1   | 1/1/-1  | 1/1/-1   | 0/0/0    | 1/0/0   | 1/1/0  | 1/1/1   |
> | 5                  | -1/-1/1 | -1/-1/0 | -1/-1/-1 | -1/0/0   | 0/0/0   | 0/0/1  | -1/-1/0 |
> | 6                | -1/-1/1 | -1/-1/0 | -1/-1/0  | -1/-1/0  | 0/-1/-1 | 0/0/0  | 0/0/-1  |
> | 7                  | 0/0/0   | -1/-1/1 | 1/1/0    | -1/-1/-1 | 1/1/0   | 0/0/1  | 0/0/0   |
>
> We then replace the original PC-based graph in CAIFormer with the ensemble graph and the bootstrap graph, respectively, and run multistep forecasting experiments on ETTh1 with input length 96. The results as below:
> | ETTh1  | PC-based |       | Ensemble -based |       | Bootstrap-based |       |
> | ------ | -------- | ----- | --------------- | ----- | --------------- | ----- |
> | Metric | MSE      | MAE   | MSE             | MAE   | MSE             | MAE   |
> | 96     | 0.372    | 0.399 | 0.382           | 0.405 | 0.402           | 0.421 |
> | 192    | 0.429    | 0.426 | 0.437           | 0.434 | 0.465           | 0.459 |
> | 336    | 0.464    | 0.449 | 0.480           | 0.461 | 0.510           | 0.492 |
> | 720    | 0.495    | 0.483 | 0.511           | 0.493 | 0.559           | 0.531 |
>
> We observe that, compared with the single PC DAG, the ensemble DAG leads to a slight degradation in forecasting performance, and the bootstrap-based DAG causes a more pronounced degradation across all horizons. Our interpretation is that bootstrapping substantially reduces the effective sample size for each individual run (10% of the training set), which introduces additional statistical noise into each estimated graph. For this reason, we keep the single PC graph as the structural prior in the main experiments, and use perturbation and ensemble/bootstrap analyses to characterize how uncertainties in causal discovery affect CAIFormer’s performance.
>
> > The paper uses linear PC throughout but mentions nonlinear variants - could the authors provide more insight into when nonlinear causal discovery would be preferred and how it might impact the overall framework?
>
> Response: Building on Table 16, we additionally report nonlinear-PC results on the high-dimensional ECL and Traffic datasets. Overall, on the six low- to medium-dimensional datasets, the linear and nonlinear PC variants yield almost identical forecasting performance, with average differences within 0.01, indicating that linear conditional independence tests are already sufficient in these settings. On the high-dimensional ECL and Traffic datasets, however, the nonlinear-PC variant achieves slightly but consistently better MSE/MAE, suggesting that nonlinear conditional independence testing can be more beneficial when the number of variables is large and the dependency structure is more complex. Taken together, these findings justify our choice of linear-PC as the default configuration in the main experiments due to its lower computational cost and adequate accuracy, while viewing the nonlinear variant as an optional extension for high-dimensional scenarios.

---

> ### Author Response · Authors · 2025-11-25
> **Analysis from the perspective of textbook definitions**
>
> Our use of the PC algorithm is exactly aligned with the causal graphical framework in [1]. The key concepts are:
>
> - Markov compatibility (Definition 1.2.2 of [1], the indices discussed in this section are all from [1]): a distribution $P$ is compatible with a DAG $G$ if it factorizes according to the parents in $G$.
> - d-separation (Definition 1.2.3) and observational equivalence (Theorem 1.2.8): d-separation in $G$ is sound and complete for the conditional independencies in $P$; two DAGs have the same observational implications if they share the same skeleton and v-structures.
> - Minimality (Section 2.3, “minimal potential structure”): a causal graph is minimal if no edge can be removed without contradicting the independencies in $P$.
> - Stability/faithfulness (Section 2.4): only independencies that persist under small parameter perturbations are treated as structural; “accidental” independencies from fine-tuned parameters are excluded.
> - Causal effect definition (Definition 2.3.6): $C$ is said to have a causal effect on $E$ only if every minimal causal structure consistent with the observed distribution contains a directed path from $C$ to $E$.
>
> The PC algorithm is the standard, constraint-based realization of exactly these notions: it uses conditional independencies to recover a minimal, faithful, Markov-compatible Markov equivalence class, represented as a CPDAG. Within Pearl’s framework, this equivalence class is precisely the set of causal structures that can be identified from observational data; no causal discovery method can legitimately claim more without additional assumptions (e.g., interventions).
>
> **To Weakness (causal sufficiency / hidden confounders):**
>
> In [1], causal sufficiency (no unmeasured confounders among the modeled variables) is explicitly taken as the standard setting for DAG-based discovery (Secs. 1.2, 2.3): under sufficiency + Markov + faithfulness, there is a well-defined mapping from independencies to a DAG, for which constraint-based methods such as PC are sound and complete. When sufficiency is violated, the book moves to more general mixed graphs (PAGs) and algorithms like FCI, and stresses that this is a fundamental identifiability limit of observational data with hidden variables, not a defect of any particular DAG learner. Our use of PC therefore follows exactly the “sufficient, fully observed” regime in [1]; in the presence of latent confounding, any method operating only on observed time series inherits the same theoretical limitation.
>
> **To Question (sensitivity/robustness of causal structures):**
>
> Definition 2.3.6 in [1] makes causal claims deliberately conservative: a relation $C \to E$ is accepted only if it appears in every minimal structure compatible with the observed, stable independencies, i.e., as the intersection over the entire Markov equivalence class. In Pearl’s framework, robustness to errors in the discovery phase is thus encoded at the level of this equivalence class: structural uncertainty that does not change the underlying independence model cannot change which causal effects are warranted. Practical ensemble or bootstrap procedures can be viewed simply as empirical ways to approximate this intersection over minimal graphs; the core robustness notion itself is already built into the graphical causal theory.
>
> [1] Pearl J. *Causality*. 2nd ed. Cambridge University Press; 2009.

---

> > ### Comment · Reviewer_Si3J · 2025-11-26
> >
> > Thank you for your detailed response, which has addressed most of my concerns. I will discuss with the other reviewers to determine the final rating.

---

> > > ### Author Response · Authors · 2025-11-27
> > >
> > > Thank you for your prompt reply. If there are any additional questions or clarifications needed, we would be happy to provide them.

---

### Official Review · Reviewer_qzzU · 2025-10-31

**Soundness:** 3
**Presentation:** 2
**Contribution:** 2
**Rating:** 4
**Confidence:** 3

**Summary:**

The authors propose a Causally Informed Transformer (CAIFormer) for multivariate time series forecasting, which reformulates the traditional all-to-all paradigm into an all-to-one forecasting framework.

Specifically, for each target variable, the model first constructs a Structural Causal Model (SCM) from observational data, and then partitions the historical sequence into four sub-segments: endogenous, direct causal, collider causal, and spurious correlation.Only the first three causally relevant sub-segments are retained for prediction.

Each sub-segment is processed through a dedicated self-attention block to capture variable-specific dependencies, and their outputs are finally concatenated and fused to generate the final forecast.

**Strengths:**

1. The paper clearly describes the target task and provides a corresponding rationale for the proposed approach. The overall problem setup is easy to follow, the motivation for moving from the all-to-all to the all-to-one forecasting paradigm is intuitive.

2. The proposed method is supported by a theoretical foundation. The authors present formal definitions and operator-based formulations to justify their causal decomposition framework, which enhances the soundness of the approach.

**Weaknesses:**

1. The paper’s presentation lacks clarity in several aspects.

a) The discussion of causal path types (a–f) is overly detailed and fragmented — intuitively, the relationships between variables could be summarized more compactly, making the current exposition unnecessarily complex.

b) The description of the Transformer architecture is vague: although matrix dimensions are provided, it remains unclear how the time-series data in each sub-segment are organized into the sequential format required by the Transformer. This ambiguity makes it difficult for readers to fully grasp the algorithmic flow.

2. The paper’s causal discovery procedure is conceptually questionable.

The authors state that “To avoid trivial autoregressive effects, conditional independence tests are only applied across variables at the same time index.” However, in time-series settings, considering only instantaneous dependencies is insufficient and incomplete, as temporal lags often encode the essential causal dynamics. Ignoring cross-time effects undermines the validity of the discovered causal structure and its utility for forecasting.

3. The authors propose leveraging causal relationships to improve forecasting performance. However, the paper does not clearly articulate the connection between causal reasoning and predictive performance. From the perspective of a Structural Causal Model (SCM), the causal generative mechanism of a variable can be fully determined by its parent nodes alone. it is unclear why identifying and removing so-called “spurious correlations”is necessary.If the goal is purely predictive performance, the conclusion from https://arxiv.org/abs/2402.09891 (NIPS2024) [1] suggests that incorporating all variables can effectively improve forecasting accuracy, which contradicts the fundamental premise of this paper.

4. The PC algorithm adopted in the first stage is order-dependent, meaning that early errors in conditional independence testing can accumulate and propagate through the causal discovery process, potentially leading to numerous incorrect or spurious causal relationships. However, the authors do not provide empirical evidence showing whether the learned causal graphs approximate the ground truth, nor do they discuss the robustness of the algorithm with respect to noise, sample size, or ordering effects.

[1] Nastl, Vivian, and Moritz Hardt. "Do causal predictors generalize better to new domains?." Advances in Neural Information Processing Systems 37 (2024): 31202-31315.

**Questions:**

1. Why do the authors only consider variables at the same time index? If this is the case, are the lagged variables in the time series ignored ? If these lagged variables are not properly masked, does that mean the resulting causal graph is inconsistent with the true temporal causal structure?

2. The paper introduces causal reasoning into forecasting but still includes collider variables in the model. From a causal standpoint, the generation of the target T should depend only on its parents; including colliders and then removing their influence via projection seems unnecessary and conceptually inconsistent. If the goal is prediction rather than causal discovery, this also contradicts recent findings (e.g., NeurIPS 2024) https://arxiv.org/abs/2402.09891[1] showing that using all features—regardless of causality—often yields the best performance. Why, then, restrict the predictor to only selected causal sub-segments?

3. Could the authors provide empirical evidence or further clarification on whether the causal graphs learned by the PC algorithm approximate the ground-truth structures, and how the overall forecasting performance is affected when the discovered graphs contain errors or spurious relations?

4. The ablation results show that the ESPB-only variant performs almost as well as the full model. Does this imply that other variables contribute little effective information to the prediction? If relying mainly on the autoregressive features of a single variable already yields near-optimal performance, might the claimed benefits of the proposed causal decomposition be overstated?

[1] Nastl, Vivian, and Moritz Hardt. "Do causal predictors generalize better to new domains?." Advances in Neural Information Processing Systems 37 (2024): 31202-31315.

---

> ### Author Response · Authors · 2025-11-20
> **Response to Weakness.1 of Reviewer qzzU**
>
> We thank the valuable comments of the reviewer qzzU.
>
> > The paper’s presentation lacks clarity in several aspects.
> >
> >a) The discussion of causal path types (a–f) is overly detailed and fragmented — intuitively, the relationships between variables could be summarized more compactly, making the current exposition unnecessarily complex.
>
> Response: In the original version, we listed all six path types (a–f) in detail in the main text, which indeed made the exposition long and fragmented. In the rebuttal revision, we have compressed the main text into a high-level summary and moved the case-by-case enumeration and formal proofs to the appendix. Concretely, the main text now reads:
>
> "Based on the causal analysis in Appendix D, the relationship between a non-target variable and the target $V_i$ in Fig.1d can be treat separately: First, $V_1$ can be an ancestor of $V_i$, including the direct-parent case and more general upstream structures such as chains and forks (e.g., $V_2 \to V_1 \to V_i$ or $V_2 \gets V_1 \to V_i$, corresponding to Paths (a)–(c)). In all these cases, $V_1$ lies on a directed path into $V_i$, and conditioning on all remaining variables $\mathcal{Z} = \{V_1,\dots,V_{i-1},V_{i+1},\dots,V_D\}$ leaves $V_1$ dependent on $V_i$, while any other $V_j \in \mathcal{Z}\setminus\{V_1\}$ becomes conditionally independent of $V_i$. Second, $V_1$ can be a descendant of $V_i$, including direct children and any downstream chain (corresponding to Paths (d)–(e)). This situation is symmetric: $V_1$ lies on a directed path out of $V_i$, and after conditioning on $\mathcal{Z}$, only $V_1$ remains conditionally dependent on $V_i$, whereas all other $V_j \in \mathcal{Z}\setminus\{V_1\}$ are independent of $V_i$. Finally, $V_1$ can act as a collider on a path from $V_i$ to another variable $V_2$ (Path (f)), e.g., $V_i \to V_1 \gets V_2 - \cdots - V_D$. In this case, conditioning on $\mathcal{Z}$ makes both $V_1$ and its spouse $V_2$ dependent on $V_i$, while any $V_j \in \mathcal{Z}\setminus\{V_1,V_2\}$ is conditionally independent of $V_i$. Formal proofs of the conditional independencies asserted for each path are provided in Appendix E."
>
> >b) The description of the Transformer architecture is vague: although matrix dimensions are provided, it remains unclear how the time-series data in each sub-segment are organized into the sequential format required by the Transformer. This ambiguity makes it difficult for readers to fully grasp the algorithmic flow.
>
> Response: We appreciate the reviewer’s comment that the organization of time-series data into Transformer-style sequences was not sufficiently clear. In the rebuttal reversion (Lines 391-395 and Lines 404-410), we explicitly describe how each sub-segment is arranged as a sequence for the Transformer modules in CAIFormer. Given a historical input sequence $X \in \mathbb{R}^{T \times D}$ (T time steps, D variables), the data are organized as follows:
>
> "In ESPB, we use only the history of the target variable $V_i$ (the ES segment). Along the time dimension, we split $V_i^{0:T}$ into $H$ non-overlapping patches of length $P$, obtaining $H$ time-ordered patch tokens. Each patch is then projected by a linear layer into a $d_E$-dimensional vector, forming a sequence of shape $(H, d_E)$ as the input to the Transformer. The self-attention operates over these $H$ patches, i.e., it models the dependencies between different time patches, and each variable is processed independently in ESPB."
>
> "In DCSPB and CCSPB, we instead adopt a variable sequence view. For each variable $V_j$, its full history $V_j^{0:T}$ is encoded by a variable-wise temporal embedding into a vector of dimension $d_D$ (or $d_C$), yielding $X^0_{\text{Enc}} \in \mathbb{R}^{D \times d_D}$. Here, the variable index $j$ serves as the sequence axis for the Transformer. In DCSPB, multi-head attention is applied along this length-$D$ sequence, and the mask $D_{\text{mask}}$ restricts attention to variables that have a direct causal relation with the target $V_i$. In CCSPB, we reuse the same variable sequence input but apply the mask $CS_{\text{mask}}$, so that attention is focused on collider- and spouse-related variables."

---

> ### Author Response · Authors · 2025-11-20
> **Response to Weakness.2 and Questions.1 of Reviewer qzzU**
>
> We thank the valuable comments of the reviewer qzzU.
>
> > Weakness.2: The paper’s causal discovery procedure is conceptually questionable.
> >
> > The authors state that “To avoid trivial autoregressive effects, conditional independence tests are only applied across variables at the same time index.” However, in time-series settings, considering only instantaneous dependencies is insufficient and incomplete, as temporal lags often encode the essential causal dynamics. Ignoring cross-time effects undermines the validity of the discovered causal structure and its utility for forecasting.
>
> > Questions.1: Why do the authors only consider variables at the same time index? If this is the case, are the lagged variables in the time series ignored? If these lagged variables are not properly masked, does that mean the resulting causal graph is inconsistent with the true temporal causal structure?
>
> Response: (1) In our implementation, the causal discovery algorithm is applied to the entire historical sequence $X \in \mathbb{R}^{T \times D}$, rather than to a single time slice. In other words, the algorithm takes the full multivariate time series as input and learns which variables provide non-redundant information about others, given their own histories, reflecting overall cross-variable influence patterns over time, rather than merely instantaneous correlations at one specific time point. The original sentence “conditional independence tests are only applied across variables at the same time index” was not precise enough and may suggest that temporal lags are completely ignored. In the rebuttal revision, we rephrase this description to avoid such a misunderstanding.
>
> (2) In CAIFormer, temporal autoregressive and lag effects are mainly handled by the neural network itself, e.g., via the patch-based temporal representations in ESPB and the temporal encodings inside each sub-module. The DAG learned from the historical data is then used as a structural prior in the variable dimension to decide which variables’ histories should influence the target and how attention is constrained in the DCS/CCS blocks. In this sense, the DAG provides a structural pattern over variables, while the temporal modules capture the dynamics along the time axis; they are designed to be complementary rather than one replacing the other.
>
> (3) In addition, we have cross-validated this design using PCMCI (details in Response to Weakness.4 and Questions.3 of Reviewer qzzU), a causal discovery method that explicitly models lagged (cross-time) dependencies. On the ETTh1, ETTm1, and Exchange datasets, we ran PCMCI to estimate causal graphs and compared them with those obtained by PC. We found that the edge sets of PCMCI and PC have Jaccard similarities greater than 90% on all three datasets. We then replaced the original PC-based graphs with the PCMCI-based graphs in CAIFormer and retrained the model; the resulting changes in MSE and MAE are smaller than 0.01, and the overall forecasting performance remained almost unchanged. This indicates that even when using a causal discovery method that explicitly accounts for cross-time effects, the structural prior fed into CAIFormer is largely consistent with that of PC, and the predictive performance of CAIFormer is essentially preserved.
>
> (4) Finally, the interpretability and robustness analyses in Appendices I and J further support the effectiveness of this design: the DAGs learned from different data splits exhibit high consistency (with Jaccard similarities of 80.9% and 90.4%), and in our DAG perturbation experiments, randomly deleting or adding edges leads to a smooth degradation of forecasting performance rather than a sudden collapse. Taken together, these results show that using a structure learned from the full historical sequence as a variable-level prior, in combination with temporal modeling modules, does not undermine the validity of the discovered structure for forecasting.

---

> ### Author Response · Authors · 2025-11-20
> **Response to Weakness.3 and Questions.2 of Reviewer qzzU**
>
> We thank the valuable comments of the reviewer qzzU.
>
> > Weakness.3: The authors propose ... which contradicts the fundamental premise of this paper.
>
> > Questions.2: The paper introduces ... only selected causal sub-segments?
>
> Response: (1) In multivariate time series forecasting, the future of a target variable is driven not only by its own temporal dynamics but also by heterogeneous influences from other variables. Existing all-to-all models usually feed the full history of all variables into a single shared predictor and treat every variable–target pair symmetrically. From an SCM perspective, this ignores that different variables play different roles with respect to the target: some are direct causal drivers, some become conditionally related through collider patterns, and others are only correlated via shared causes and become irrelevant once those causes are accounted for. As illustrated by the weather example in our introduction (temperature, wind, pressure, precipitation), indiscriminately mixing all these histories conflates true causal signals with spurious correlations. CAIFormer makes this structural heterogeneity explicit: for each target, we use the learned DAG to decompose its historical window into four sub-segments and train a predictor that retains the target’s own history (ES), variables that are directly causal or structurally close (DCS), and collider/spouse-related histories (CCS), while filtering out only SCS variables that are neither ancestors nor descendants of the target nor part of any collider/spouse structure. In the observational distribution, descendants and collider/spouse variables are not independent of the target and often carry strong predictive signals, so this design aims to preserve structurally supported information while removing variables whose correlations are more likely to be noisy or unstable.
>
> (2) We further analyze whether the PC-guided causal sub-segments coincide with the signals actually used by the models (Appendix H). On the Weather dataset, we perform univariate forecasting with “OT” as target, compute variable attributions for CAIFormer and iTransformer using Integrated Gradients, and form three sets from their top-5 important variables: (a) shared, (b) CAIFormer-only, and (c) iTransformer-only. Removing set (b) leads to a large performance drop for CAIFormer and also hurts iTransformer, whereas removing set (c) has only a minor effect on CAIFormer but degrades iTransformer to a similar extent as removing set (b) (Table 6), suggesting that the variables highlighted by our causal decomposition are indeed the ones both models rely on. In addition, on ETTh1 we conduct an all-to-one ablation with a simple MLP trained either on all variables or only on the causal-related variables (ES, DCS, CCS). Using only the causal sub-segments achieves comparable or slightly better accuracy on all targets, reducing the average MSE from 0.395 to 0.392 (Table 5), which shows that the ES/DCS/CCS decomposition is helpful even in a lightweight architecture.
>
> (3) Compared to [1], our setting is substantially different. [1] focuses on static tabular data and domain generalization, comparing “causal predictors” against using all available features. In contrast, our work targets multivariate time series forecasting, uses a PC-estimated DAG to segment variables into ES/DCS/CCS/SCS, and relies on block-wise sub-segment modeling. The question in our case is therefore not simply “causal features vs. all features,” but how to organize information according to the learned structure.
>
> (4) Regarding the reviewer’s concern that “introducing colliders and then removing their influence via projection” might be unnecessary or conceptually inconsistent, we would like to clarify that CCS in CAIFormer is not simply “added and then erased.” In the CCS sub-segment we allow the model to exploit the predictive information carried along collider paths, i.e., variables that are highly correlated with the target but are not its direct parents. The projection step in Section 3.5 only removes the component of this representation that violates the desired independence structure, so that the final representation respects the conditional independencies implied by the causal graph. In this sense, including colliders and then applying a projection is a two-step design, first leveraging observational correlations, then restoring the intended independence structure in the representation space, rather than a contradiction.
>
> [1] Nastl, Vivian, and Moritz Hardt. "Do causal predictors generalize better to new domains?." Advances in Neural Information Processing Systems 37 (2024): 31202-31315.

---

> ### Author Response · Authors · 2025-11-20
> **Response to Weakness.4 and Questions.3 of Reviewer qzzU**
>
> We thank the valuable comments of the reviewer qzzU.
>
> > Weakness.4: The PC algorithm adopted in the first stage is order-dependent, meaning that early errors in conditional independence testing can accumulate and propagate through the causal discovery process, potentially leading to numerous incorrect or spurious causal relationships. However, the authors do not provide empirical evidence showing whether the learned causal graphs approximate the ground truth, nor do they discuss the robustness of the algorithm with respect to noise, sample size, or ordering effects.
>
> > Questions.3: Could the authors provide empirical evidence or further clarification on whether the causal graphs learned by the PC algorithm approximate the ground-truth structures, and how the overall forecasting performance is affected when the discovered graphs contain errors or spurious relations?
>
> Response: (1) In the paper we have already evaluated the stability of the PC-estimated graphs and their impact on forecasting performance (Appendix I). On ETTh1, we run PC on the training set, test set, and full dataset, and compute the Jaccard similarity between the resulting DAGs; the train–test and train–full similarities are 80.9% and 90.4%, respectively (Table 10), showing that the learned structures are highly consistent across data splits. We also study how prediction changes when the graph contains errors: on the Weather dataset, we inject 0%–30% perturbations into the PC graph (removing true edges, adding spurious edges, and their mixture), retrain CAIFormer for each perturbed graph, and report the corresponding MSE/MAE. As shown in Fig. 4, performance degrades monotonically and smoothly with the perturbation rate, indicating that CAIFormer clearly benefits from the PC graph while remaining robust to moderate structural noise.
>
> (2) We further analyze whether the PC-guided causal sub-segments coincide with the signals actually used by the models (Appendix H). On the Weather dataset, we perform univariate forecasting with “OT” as target, compute variable attributions for CAIFormer and iTransformer using Integrated Gradients, and form three sets from their top-5 important variables: (a) shared, (b) CAIFormer-only, and (c) iTransformer-only. Removing set (b) leads to a large performance drop for CAIFormer and also hurts iTransformer, whereas removing set (c) has only a minor effect on CAIFormer but degrades iTransformer to a similar extent as removing set (b) (Table 6), suggesting that the variables highlighted by our causal decomposition are indeed the ones both models rely on. In addition, on ETTh1 we conduct an all-to-one ablation with a simple MLP trained either on all variables or only on the causal-related variables (ES, DCS, CCS). Using only the causal sub-segments achieves comparable or slightly better accuracy on all targets, reducing the average MSE from 0.395 to 0.392 (Table 5), which shows that the ES/DCS/CCS decomposition is helpful even in a lightweight architecture.
>
> (3) We cross-validate the PC estimated structures using multiple causal discovery methods (Appendix I.4 of Rebuttal Revision). On the ETTh1 dataset, we apply PC, FCI, and PCMCI to estimate causal graphs, and list their corresponding adjacency matrices (encoded with -1/0/1 for edge directions. Note that -1 indicates a directed edge from the row variable to the column variable, 0 indicates no edge, and 1 indicates the opposite direction.) in table below. Based on these matrices, we compute the Jaccard similarity of the edge sets and obtain 95% between PC and FCI, and 90% between PC and PCMCI. This shows that even when using FCI or PCMCI, the resulting structures remain highly consistent with the PC graph at the level of edges.
> |PC/FCI/PCMCI|1|2|3|4|5|6|7|
> |-|-|-|-|-|-|-|-|
> |1|0/0/0|-1/1/1|0/-1/1|-1/-1/-1|1/1/-1|1/-1/1|0/0/0|
> |2|1/-1/-1|0/0/0|0/-1/0|-1/-1/1|1/1/1|1/0/1|1/1/-1|
> |3|0/1/-1|0/1/0|0/0/0|-1/-1/1|1/1/1|1/1/1|-1/-1/0|
> |4|1/1/1|1/1/-1|1/1/-1|0/0/0|1/0/0|1/1/0|1/0/1|
> |5|-1/-1/1|-1/-1/-1|-1/-1/-1|-1/0/0|0/0/0|0/-1/1|-1/-1/-1|
> |6|-1/1/-1|-1/0/-1|-1/-1/-1|-1/-1/0|0/-1/-1|0/0/0|0/0/-1|
> |7|0/0/0|-1/-1/1|1/1/0|-1/0/-1|1/1/1|0/0/1|0/0/0|
>
> We further replace the PC-based DAG in CAIFormer with the FCI-based and PCMCI-based DAGs, and run forecasting experiments on ETTh1 with an input length of 96. As reported in table below, the MSE/MAE of the PC-based, FCI-based, and PCMCI-based variants differ by less than 0.01 across all four horizons (96/192/336/720). Overall, these results indicate that, under the same model architecture, switching between different mainstream causal discovery algorithms has almost no effect on the forecasting performance of CAIFormer.
> |ETTh1|PC-based||FCI-based||PCMCI-based||
> |-|-|-|-|-|-|-|
> |Metric|MSE|MAE|MSE|MAE|MSE|MAE|
> |96|0.372|0.399|0.374|0.399|0.372|0.400|
> |192|0.429|0.426|0.428|0.425|0.430|0.426|
> |336|0.464|0.449|0.464|0.450|0.463|0.447|
> |720|0.495|0.483|0.496|0.485|0.494|0.483|

---

> ### Author Response · Authors · 2025-11-20
> **Response to Questions.4 of Reviewer qzzU**
>
> We thank the valuable comments of the reviewer qzzU.
>
> > The ablation results show that the ESPB-only variant performs almost as well as the full model. Does this imply that other variables contribute little effective information to the prediction? If relying mainly on the autoregressive features of a single variable already yields near-optimal performance, might the claimed benefits of the proposed causal decomposition be overstated?
>
> Response: We would like to first clarify that the numbers reported below are exactly the same as those in Table 2 of the original manuscript. In the revision, we simply isolate the two relevant rows—“full CAIFormer” and “Only ESPB”—and explicitly report the relative improvements, so that the difference between them is easier to see quantitatively. The comparison is as follows (all values are unchanged from the original table; we only add the “Improve” row):
> |           | Weather |       | ETTh1 |       | Exchange |       |
> | --------- | ------- | ----- | ----- | ----- | -------- | ----- |
> |           | MSE     | MAE   | MSE   | MAE   | MSE      | MAE   |
> | CAIFormer | 0.239   | 0.268 | 0.439 | 0.439 | 0.345    | 0.395 |
> | Only ESPB | 0.259   | 0.281 | 0.469 | 0.454 | 0.367    | 0.404 |
> | Imporve   | 7.7%    | 4.6%  | 6.3%  | 3.3%  | 6.0%     | 2.2%  |
>
> As shown, across all three datasets, the full model consistently improves over the ESPB-only variant in both MSE and MAE, with relative MSE gains of about 6–8% and MAE gains of about 2–5%. Presenting the relative improvements makes it clearer that incorporating additional historical information from other variables via DCS/CCS still brings meaningful gains in forecasting accuracy.
>
> Furthermore, CAIFormer's causal decomposition provides structural interpretability that cannot be provided by using the ESPB model alone: the ESPB model alone cannot provide hard data on other variables, while the full model clearly distinguishes the contributions of autoregressive, direct causality, and collisional causality.

---

> ### Author Response · Authors · 2025-11-25
> **Analysis from the perspective of textbook definitions**
>
> Our use of the PC algorithm is exactly aligned with the causal graphical framework in [1]. The key concepts are:
>
> - Markov compatibility (Definition 1.2.2 of [1], the indices discussed in this section are all from [1]): a distribution $P$ is compatible with a DAG $G$ if it factorizes according to the parents in $G$.
> - d-separation (Definition 1.2.3) and observational equivalence (Theorem 1.2.8): d-separation in $G$ is sound and complete for the conditional independencies in $P$; two DAGs have the same observational implications if they share the same skeleton and v-structures.
> - Minimality (Section 2.3, “minimal potential structure”): a causal graph is minimal if no edge can be removed without contradicting the independencies in $P$.
> - Stability/faithfulness (Section 2.4): only independencies that persist under small parameter perturbations are treated as structural; “accidental” independencies from fine-tuned parameters are excluded.
> - Causal effect definition (Definition 2.3.6): $C$ is said to have a causal effect on $E$ only if every minimal causal structure consistent with the observed distribution contains a directed path from $C$ to $E$.
>
> The PC algorithm is the standard, constraint-based realization of exactly these notions: it uses conditional independencies to recover a minimal, faithful, Markov-compatible Markov equivalence class, represented as a CPDAG. Within Pearl’s framework, this equivalence class is precisely the set of causal structures that can be identified from observational data; no causal discovery method can legitimately claim more without additional assumptions (e.g., interventions).
>
> **To Weakness.4 (order-dependence / error propagation):**
>
> In [1], the target of causal discovery is the independence model $I(P)$ of the true distribution. Given $I(P)$, the induced Markov equivalence class of DAGs is uniquely determined by d-separation (Def. 1.2.3 and Thm. 1.2.8), and the corresponding minimal structures are characterized in Sec. 2.3. Under the Markov and faithfulness assumptions, the PC algorithm is sound and complete for recovering this equivalence class from exact independence information. Thus, at the level of causal semantics, the result is uniquely determined and does not depend on variable ordering; order-dependence arises only from finite-sample estimation of conditional independencies (a generic statistical issue), not from the underlying graphical causal theory.
>
> **To Questions.3 (approximation to ground truth and effect of errors):**
>
> In [1], from purely observational data the “ground-truth” causal structure is identifiable only up to its Markov equivalence class; approximating the true graph therefore means approximating this class rather than a single DAG. Under the Markov and faithfulness assumptions, PC is designed precisely for this purpose: it recovers the set of minimal DAGs whose d-separation relations match the (stable) independencies of $P$ (Secs. 2.3–2.4). Structural differences that do not change the induced independence model are observationally indistinguishable and thus irrelevant for causal identification in the sense of [1]; only errors that alter the independence pattern correspond to genuinely different causal hypotheses, and their impact on forecasting is a general issue of model misspecification rather than a limitation of the causal semantics.
>
> [1] Pearl J. *Causality*. 2nd ed. Cambridge University Press; 2009.

---

### Official Review · Reviewer_QtHD · 2025-11-01

**Soundness:** 3
**Presentation:** 2
**Contribution:** 3
**Rating:** 4
**Confidence:** 5

**Summary:**

This paper proposes a novel “one-versus-all” prediction model called CAIFormer, which predicts each target variable individually. Specifically, CAIFormer first constructs a structural causal model based on observational data. For each target variable, it then divides the historical sequence into four sub-segments according to the inferred causal structure: endogeneity, direct causality, collision causality, and spurious correlation. Predictions rely solely on the first three causally relevant sub-segments, while the spurious correlation segment is excluded. CAIFormer achieves state-of-the-art performance across eight datasets.

**Strengths:**

1. State-of-the-art performance. As shown in Table 1, CAIFormer achieves state-of-the-art performance across eight datasets.

2. Solid theoretical foundation. The author provides extensive theoretical proofs in the methodology and appendices, enhancing the reliability of the approach.

3. Clear Motivation. The paper argues that the non-partitioned model makes it difficult to identify the causal effects of specific variables and often confuses causally relevant information with spurious correlations. To address this limitation, the paper proposes a “one-versus-all” prediction model, which separately predicts each target variable. The motivation is clear.

**Weaknesses:**

1. Lack of relevant prior work. The authors claim in the abstract and introduction that previous work has overlooked the differing causal effects various variables may exert relative to the target. However, a similar approach (i.e., multiple lags) has already been proposed in TimePro[1]. The authors should include a discussion of TimePro and explicitly state the differences between their method and TimePro. Furthermore, Table 1 needs to demonstrate that CAIFormer outperforms TimePro.

[1] TimePro: Efficient Multivariate Long-term Time Series Forecasting with Variable- and Time-Aware Hyper-state

2. Unsatisfactory readability. For lines 195-213, the author should add appropriate paragraph breaks. The Avg column in Table 1 should be removed. What does “condition e” mean in Figure 2? The author should add a note.

3. Many formulas are unnecessary. For example, Line 327-344. The introduction of numerous letters or formulas makes the Methods section overly redundant. As a result, the author provides less content on the Related works and lacks clarity in describing the background of the methods.

**Questions:**

Please refer to the weaknesses.

---

> ### Author Response · Authors · 2025-11-20
> **Response to Weakness.1 of Reviewer QtHD**
>
> We thank the valuable comments of the reviewer QtHD.
>
> > Lack of relevant prior work. The authors claim in the abstract and introduction that previous work has overlooked the differing causal effects various variables may exert relative to the target. However, a similar approach (i.e., multiple lags) has already been proposed in TimePro[1]. The authors should include a discussion of TimePro and explicitly state the differences between their method and TimePro. Furthermore, Table 1 needs to demonstrate that CAIFormer outperforms TimePro.
>
> Response: We thank the reviewer for pointing out the missing discussion of TimePro [1]. This is an important and closely related recent method, and we have updated the manuscript accordingly.
>
> (1) In the original version, Section 2 reviewed major forecasting paradigms, and Appendix C discussed works on variable relationships and causal discovery. Following the reviewer’s suggestion, we have now added a discussion of TimePro in Section 2.2 (see lines 116-119 of rebuttal revision), and placed it within the same category as other variable-based methods. The added sentence reads: "Similarly, TimePro [1] introduces a multi-lag approach, incorporating both time- and variable-aware hyper-state embeddings to capture complex, dynamic inter-variable dependencies."
>
> Similar to the variable-based methods already discussed in the main text, TimePro performs rich pairwise modeling of cross-variable and multi-lag dependencies, but it does not explicitly distinguish different types of inter-variable relations. In this sense, it still treats all variable pairs in a largely unconstrained way and may mix true causal effects with spurious correlations induced by confounding or collider paths, which is precisely the limitation we discuss in our introduction.
>
> (2) We have also added TimePro as a baseline in Table 1 of the main text and Table 18 in the appendix (an additional column for TimePro). Under the same setting (input length 96, prediction length {96,192,336,720}) used by TimePro, the average results on the 7 datasets are as follows (**bold** indicates the best result in each column):
> | Datasets | CAIFormer-MSE | Ours      | TimePro   | ICML25    |
> | -------- | ------------- | --------- | --------- | --------- |
> | ETTm1    | **0.382**     | **0.395** | 0.391     | 0.400     |
> | ETTm2    | **0.276**     | **0.323** | 0.281     | 0.326     |
> | ETTh1    | 0.439         | 0.439     | **0.438** | **0.438** |
> | ETTh2    | 0.380         | **0.403** | **0.377** | **0.403** |
> | Exchange | **0.345**     | **0.395** | 0.352     | 0.399     |
> | Weather  | **0.239**     | **0.268** | 0.251     | 0.276     |
> | ECL      | **0.168**     | **0.261** | 0.169     | 0.262     |
>
> We observe that CAIFormer outperforms TimePro on most datasets.
>
> [1] TimePro: Efficient Multivariate Long-term Time Series Forecasting with Variable- and Time-Aware Hyper-state

---

> ### Author Response · Authors · 2025-11-20
> **Response to Weakness.2 of Reviewer QtHD**
>
> We thank the valuable comments of the reviewer QtHD.
>
> > Unsatisfactory readability. For lines 195-213, the author should add appropriate paragraph breaks. The Avg column in Table 1 should be removed. What does “condition e” mean in Figure 2? The author should add a note.
>
> Response: We have revised the manuscript to improve clarity and presentation as follows:
>
> (1) For lines 195–213, the original text formed a single long paragraph. We have slightly streamlined the wording and split it into three shorter paragraphs to improve readability. The revised passage now reads:
>
> "Based on the causal analysis in Appendix D, the relationship between a non-target variable and the target $V_i$ in Fig. 1(d) can be treated separately as follows.
>
> First, $V_1$ can be an ancestor of $V_i$, including the direct-parent case and more general upstream structures such as chains and forks (e.g., $V_2 \to V_1 \to V_i$ or $V_2 \gets V_1 \to V_i$, corresponding to Paths (a)–(c)). In all these cases, $V_1$ lies on a directed path into $V_i$, and conditioning on all remaining variables $\mathcal{Z} = \{V_1,\dots,V_{i-1},V_{i+1},\dots,V_D\}$ leaves $V_1$ dependent on $V_i$, while any other $V_j \in \mathcal{Z}\setminus\{V_1\}$ becomes conditionally independent of $V_i$.
>
> Second, $V_1$ can be a descendant of $V_i$, including direct children and any downstream chain (corresponding to Paths (d)–(e)). This situation is symmetric: $V_1$ lies on a directed path out of $V_i$, and after conditioning on $\mathcal{Z}$, only $V_1$ remains conditionally dependent on $V_i$, whereas all other $V_j \in \mathcal{Z}\setminus\{V_1\}$ are independent of $V_i$.
>
> Finally, $V_1$ can act as a collider on a path from $V_i$ to another variable $V_2$ (Path (f)), e.g., $V_i \to V_1 \gets V_2 - \cdots - V_D$. In this case, conditioning on $\mathcal{Z}$ makes both $V_1$ and its spouse $V_2$ dependent on $V_i$, while any $V_j \in \mathcal{Z}\setminus\{V_1,V_2\}$ is conditionally independent of $V_i$. Formal proofs of the conditional independencies asserted for each path are provided in Appendix E."
>
> (2) Following the reviewer’s suggestion, we have removed the redundant “Avg” column from Table 1 to avoid repetition and make the table more concise.
>
> (3) Regarding “condition e” in Fig. 2, we have added an explicit note in the caption to clarify its meaning. The revised caption now includes: "In CCSPB, Condition E refers to the collider constraint." And collider constraint is detailed in lines 420-426 of rebuttal revision.

---

> ### Author Response · Authors · 2025-11-20
> **Response to Weakness.3 of Reviewer QtHD**
>
> We thank the valuable comments of the reviewer QtHD.
>
> > Many formulas are unnecessary. For example, Line 327-344. The introduction of numerous letters or formulas makes the Methods section overly redundant. As a result, the author provides less content on the Related works and lacks clarity in describing the background of the methods.
>
> Response: We have revised the manuscript to reduce redundancy in the Methods section and to make the related work more accessible, as follows:
>
> (1) For lines 327–344, where many symbols and formulas were originally introduced, we have simplified the main text and moved the full formal definitions to an appendix. The revised paragraph in the main Methods section now reads:
>
> "In our method, we assume causal sufficiency, meaning there are no latent confounders beyond the observed variables. The algorithm generates an adjacency matrix $W_{\text{adjm}} \in \{-1, 0, 1\}^{D \times D}$, where each element $W_{\text{adjm}}[i][j] = -1$ indicates a directed edge $V_i \to V_j$, and $W_{\text{adjm}}[i][j] = W_{\text{adjm}}[j][i] = -1$ represents an undirected edge. For each target variable $V_i$, we define its causal relationships by extracting sets from the adjacency matrix. Specifically, we identify Direct Parents $\mathcal{S}^P_i$, Direct Children $\mathcal{S}^K_i$, and Colliders $\mathcal{S}^C_i$, based on the structure of the matrix. The corresponding masks $D_{\text{mask}}$, $CS_{\mathrm{mask}}$, and $S_{\mathrm{mask}}$ are then created to capture the causal influences, where the mask values are 1 if the relationship is present and 0 otherwise. Detailed set definitions and their corresponding formulas can be found in Appendix R."
>
> In this way, the main text contains only the intuitive description needed to follow the method, while the full set-theoretic and symbolic definitions are moved to Appendix R, avoiding an overload of letters and formulas in the core Methods section.
>
> (2) Regarding the concern that the emphasis on formulas left less room for related work and background, we have also reorganized the placement of existing material. Specifically, we have moved part of the discussion that was previously in Appendix C of original revision into Section 2 of the main text. This discussion focuses on variable-relationship modeling and causal discovery, including methods such as TimePro, Timexer, and Causal-TSF. Only more detailed technical remarks are now kept in the appendix. This relocation brings the relevant related work and background closer to the main narrative of the method, so that readers can more easily understand how CAIFormer relates to and differs from prior approaches, while keeping the Methods section itself lighter in notation.

---

### Official Review · Reviewer_PUAg · 2025-11-01

**Soundness:** 3
**Presentation:** 3
**Contribution:** 3
**Rating:** 6
**Confidence:** 3

**Summary:**

This paper proposes CAIFormer, a causal-informed transformer framework for multivariate time-series forecasting (MTSF). Based on a causal discovery step using the Peter-Clark (PC) algorithm, the method decomposes the information flow into Direct Causal Segments (DCS) and Collider Causal Segments (CCS). The authors further provide a theoretical analysis showing that an orthogonal projection operator $\Psi$ can remove spurious dependencies arising from collider structures, thereby improving generalization. Empirically, CAIFormer is evaluated on several standard MTSF benchmarks.

**Strengths:**

1. The paper's main strength lies in its novel reframing of the MTSF problem from a causal perspective. Moving from a monolithic "all-to-all" approach to a causally informed "all-to-one" paradigm is a significant and well-motivated conceptual contribution that could inspire future research in this area.

2. The CAIFormer architecture is not arbitrary; its design is directly motivated by the proposed causal decomposition. The use of separate modules for different causal roles (ES, DCS, CCS) is elegant and provides a clear path to improved model interpretability.

**Weaknesses:**

1. The theoretical derivation in Section 3.5 suffers from an inconsistency and relies on an overly strong independence assumption. Specifically, it critically assumes that the target variable $V_i$ and its spouse variable $V_s$ are strictly independent, a condition that rarely holds in practice.

2. The experimental scope is limited. The comparison includes only baselines published up to 2023. Given the rapid progress in MTSF, evidenced by numerous recent works from 2024 and 2025, it remains unclear whether CAIFormer can still achieve competitive performance against the latest state-of-the-art methods. Without such comparisons, the empirical evidence supporting the claimed advantages remains incomplete.

3. The paper provides insufficient analysis of the reliability and interpretability of the causal discovery step. The entire framework depends on the DAG learned by the PC algorithm, which is sensitive to noise and relies on the faithfulness assumption. However, the authors present no quantitative evaluation of the learned graph’s stability or correctness, nor do they discuss how errors in causal discovery affect the downstream segmentation (DCS and CCS) and forecasting performance.

**Questions:**

1. The Granger-causality heatmaps in Fig. 1 clearly show that almost all variable pairs exhibit non-zero causal influence, implying that the variables are highly interdependent. However, the theoretical derivation in Section 3.5 (Eqs. (3)–(7)) explicitly assumes that the target variable $V_i$ and its spouse $V_s$ are independent. This raises a fundamental concern: if such independence never holds empirically—as suggested by Fig. 1—does this assumption invalidate or substantially weaken the correctness and rigor of all subsequent theoretical results (e.g., the definition of $\mathcal{F}_\Psi$, the projection property, and Theorem 3.1)? Could the authors clarify whether the claimed generalization guarantee still holds when variables are only weakly dependent, and how the proposed framework behaves under the dense causal graphs observed in Fig. 1?

2. Figures 1 and 4 appear to convey different causal densities: the Granger causality heatmaps show dense interactions, whereas the DAGs discovered by the PC algorithm are sparse and partly disconnected.

3. Why are the latest state-of-the-art (SOTA) models not included in the benchmark comparison? Do the authors expect CAIFormer to maintain its advantage against these stronger baselines?

4. Which representation (Granger vs. PC) should readers regard as the underlying causal structure assumed by CAIFormer?

5. How robust is CAIFormer to errors in the discovered causal graph? For instance, what happens to performance if a certain percentage of edges are randomly perturbed—for example, by missing true causal links or introducing spurious ones? This would directly test the fragility of the entire framework.

6. Could the authors provide a discussion comparing the “hard” discarding of the SCS block to a “soft” alternative? For example, one could still feed the SCS variables into a separate block but apply strong regularization (e.g., a large L1 penalty on its output weights) or use an attention mechanism that strongly suppresses attention to this block. This would clarify whether complete removal is truly necessary or whether a less drastic approach would suffice.

---

> ### Author Response · Authors · 2025-11-20
> **Response to Weakness.1 of Reviewer PUAg**
>
> We thank the valuable comments of the reviewer PUAg.
>
> > The theoretical derivation in Section 3.5 suffers from an inconsistency and relies on an overly strong independence assumption. Specifically, it critically assumes that the target variable $V_i$ and its spouse variable $V_s$ are strictly independent, a condition that rarely holds in practice.
>
> Response: (1) Under the standard semantics of causal graphs, in a collider structure $V_i \to V_c \gets V_s$, if there is no other path between $V_i$ and $V_s$ and we do not condition on $V_c$ or its descendants, then $V_i$ and $V_s$ are independent; this can be formalized via d-separation [1]. The derivation in Section 3.5 is carried out under exactly this kind of local, idealized pattern: we assume that the only connection between the target $V_i$ and its spouse $V_s$ goes through the collider $V_c$, and therefore treat $V_i$ and $V_s$ as independent in the derivation. This is a standard local assumption in causal graph analysis, rather than a claim that arbitrary pairs of variables are strictly independent in the entire graph.
>
> (2) In our implementation, we do not impose this assumption on all pairs of variables. We first estimate a causal graph using the PC algorithm, and then apply the result of Section 3.5 only in local regions that match the collider pattern. Concretely, in line 1611 of rebuttal revision, we constrain the spouse variable $V_s$ so that it is neither a parent nor a child of $V_i$; when constructing the spouse set, we also exclude variables that already belong to the parent/child sets. This removes cases where there is a direct edge or an obvious additional path between $V_i$ and $V_s$, and restricts the independence assumption to local configurations consistent with the collider structure, rather than assuming strict independence everywhere in the graph.
>
> (3) In Figs. 7(c), 7(f), and 7(i), the green blocks indicate that the PC algorithm identifies column variables as spouses of row variables. In our construction, these green blocks correspond exactly to the spouse variables $V_s$ used in the derivation in Section 3.5, i.e., the variables that are independent of the target $V_i$. The figure shows that such spouse nodes appear frequently across datasets and targets, so the local derivation based on the independence between $V_i$ and $V_s$ is applicable to many regions, rather than relying on a very rare or artificial configuration.
>
> [1] Pearl, Judea. *Causality*. Cambridge University Press, 2009.

---

> ### Author Response · Authors · 2025-11-20
> **Response to Weakness.2 and Questions.3 of Reviewer PUAg**
>
> We thank the valuable comments of the reviewer PUAg.
>
> > Weakness.2: The experimental scope is limited. The comparison includes only baselines published up to 2023. Given the rapid progress in MTSF, evidenced by numerous recent works from 2024 and 2025, it remains unclear whether CAIFormer can still achieve competitive performance against the latest state-of-the-art methods. Without such comparisons, the empirical evidence supporting the claimed advantages remains incomplete.
>
> > Questions.3: Why are the latest state-of-the-art (SOTA) models not included in the benchmark comparison? Do the authors expect CAIFormer to maintain its advantage against these stronger baselines?
>
> Response: (1) On the representativeness of existing baselines. The baselines reported in Table 1, such as PatchTST and iTransformer [2], are still among the most commonly used and representative forecasting models in recent MTSF papers from 2024–2025. In the original manuscript, we also included targeted comparisons with newer methods that are closely related to our motivations: since CAIFormer is causality-inspired, we compare it with the recent causal time-series forecasting method Causal-TSF [3] in Table 13; and since CAIFormer explicitly models inter-variable relations, we compare it with Timexer [4], which focuses on exogenous variables and cross-variable dependencies, in Table 14. These original results show that CAIFormer remains superior or at least competitive in those settings.
>
> (2) On comparisons with the latest SOTA models, to further address the reviewer’s concern about performance relative to the most recent methods from 2024–2025, we additionally report comparisons on the same 8 datasets as in Table 1 between CAIFormer and several new multivariate forecasting models, namely TimePro [5], SEMPO [6], and TFPS [7]. The results are summarized below (**bold** indicates the best result in each column, “–” denotes metrics not reported in the original papers, and each value is the average performance for input length 96 and output horizons {96, 192, 336, 720}).
> | Models   | CAIFormer | Ours      | TimePro   | ICML25    | SEMPO     | NIPS25    | TFPS      | NIPS25    |
> | -------- | --------- | --------- | --------- | --------- | --------- | --------- | --------- | --------- |
> | dataset  | MSE       | MAE       | MSE       | MAE       | MSE       | MAE       | MSE       | MAE       |
> | ETTm1    | **0.382** | **0.395** | 0.391     | 0.400     | 0.503     | 0.466     | 0.395     | 0.406     |
> | ETTm2    | **0.276** | 0.323     | 0.281     | 0.326     | 0.286     | 0.341     | **0.276** | **0.321** |
> | ETTh1    | 0.439     | 0.439     | 0.438     | 0.438     | **0.410** | **0.430** | 0.448     | 0.443     |
> | ETTh2    | 0.380     | 0.403 | 0.377 | 0.403 | **0.341**     | **0.391**     | 0.380     | 0.403 |
> | Exchange | **0.345** | **0.395** | 0.352     | 0.399     | -         | -         | 0.395     | 0.414     |
> | Weather  | **0.239** | **0.268** | 0.251     | 0.276     | 0.248     | 0.287     | 0.241     | 0.271     |
> | ECL      | **0.168** | **0.261** | 0.169     | 0.262     | 0.196     | 0.295     | 0.183     | 0.280     |
> | Traffic  | **0.421** | **0.275** | -         | -         | 0.466     | 0.344     | -         | -         |
>
> We observe that CAIFormer achieves the best or highly competitive performance on most datasets, with particularly strong gains on high-dimensional datasets such as ECL and Traffic. This suggests that leveraging the causal graph to organize variable sub-segments helps CAIFormer better model inter-variable relations.
>
> [2]iTransformer: Inverted Transformers Are Effective for Time Series Forecasting. ICLR 2024.
>
> [3]Causal-TSF: A Causal Intervention Approach to Mitigate Confounding Bias in Time Series Forecasting. TKDE 2025.
>
> [4]Timexer: Empowering transformers for time series forecasting with exogenous variables. NeurIPS 2024.
>
> [5]TimePro: Efficient Multivariate Long-term Time Series Forecasting with Variable- and Time-Aware Hyper-state. ICML 2025
>
> [6]SEMPO: Lightweight Foundation Models for Time Series Forecasting. NeurIPS 2025.
>
> [7]Learning Pattern-Specific Experts for Time Series Forecasting Under Patch-level Distribution Shift. NeurIPS 2025.

---

> ### Author Response · Authors · 2025-11-20
> **Response to Weakness.3 and Questions.5 of Reviewer PUAg**
>
> We thank the valuable comments of the reviewer PUAg.
>
> > Weakness.3: The paper provides insufficient analysis of the reliability and interpretability of the causal discovery step. The entire framework depends on the DAG learned by the PC algorithm, which is sensitive to noise and relies on the faithfulness assumption. However, the authors present no quantitative evaluation of the learned graph’s stability or correctness, nor do they discuss how errors in causal discovery affect the downstream segmentation (DCS and CCS) and forecasting performance.
>
> > Questions.5: How robust is CAIFormer to errors in the discovered causal graph? For instance, what happens to performance if a certain percentage of edges are randomly perturbed—for example, by missing true causal links or introducing spurious ones? This would directly test the fragility of the entire framework.
>
> Response: (1) In the paper we have already evaluated the stability of the PC-estimated graphs and their impact on forecasting performance (Appendix I). On ETTh1, we run PC on the training set, test set, and full dataset, and compute the Jaccard similarity between the resulting DAGs; the train–test and train–full similarities are 80.9% and 90.4%, respectively (Table 10), showing that the learned structures are highly consistent across data splits. We also study how prediction changes when the graph contains errors: on the Weather dataset, we inject 0%–30% perturbations into the PC graph (removing true edges, adding spurious edges, and their mixture), retrain CAIFormer for each perturbed graph, and report the corresponding MSE/MAE. As shown in Fig. 4, performance degrades monotonically and smoothly with the perturbation rate, indicating that CAIFormer clearly benefits from the PC graph while remaining robust to moderate structural noise.
>
> (2) We further analyze whether the PC-guided causal sub-segments coincide with the signals actually used by the models (Appendix H). On the Weather dataset, we perform univariate forecasting with “OT” as target, compute variable attributions for CAIFormer and iTransformer using Integrated Gradients, and form three sets from their top-5 important variables: (a) shared, (b) CAIFormer-only, and (c) iTransformer-only. Removing set (b) leads to a large performance drop for CAIFormer and also hurts iTransformer, whereas removing set (c) has only a minor effect on CAIFormer but degrades iTransformer to a similar extent as removing set (b) (Table 6), suggesting that the variables highlighted by our causal decomposition are indeed the ones both models rely on. In addition, on ETTh1 we conduct an all-to-one ablation with a simple MLP trained either on all variables or only on the causal-related variables (ES, DCS, CCS). Using only the causal sub-segments achieves comparable or slightly better accuracy on all targets, reducing the average MSE from 0.395 to 0.392 (Table 5), which shows that the ES/DCS/CCS decomposition is helpful even in a lightweight architecture.
>
> (3) We cross-validate the PC estimated structures using multiple causal discovery methods. On the ETTh1 dataset, we apply PC, FCI, and PCMCI to estimate causal graphs, and list their corresponding adjacency matrices (encoded with -1/0/1 for edge directions. Note that -1 indicates a directed edge from the row variable to the column variable, 0 indicates no edge, and 1 indicates the opposite direction.) in table below. Based on these matrices, we compute the Jaccard similarity of the edge sets and obtain 95% between PC and FCI, and 90% between PC and PCMCI. This shows that even when using FCI or PCMCI, the resulting structures remain highly consistent with the PC graph at the level of edges.
> |PC/FCI/PCMCI|1|2|3|4|5|6|7|
> |-|-|-|-|-|-|-|-|
> |1|0/0/0|-1/1/1|0/-1/1|-1/-1/-1|1/1/-1|1/-1/1|0/0/0|
> |2|1/-1/-1|0/0/0|0/-1/0|-1/-1/1|1/1/1|1/0/1|1/1/-1|
> |3|0/1/-1|0/1/0|0/0/0|-1/-1/1|1/1/1|1/1/1|-1/-1/0|
> |4|1/1/1|1/1/-1|1/1/-1|0/0/0|1/0/0|1/1/0|1/0/1|
> |5|-1/-1/1|-1/-1/-1|-1/-1/-1|-1/0/0|0/0/0|0/-1/1|-1/-1/-1|
> |6|-1/1/-1|-1/0/-1|-1/-1/-1|-1/-1/0|0/-1/-1|0/0/0|0/0/-1|
> |7|0/0/0|-1/-1/1|1/1/0|-1/0/-1|1/1/1|0/0/1|0/0/0|
>
> We further replace the PC-based DAG in CAIFormer with the FCI-based and PCMCI-based DAGs, and run forecasting experiments on ETTh1 with an input length of 96. As reported in table below, the MSE/MAE of the PC-based, FCI-based, and PCMCI-based variants differ by less than 0.01 across all four horizons (96/192/336/720). Overall, these results indicate that, under the same model architecture, switching between different mainstream causal discovery algorithms has almost no effect on the forecasting performance of CAIFormer.
> |ETTh1|PC-based||FCI-based||PCMCI-based||
> |-|-|-|-|-|-|-|
> |Metric|MSE|MAE|MSE|MAE|MSE|MAE|
> |96|0.372|0.399|0.374|0.399|0.372|0.400|
> |192|0.429|0.426|0.428|0.425|0.430|0.426|
> |336|0.464|0.449|0.464|0.450|0.463|0.447|
> |720|0.495|0.483|0.496|0.485|0.494|0.483|

---

> ### Author Response · Authors · 2025-11-20
> **Response to Questions.1 of Reviewer PUAg**
>
> We thank the valuable comments of the reviewer PUAg.
>
> > The Granger-causality heatmaps in Fig. 1 clearly show that almost all variable pairs exhibit non-zero causal influence, implying that the variables are highly interdependent. However, the theoretical derivation in Section 3.5 (Eqs. (3)–(7)) explicitly assumes that the target variable $V_i$ and its spouse $V_s$ are independent. This raises a fundamental concern: if such independence never holds empirically—as suggested by Fig. 1—does this assumption invalidate or substantially weaken the correctness and rigor of all subsequent theoretical results (e.g., the definition of $\mathcal{F}_\Psi$, the projection property, and Theorem 3.1)? Could the authors clarify whether the claimed generalization guarantee still holds when variables are only weakly dependent, and how the proposed framework behaves under the dense causal graphs observed in Fig. 1?
>
> Response: (1) First, the Granger-causality heatmaps in Fig. 1 and the DAGs used in our theory describe two fundamentally different notions of dependence, so the fact that almost all pairs show non-zero Granger influence does not directly conflict with the local independence assumption between the target $V_i$ and its spouse $V_s$ used in Section 3.5. Granger causality measures whether adding the past of one variable improves the prediction of another, and thus captures predictive associations rather than direct structural edges. This naturally leads to very dense patterns. For example, in a simple chain $V_1 \to V_2 \to V_3$, the structural graph has only two edges, but Granger analysis typically concludes that $V_1$ also influences $V_3$, because information from $V_1$ is transmitted to $V_3$ through $V_2$. Therefore, the “almost fully non-zero” pattern in Fig. 1 mainly reflects that past histories can help predict each other, rather than implying that all variable pairs are structurally dependent or that certain local independencies cannot hold.
>
> (2) In contrast, our theoretical analysis is based on a structural graph estimated by the PC algorithm. PC uses conditional independence tests to remove dependencies induced by indirect paths such as chains, and aims to recover a relatively sparse graph of direct causal relations. On top of this graph, we only apply the derivation in Section 3.5 to local regions matching the collider pattern $V_i \to V_c \gets V_s$. As stated in lines 1611, we explicitly constrain the spouse variable $V_s$ so that it is neither a parent nor a child of $V_i$, and we exclude variables already assigned to the parent/child sets when constructing the spouse set. This eliminates cases where there is a direct edge or an obvious additional path between $V_i$ and $V_s$, so spouse relations occur only in collider structures. In such a local pattern, as long as we do not condition on the collider or its descendants, the two parents are independent under the semantics of causal graphs; Section 3.5 simply exploits this standard local property, rather than assuming that arbitrary pairs of variables are independent throughout the entire graph. Consequently, the density of the Granger heatmaps in Fig. 1 neither implies that these collider parents must be structurally dependent, nor invalidates the independence assumption in our local derivation, nor affects the correctness of $\mathcal{F}_\Psi$, the projection property, or Theorem 3.1 under their stated conditions.

---

> ### Author Response · Authors · 2025-11-20
> **Response to Questions.2 of Reviewer PUAg**
>
> We thank the valuable comments of the reviewer PUAg.
>
> > Figures 1 and 4 appear to convey different causal densities: the Granger causality heatmaps show dense interactions, whereas the DAGs discovered by the PC algorithm are sparse and partly disconnected.
>
> Response: We agree that the Granger-causality heatmaps in Fig. 1 are dense while the PC-based DAGs in Fig. 6 (Rebuttal Revision) are sparse, but this difference arises because they represent fundamentally different types of relations, rather than a contradiction between methods.
>
> (1) Fig. 1 visualizes predictive relations in the sense of Granger causality: whenever the past of one variable provides even a small improvement in predicting another variable, the Granger test assigns a non-zero value. As a result, even if there is no direct causal edge between two variables, Granger will still indicate an influence as long as information can flow indirectly through other paths. For example, in a simple chain $V_1 \to V_2 \to V_3$, the structural graph has only the two edges $V_1 \to V_2$ and $V_2 \to V_3$, but Granger analysis typically also reports an effect from $V_1$ to $V_3$, because information from $V_1$ eventually reaches $V_3$ via $V_2$. This is exactly why Fig. 1 looks “almost fully non-zero”: it reflects which past histories can be exploited to improve prediction, rather than which pairs of variables are directly connected in the structural causal model.
>
> (2) By contrast, Fig. 6 (Rebuttal Revision) shows structural causal graphs estimated by the PC algorithm, whose goal is to identify which variables are in direct dependence. PC uses conditional independence tests to remove associations induced by chains and common causes, keeping only those relations that cannot be explained away by other variables. Consequently, the resulting graphs are expected to be much sparser and may naturally contain several disconnected or weakly connected components. This sparsity is not a failure mode but a desired property of structural graphs.
>
> (3) In this light, the apparent discrepancy between Fig. 1 and Fig. 6 (Rebuttal Revision) simply reflects the difference between the two notions: the Granger graphs encode “any relation that improves prediction,” which tends to be dense; the PC graphs encode “direct causal relations,” which tend to be sparse. Thus, the density gap between Figs. 1 and 6 (Rebuttal Revision) are fully consistent with the theoretical roles of the two methods and do not indicate an inconsistency in our approach.

---

> ### Author Response · Authors · 2025-11-20
> **Response to Questions.4 of Reviewer PUAg**
>
> We thank the valuable comments of the reviewer PUAg.
>
> > Which representation (Granger vs. PC) should readers regard as the underlying causal structure assumed by CAIFormer?
>
> Response: The causal structure actually assumed and used by CAIFormer comes entirely from the DAGs estimated by the PC algorithm (Fig. 6 of Rebuttal Revision), rather than from the Granger-causality heatmaps in Fig. 1. The Granger plots in Fig. 1 are included only for motivation, to illustrate that multivariate time series typically exhibit rich predictive interactions across variables. They are not treated as structural causal graphs and do not enter the mask construction, module design, theoretical derivations, or training procedure of CAIFormer. In contrast, the causal segmentation in CAIFormer (e.g., the definitions of DCS and CCS), the function class $\mathcal{F}_\Psi$, and all theoretical analyses in Section 3 are defined strictly based on the DAG learned by PC. Therefore, when interpreting the causal assumptions underlying CAIFormer, readers should regard the PC-based graphs in Fig. 6 of Rebuttal Revision as the underlying causal structure.

---

> ### Author Response · Authors · 2025-11-20
> **Response to Questions.6 of Reviewer PUAg**
>
> We thank the valuable comments of the reviewer PUAg.
>
> > Could the authors provide a discussion comparing the “hard” discarding of the SCS block to a “soft” alternative? For example, one could still feed the SCS variables into a separate block but apply strong regularization (e.g., a large L1 penalty on its output weights) or use an attention mechanism that strongly suppresses attention to this block. This would clarify whether complete removal is truly necessary or whether a less drastic approach would suffice.
>
> Response: (1) Our choice of “hard” discarding the SCS block is mainly motivated by how SCS variables are defined in the structural causal model. Under the structural causal model, SCS variables do not have a causal effect on the target. In our framework, they are therefore treated as “non-essential signals.” Based on this causal interpretation, we adopt the most direct and simple strategy, namely, removing SCS variables from the model input, so as to avoid additional spurious correlations introduced by them.
>
> (2) In response to the reviewer’s suggestion of a “soft” alternative, we also perform an explicit test (Appendix M of Rebuttal Revision). Concretely, in addition to the existing ESPB, DCSPB, and CCSPB, we added an extra SCSPB module that receives and models only the SCS variables, and then fuses the predictions from all four blocks. The results on the ETTh1 and ETTm1 datasets are shown below:
>
> | Model           | Metric | ETTh1-96 | ETTh1-192 | ETTh1-336 | ETTh1-720 | ETTm1-96 | ETTm1-192 | ETTm1-336 | ETTm1-720 |
> | --------------- | ------ | -------- | --------- | --------- | --------- | -------- | --------- | --------- | --------- |
> | CAIFormer       | MSE    | 0.372    | 0.429     | 0.464     | 0.495     | 0.327    | 0.361     | 0.391     | 0.449     |
> | CAIFormer       | MAE    | 0.399    | 0.426     | 0.449     | 0.483     | 0.364    | 0.377     | 0.402     | 0.437     |
> | CAIFormer+SCSPB | MSE    | 0.380    | 0.433     | 0.469     | 0.499     | 0.332    | 0.369     | 0.401     | 0.458     |
> | CAIFormer+SCSPB | MAE    | 0.401    | 0.433     | 0.452     | 0.487     | 0.367    | 0.381     | 0.408     | 0.4443    |
>
> We observe that after adding SCSPB and fusing the four blocks, the MSE/MAE consistently become slightly worse on both datasets and across different forecasting horizons. This indicates that, within our current framework, explicitly exploiting SCS variables in a “soft” manner does not bring improvements and may introduce additional noise or spurious correlations, which is consistent with our causal motivation.
>
> (3) In addition, our all-to-one ablation experiment further shows that hard removal of spurious variables is beneficial in practice. On the ETTh1 dataset, the MLP-based experiment demonstrates that using only causally related variables (ES, DCS, CCS) achieves comparable or slightly better performance than using all variables for all targets, reducing the average MSE from 0.395 to 0.392 (Table 5). This result, obtained with a lightweight model, further supports the role of causal decomposition in reducing spurious dependencies and suggests that hard discarding SCS does not weaken the model capacity; instead, it helps stabilize prediction and improves generalization in our current design.

---

> ### Author Response · Authors · 2025-11-25
> **Analysis from the perspective of textbook definitions**
>
> Our use of the PC algorithm is exactly aligned with the causal graphical framework in [1]. The key concepts are:
>
> - Markov compatibility (Definition 1.2.2 of [1], the indices discussed in this section are all from [1]): a distribution $P$ is compatible with a DAG $G$ if it factorizes according to the parents in $G$.
> - d-separation (Definition 1.2.3) and observational equivalence (Theorem 1.2.8): d-separation in $G$ is sound and complete for the conditional independencies in $P$; two DAGs have the same observational implications if they share the same skeleton and v-structures.
> - Minimality (Section 2.3, “minimal potential structure”): a causal graph is minimal if no edge can be removed without contradicting the independencies in $P$.
> - Stability/faithfulness (Section 2.4): only independencies that persist under small parameter perturbations are treated as structural; “accidental” independencies from fine-tuned parameters are excluded.
> - Causal effect definition (Definition 2.3.6): $C$ is said to have a causal effect on $E$ only if every minimal causal structure consistent with the observed distribution contains a directed path from $C$ to $E$.
>
> The PC algorithm is the standard, constraint-based realization of exactly these notions: it uses conditional independencies to recover a minimal, faithful, Markov-compatible Markov equivalence class, represented as a CPDAG. Within Pearl’s framework, this equivalence class is precisely the set of causal structures that can be identified from observational data; no causal discovery method can legitimately claim more without additional assumptions (e.g., interventions).
>
> **To Weakness.3 (reliability/interpretability/faithfulness):**
>
> In [1], a causal DAG is identifiable from observational data (up to Markov equivalence) under four standard conditions: Markov compatibility, d-separation, minimality, and stability/faithfulness (Defs. 1.2.2, 1.2.3, Sec. 2.3–2.4). The PC algorithm is exactly the canonical constraint-based realization of this framework: it removes edges until minimality holds while preserving the conditional independencies implied by d-separation for a faithful, Markovian distribution. The resulting CPDAG therefore represents the Markov equivalence class of minimal structures compatible with the observed stable independence pattern, rather than an arbitrary single graph. Within Pearl’s theory, this class is precisely the causal content that is identifiable from noisy observational data; finite-sample noise affects the estimation of independencies, but not the underlying causal semantics.
>
> **To Questions.5 (robustness to edge perturbations):**
>
> By Theorem 1.2.8 in [1], all DAGs in the same Markov equivalence class share the same skeleton and v-structures and thus encode exactly the same set of conditional independencies. In this framework, identifiable causal information is determined by that independence model, not by any particular orientation choice within the class. Consequently, perturbations of individual edges that do not change the induced skeleton and v-structures leave the theoretically justified causal claims—and hence the causal guidance provided to CAIFormer—unchanged; only perturbations that alter the underlying independencies correspond to a genuinely different causal hypothesis.
>
> [1] Pearl J. *Causality*. 2nd ed. Cambridge University Press; 2009.

---

### Author Response · Authors · 2025-11-25
**Initial global response to all reviewers**

We sincerely thank all reviewers for their careful reading, constructive feedback, and thoughtful suggestions. We have revised the manuscript accordingly, and all changes are in red font. Below, we summarize the major updates:

(Reviewer QtHD): Move the relevant work from the appendix to Sections 2.2-2.3.  Add discussion about TimePro (Line 116-119). Add descriptions to the caption in Figure 2. (Line 336-337).

(Reviewer QtHD, qzzU): Re-editing the analysis in section 3.4 (Line 240-255).

(Reviewer QtHD): Streamline the wording and split it into three shorter paragraphs to improve readability. (Line 374-382) Detailed set definitions and their corresponding formulas are moved to Appendix R.

(Reviewer qzzU): Add the description of the Transformer architecture. (Line 391-395 and Line 404-409)

(Reviewer PUAg, QtHD): Add baselines to Tables 1 and 12.

(Reviewer PUAg): Provide a discussion comparing the “hard” discarding of the SCS block to a “soft” alternative in Appendix M.

(Reviewer Si3J): Provide the runtime analysis of causal discovery methods.

Additionally, we add an index to the appendix in the corresponding section of the main text (Line 472-374, 471-473, 483-485, 523-526).

---

> ### Author Response · Authors · 2025-11-27
>
> Dear Reviewers:
>
> We hope the responses have fully addressed your concerns. If you have any further questions or require clarification, we would be happy to provide them.
>
> Best Regards,
>
> The Authors

---

### Author Response · Authors · 2025-12-02
**Rebuttal Summary for PCs, SAC, and ACs**

Dear PCs, SAC, and ACs,

Thank you very much for overseeing our submission.
 During the rebuttal period, we carefully examin every concern raised by the reviewers and provid detailed clarifications, additional experiments, and expand explanations in both the main text and the appendix. Below we provide a concise reviewer-wise summary of how we directly address each set of comments.

**The reviewer PUAg** raised several conceptual concerns regarding the causal assumptions, the collider/spouse treatment, and the robustness of the causal discovery step. In rebuttal, we (i) clarify the theoretical motivation and the intend local assumptions, (ii) emphasize existing experiments already included in the submission (e.g., PC stability tests and causal-subset ablations), (iii) add new robustness analyses (e.g., graph-perturbation experiments), (iv) compare PC with alternatives frequently discussed in causal discovery (FCI and PCMCI), and (v) expand the SOTA baseline table.

------

**The reviewer  QtHD**’s concerns center on clarity of presentation and positioning relative to prior work. In rebuttal, we add an explicit discussion situating TimePro within the “variable-based” paradigm and clarify that its treatment of variable relationships (i.e., not distinguishing causal roles) is already discussed in our main text. We also revised several passages for readability, improve explanations of technical definitions, and move redundant formalism to the appendix for clarity.

------

**The reviewer qzzU** highlight three themes: clarity of causal-path exposition, transparency of the model pipeline, and the conceptual framing of causal discovery. In rebuttal, we (i) compress the original long causal-path enumeration into a clearer high-level summary, (ii) add concise explanatory text describing how ESPB forms temporal patch sequences and how DCSPB/CCSPB operate along the variable axis, and (iii) provide expanded discussion on how causal discovery is applied to the full multivariate series rather than a single time index.

------

**The reviewer Si3J** ask about robustness to causal-discovery errors, computational overhead, and nonlinear variants. In rebuttal, we (i) summary stability tests already present in the paper (graph-consistency across splits, perturbation robustness, multiple discovery algorithms), (ii) add explicit runtime comparisons for PC/FCI/PCMCI, (iii) explore ensemble/bootstrapped causal discovery, and (iv) report nonlinear-PC results on high-dimensional datasets.

------

Since the rebuttal was posted, there has been very limited follow-up discussion among the reviewers. Among the four reviewers, only Reviewer Si3J respond after reading our rebuttal and explicitly state that our clarifications address most of their concerns. We would be very grateful if you could take our rebuttal and these updates into account when forming your final recommendation.

We fully understand that this situation may have imposed additional workload on you, and we would like to express our sincere appreciation for your time and effort.

If any part of the reviews or our rebuttal is unclear, we would be more than happy to provide further clarification or engage in additional discussion.

Thank you again for your consideration and for your service to the community.

Best regards,

Authors

---

### Meta-Review · Area_Chair_CYwD · 2026-01-07

**Summary:**

This paper proposes a causal informed transformer (CAIFormer) for multivariate time series forecasting. While the paper presents an interesting idea, based on the reviews, the authors’ rebuttal, and the reviewers’ responses, I would recommend weak rejection for ICLR 2026.

Most reviewers are negative towards this submission even after the rebuttal. Multiple reviewers’ core concern is that the method’s key “causal” benefit depends on PC discovery and assumptions like causal faithfulness, which are often violated in real MTS settings.

In particular, the evaluation experiment is still very limited (only two pages after revision), and several key empirical validations are missing. Most notably, the ablation studies are not conducted consistently across all benchmark datasets, which makes it difficult to assess the actual contribution of those combined modules. Given the complexity of the framework and its reliance on causal role decomposition, such comprehensive ablations are essential.

As also noted by multiple reviewers (QtHD, qzzU), the paper’s presentation lacks clarity in several places, and the proposed method is difficult to follow, particularly in the Method section. The overall presentation of the paper requires further improvement for better readability.

**Reviewer Concerns:**

These concerns are somehow addressed by the rebuttal: (1) positioning with respect to closely related work; (2) clarity of the proposed method; (3) robustness analysis.

However, to me, the following concerns are still outstanding after reading the rebuttal and the revised manuscript: (1) incomplete experimental validation; (2) causal discovery procedures with strong assumptions; (3) overall presentation and readability.

**Reviewer Scores:**

All reviewers will tend to maintain their initial scores, 6, 4, 4, 4.

---

### Decision · Program_Chairs · 2026-01-26

Reject